# DIMENSION AGNOSTIC NEURAL PROCESSES

**Hyungi Lee, Chaeyun Jang, Dong bok Lee, Juho Lee**
KAIST
{lhk2708, jcy9911, markhi, juholee}@kaist.ac.kr

## ABSTRACT

Meta-learning aims to train models that can generalize to new tasks with limited labeled data by extracting shared features across diverse task datasets. Additionally, it accounts for prediction uncertainty during both training and evaluation, a concept known as uncertainty-aware meta-learning. Neural Process (NP) is a well-known uncertainty-aware meta-learning method that constructs implicit stochastic processes using parametric neural networks, enabling rapid adaptation to new tasks. However, existing NP methods face challenges in accommodating diverse input dimensions and learned features, limiting their broad applicability across regression tasks. To address these limitations and advance the utility of NP models as general regressors, we introduce Dimension Agnostic Neural Process (DANP). DANP incorporates Dimension Aggregator Block (DAB) to transform input features into a fixed-dimensional space, enhancing the model's ability to handle diverse datasets. Furthermore, leveraging the Transformer architecture and latent encoding layers, DANP learns a wider range of features that are generalizable across various tasks. Through comprehensive experimentation on various synthetic and practical regression tasks, we empirically show that DANP outperforms previous NP variations, showcasing its effectiveness in overcoming the limitations of traditional NP models and its potential for broader applicability in diverse regression scenarios.

## 1 INTRODUCTION

In real-world datasets, there are many tasks that come with various configurations (such as input feature dimensions, quantity of training data points, correlation between training and validation data, etc.). However, each task has a limited number of data points available, making it difficult to train a model capable of robust generalization solely based on the provided training data. To tackle this issue, meta-learning aims to train a model capable of generalizing to new tasks with few labeled data by learning generally shared features from diverse training task datasets. In cases of limited labeled data for new target tasks, ensuring model trustworthiness involves accurately quantifying prediction uncertainty, which is as critical as achieving precise predictions. A meta-learning strategy that considers prediction uncertainty during training and evaluation is known as uncertainty-aware meta-learning (Nguyen & Grover, 2022; Almecija et al., 2022).

One of the well-known uncertainty-aware meta-learning methods is Neural Process (NP) (Garnelo et al., 2018a;b). NP employs meta-learning to understand the data-generation process governing the relationship between input-output pairs in meta-training and meta-validation data. Unlike the traditional approach to learning stochastic processes, where model selection from a known class, e.g. Gaussian Processes (GPs), precedes computing predictive distributions based on training data, NP constructs an implicit stochastic process using parametric neural networks trained on meta-training data. It then optimizes parameters to maximize the predictive likelihood for both the meta-train and meta-validation data. Consequently, when NP effectively learns the data-generation process solely from data, it can quickly identify suitable stochastic processes for new tasks. Thus, NP can be viewed as a data-driven uncertainty-aware meta-learning method for defining stochastic processes.

However, previous works (Gordon et al., 2020; Foong et al., 2020; Lee et al., 2020; Nguyen & Grover, 2022; Lee et al., 2023) in NP literature lack two crucial attributes essential for broad applicability across different regression tasks: 1) the ability to directly accommodate diverse input and output dimensions, and 2) the adaptability of learned features for fine-tuning on new tasks that exhibit varying input and output dimensions. Due to the absence of these two properties, it is necessary

to train each NP model separately for different dimensional tasks. These limitations hinder the utility of NP models as general regressors across diverse datasets compared to traditional stochastic processes (Lee et al., 2021). Traditional stochastic processes naturally accommodate varying input dimensions, particularly in regression tasks involving high-dimensional input features with limited training data, such as hyperparameter optimization tasks.

To tackle these limitations and advance the utilization of NP models as general regressors for a wide range of regression tasks, we introduce a novel extension of NP called Dimension Agnostic Neural Process (DANP). In DANP, we propose a module called Dimension Aggregator Block (DAB), which transforms input features of varying dimensions into a fixed-dimensional representation space. This allows subsequent NP modules to effectively handle diverse datasets and generate predictive density for the meta-validation data. We also add the Transformer architecture (Vaswani et al., 2017) alongside latent encoding layers based on the architecture of Transformer Neural Processes (TNP) (Nguyen & Grover, 2022) to enhance the model's ability to learn a wider range of features and effectively capture functional uncertainty, which can be applied across different tasks. Through experimentation on a variety of synthetic and real-world regression tasks with various situations, we show that DANP achieves notably superior predictive performance compared to previous NP variations.

## 2 BACKGROUND

### 2.1 PROBLEM SETTINGS

Let $\mathcal{X}$ be an input space defined as $\bigcup_{i\in\mathbb{N}} \mathcal{X}_i$ with each $\mathcal{X}_i \subseteq \mathbb{R}^i$ for all $i \in \mathbb{N}$. Similarly, let $\mathcal{Y} = \bigcup_{i\in\mathbb{N}} \mathcal{Y}_i$ represent the output space, where each $\mathcal{Y}_i \subseteq \mathbb{R}^i$ for all $i \in \mathbb{N}$. Let $\mathbf{T} = \{\tau_j\}_{j\in\mathbb{N}}$ be a task set drawn in i.i.d. fashion from a task distribution $p_{\text{task}}(\tau)$. Given two dimension mapping functions $u, v : \mathbb{N} \to \mathbb{N}$, each task $\tau_j$ comprises a dataset $\mathcal{D}_j = \{\mathbf{d}_{j,k}\}_{k=1}^{n_j}$, where $\mathbf{d}_{j,k} = (\mathbf{x}_{j,k}, \mathbf{y}_{j,k}) \in \mathcal{X}_{u(j)} \times \mathcal{Y}_{v(j)}$ represents an input-output data pair, along with an index set $c_j \subsetneq [n_j]$ where $[m] := \{1, \dots, m\}$ for all $m \in \mathbb{N}$. We assume elements in $\mathcal{D}_j$ are i.i.d. conditioned on some function $f_j$. Here, the set of indices $c_j$ defines the context set $\mathcal{D}_{j,c} := \{\mathbf{d}_{j,k}\}_{k\in c_j}$. Similarly, the target set is defined as $\mathcal{D}_{j,t} := \{\mathbf{d}_{j,k}\}_{k\in t_j}$ where $t_j := [n_j] \setminus c_j$. We aim to meta-learn a collection of random functions $f_j : \mathcal{X}_{u(j)} \to \mathcal{Y}_{v(j)}$, where each function within this set effectively captures and explains the connection between input $x$ and output $y$ pairs. For any given meta-training task $\tau_j$, we can regard its context set $\mathcal{D}_{j,c}$ as the meta-training set and its target set $\mathcal{D}_{j,t}$ as the meta-validation set.

### 2.2 NEURAL PROCESSES

For the previous NP variants, their objective was to meta-learn a set of random functions $f_j : \mathcal{X}_{d_{\text{in}}} \to \mathcal{Y}_{d_{\text{out}}}$, for some fixed $d_{\text{in}}, d_{\text{out}} \in \mathbb{N}$, which is equal to the situation where dimension mapping functions $u, v$ are constant functions, i.e., $u(j) = d_{\text{in}}$ and $v(j) = d_{\text{out}}$ for all $j \in \mathbb{N}$. In this context, to select a suitable random function $f_j$ for the task $\tau_j$, NPs meta-learns how to map the context set $\mathcal{D}_{j,c}$ to a random function $f_j$ that effectively represents both the context set $\mathcal{D}_{j,c}$ and the target set $\mathcal{D}_{j,t}$. This entails maximizing the likelihood for both meta-training and meta-validation datasets within an uncertainty-aware meta-training framework. The process involves learning a predictive density that maximizes the likelihood using the following equation:

$$p(\mathbf{Y}_j|\mathbf{X}_j, \mathcal{D}_{j,c}) = \int \left[ \prod_{k\in[n_j]} p(\mathbf{y}_{j,k}|f_j, \mathbf{x}_{j,k}) \right] p(f_j|\mathcal{D}_{j,c}) \mathrm{d}f_j, \tag{1}$$

where $\mathbf{X}_j = \{\mathbf{x}_{j,k}\}_{k=1}^{n_j}$ and $\mathbf{Y}_j = \{\mathbf{y}_{j,k}\}_{k=1}^{n_j}$. In line with our discussion in Section 2.1, we make the assumption that given the random function $f_j$, the outputs collection $\mathbf{Y}_j$ are i.i.d. Employing the Gaussian likelihood and parameterizing $f_j$ with **latent variable** $r_j \in \mathbb{R}^{d_j}$, Eq. 1 reduces to,

$$p(\mathbf{Y}_j|\mathbf{X}_j, \mathcal{D}_{j,c}) = \int \left[ \prod_{k\in[n_j]} \mathcal{N}\left(\mathbf{y}_{j,k}|\mu_{r_j}(\mathbf{x}_{j,k}), \mathrm{diag}(\sigma^2_{r_j}(\mathbf{x}_{j,k}))\right) \right] p(r_j|\mathcal{D}_{j,c}) \mathrm{d}r_j, \tag{2}$$

where $\mu_{r_j} : \mathcal{X}_{d_{\text{in}}} \to \mathcal{Y}_{d_{\text{out}}}$ and $\sigma^2_{r_j} : \mathcal{X}_{d_{\text{in}}} \to \mathbb{R}^{d_{\text{out}}}_+$. Then different NP variants aim to effectively design the model structures of the encoder, denoted as $f_{\text{enc}}$, and the decoder, de-

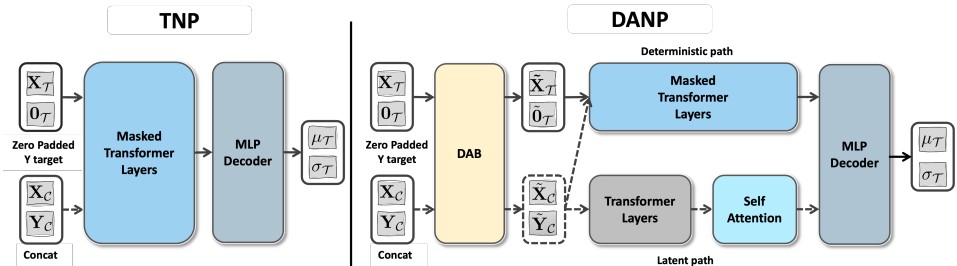

**Figure 1:** Model comparison between TNP and DANP. While TNP (Nguyen & Grover, 2022) solely employs a deterministic pathway with Masked Transformer layers, DANP incorporates both DAB and an extra latent pathway alongside Transformer layers and a Self-Attention layer.

noted as $f_{\text{dec}}$. These components are responsible for describing the distributions $p(r_j|\mathcal{D}_{j,c})$ and $\mathcal{N}\left(\mathbf{y}_{j,k}|\mu_{r_j}(\mathbf{x}_{j,k}), \text{diag}(\sigma^2_{r_j}(\mathbf{x}_{j,k}))\right)$, respectively.

NP variations can be roughly categorized into two classes based on their approach to modeling $p(r_j|\mathcal{D}_{j,c})$: 1) Conditional Neural Processes (CNPs) (Garnelo et al., 2018a; Gordon et al., 2020; Nguyen & Grover, 2022) and 2) (latent) NPs (Garnelo et al., 2018b; Foong et al., 2020; Lee et al., 2023). CNPs establish a deterministic function, called *deterministic path*, from $\mathcal{D}_{j,c}$ to $r_j$ and represent $p(r_j|\mathcal{D}_{j,c})$ as a discrete point measure, expressed as:

$$p(r_j|\mathcal{D}_{j,c}) = \delta_{\bar{r}_j}(r), \quad \bar{r}_j = f_{\text{enc}}(\mathcal{D}_{j,c}; \phi), \tag{3}$$

where $\phi$ is the parameter of $f_{\text{enc}}$. In contrast, NPs address functional uncertainty or model uncertainty in modeling $p(r_j|\mathcal{D}_{j,c})$. Typically, they employ a variational posterior $q(r_j|\mathcal{D}_{j,s})$, called *latent path*, to approximate $p(r_j|\mathcal{D}_{j,s})$ for any subset $\mathcal{D}_{j,s} \subseteq \mathcal{D}_j$, defined as:

$$q(r_j|\mathcal{D}_{j,s}) = \mathcal{N}(r_j|\mathbf{m}_{\mathcal{D}_{j,s}}, \text{diag}(\mathbf{s}^2_{\mathcal{D}_{j,s}})), \quad (\mathbf{m}_{\mathcal{D}_{j,s}}, \mathbf{s}^2_{\mathcal{D}_{j,s}}) = f_{\text{enc}}(\mathcal{D}_{j,s}; \phi). \tag{4}$$

Then both of the classes decode the mean and variance of the input $\mathbf{x}_{j,k}$ as follows:

$$(\mu_{r_j}(\mathbf{x}_{j,k}), \sigma^2_{r_j}(\mathbf{x}_{j,k})) = f_{\text{dec}}(\mathbf{x}_{j,k}, r_j; \psi), \tag{5}$$

where $f_{\text{dec}}$ is another feedforward neural net $\psi$.

Training CNPs involves maximizing the average predictive log-likelihood across meta-training tasks $\tau_j$, i.e. $\mathbb{E}_{\tau_j}[\log p(\mathbf{Y}_j|\mathbf{X}_j, \mathcal{D}_{j,c})]$. On the other hand, NPs are typically trained by maximizing the Evidence Lower BOund (ELBO), which is expressed as:

$$\mathbb{E}_{\tau_j}[\log p(\mathbf{Y}_j|\mathbf{X}_j, \mathcal{D}_{j,c})] \geq \mathbb{E}_{\tau_j}\left[\sum_{k \in [n_j]} \mathbb{E}_{q(r_j|\mathcal{D}_j)}[\log \mathcal{N}_{j,k}] - \text{KL}[q(r_j|\mathcal{D}_j)|q(r_j|\mathcal{D}_{j,c})]\right], \tag{6}$$

where $\mathcal{N}_{j,k}$ is a shorthand for $\mathcal{N}\left(\mathbf{y}_{j,k}|\mu_{r_j}(\mathbf{x}_{j,k}), \text{diag}(\sigma^2_{r_j}(\mathbf{x}_{j,k}))\right)$.

There have been several attempts to enhance the flexibility of the encoder $f_{\text{enc}}$ to improve the predictive performance (Garnelo et al., 2018a; Kim et al., 2018; Gordon et al., 2020; Nguyen & Grover, 2022). In this study, we adopt the state-of-the-art TNP model as our base structure, which leverages masked self-attention layers as encoding layers and also belongs to the category of CNPs variants.

## 3 DIMENSION AGNOSTIC NEURAL PROCESS

As we mentioned in Section 1 and Section 2.2, the limitation of the previous NP variants is that they primarily handle scenarios where the input space and output space are confined to $\mathcal{X}_{d_{\text{in}}}$ and $\mathcal{Y}_{d_{\text{out}}}$ for the fixed $d_{\text{in}}, d_{\text{out}} \in \mathbb{N}$. To address this constraint, we introduce a novel NP variant called DANP. Initially, we elucidate how the predictive density evolves when the dimension mapping functions $u, v$ are not constant in Section 3.1. Subsequently, we expound on how we can convert input features of varying dimensions into a fixed-dimensional representation space utilizing the DAB module in Section 3.2. Finally, we detail the strategies employed to augment the model's capacity for learning diverse features in Section 3.3.

### 3.1 A MODEL OF PREDICTIVE DENSITY WITH VARYING DIMENSIONS

Given our assumption that all the tasks may have varying input and output dimensions, we write $\mathcal{D}_j = \{\mathbf{d}_{j,k}\}_{k=1}^{n_j}$ where $\mathbf{x}_{j,k} := [\mathbf{x}_{j,k}^1 \cdots \mathbf{x}_{j,k}^{u(j)}] \in \mathbb{R}^{u(j)}$ and $\mathbf{y}_{j,k} := [\mathbf{y}_{j,k}^1 \cdots \mathbf{y}_{j,k}^{v(j)}] \in \mathbb{R}^{v(j)}$. Given the context $\mathcal{D}_{j,c}$, the equation for the predictive density $p(\mathbf{Y}_j | \mathbf{X}_j, \mathcal{D}_{j,c})$ remains the same with Eq. 2. However, due to the varying dimensions, the computation of both the likelihood $\mathcal{N}_{j,k}$ and the context representation posterior $p(r_j | \mathcal{D}_{j,c})$ poses a challenge. In a fixed dimension setting, only the size of the context varies across different tasks, and this could be processed by choosing $f_{\text{enc}}$ as a permutation-invariant set functions (Zaheer et al., 2017). However, in our scenario, for two different tasks $\tau_1$ and $\tau_2$, a single encoder should compute,

$$f_{\text{enc}}(\mathcal{D}_{1,c}; \phi) = f_{\text{enc}}(\{((\mathbf{x}_{1,k}^1, \ldots, \mathbf{x}_{1,k}^{u(1)}), (\mathbf{y}_{1,k}^1, \ldots, \mathbf{y}_{1,k}^{v(1)}))\}_{k \in c_1}; \phi), \tag{7}$$

$$f_{\text{enc}}(\mathcal{D}_{2,c}; \phi) = f_{\text{enc}}(\{((\mathbf{x}_{2,k}^1, \ldots, \mathbf{x}_{2,k}^{u(2)}), (\mathbf{y}_{2,k}^1, \ldots, \mathbf{y}_{2,k}^{v(2)}))\}_{k \in c_2}; \phi). \tag{8}$$

The existing permutation-invariant encoder can process when $|c_1| \neq |c_2|$, it cannot handle when $(u(1), v(1)) \neq (u(2), v(2))$, because this disparity happens at the lowest level of the encoder, typically implemented with a normal feed-forward neural network. The standard architecture in the previous NP models is to employ a Multi-Layer Perceptron (MLP) taking the concatenated inputs, for instance,

**Figure 2:** The overview of DAB module. A DAB can encode and decode inputs and outputs of varying dimensions.

$$f_{\text{enc}}(\mathcal{D}_{j,c}; \phi) = \frac{1}{|c_j|} \sum_{k \in c_j} \text{MLP}(\text{concat}(\mathbf{x}_{j,k}, \mathbf{y}_{j,k})), \tag{9}$$

and the MLP encoder can only process fixed-dimensional inputs. A similar challenge also applies to the decoder $f_{\text{dec}}$ computing the predictive mean $\mu_{r_j}(\mathbf{x}_{j,k})$ and variance $\sigma_{r_j}^2(\mathbf{x}_{j,k})$. To address this challenge, we need a new neural network that is capable of handling sets of varying dimensions.

### 3.2 DIMENSION AGGREGATOR BLOCK

To enable our model to process sets with elements of varying dimensions, we introduce a module called Dimension Aggregator Block (DAB). The module can encode inputs with varying feature dimensions into a fixed-dimensional representation and varying dimensional representations, which we will describe individually below. The overall architecture is depicted in Fig. 2.

**Encoding $\mathbf{x}$ into a fixed dimensional representation.** Consider an input $(\mathbf{x}, \mathbf{y})$ where $\mathbf{x} = [\mathbf{x}^1 \ldots \mathbf{x}^{d_x}] \in \mathbb{R}^{d_x}$ and $\mathbf{y} = [\mathbf{y}^1 \ldots \mathbf{y}^{d_y}] \in \mathbb{R}^{d_y}$. To maintain sufficient information for each dimension of the input data, even after it has been mapped to a fixed-dimensional representation, we initially expand each of the $d_x + d_y$ dimensions of the input data to $d_r$ dimensions using learnable linear projection $\mathbf{w} \in \mathbb{R}^{d_r}$ as follows:

$$[\tilde{\mathbf{x}}, \tilde{\mathbf{y}}] = \mathbf{w}[\text{concat}(\mathbf{x}, \mathbf{y})]^\top \in \mathbb{R}^{d_r \times (d_x + d_y)}. \tag{10}$$

When encoding a point in the target set $\mathcal{D}_{j,t}$ without the label $\mathbf{y}$, we simply encode the zero-padded value $\text{concat}(\mathbf{x}, \mathbf{0})$. Next, following Vaswani et al. (2017), we incorporate cosine and sine positional encoding to distinguish and retain positional information for each input dimension as follows:

$$\text{PEX}_{(2i,j)} = \sin(j/P(i)), \ \text{PEX}_{(2i+1,j)} = \cos(j/P(i)), \tag{11}$$

$$\text{PEY}_{(2i,l)} = \cos(l/P(i)), \ \text{PEY}_{(2i+1,l)} = \sin(l/P(i)), \tag{12}$$

where $P(i) = 10000^{2i/d_r}$ and PEX, PEY represent the positional encoding for $\mathbf{x}$ and $\mathbf{y}$ respectively. Here, $j \in [d_x]$ and $l \in [d_y]$ respectively denote the position indices of $\tilde{\mathbf{x}}$ and $\tilde{\mathbf{y}}$, while $i \in \lfloor \frac{d_r}{2} \rfloor$

represents a dimension in the representation space. Since the dimensions of both $\mathbf{x}$ and $\mathbf{y}$ vary and the representation must be divided between corresponding positions in $\mathbf{x}$ and $\mathbf{y}$, e.g., $\mathbf{x}^1$ and $\mathbf{y}^1$, we use distinct positional embeddings, PEX for $\mathbf{x}$ and PEY for $\mathbf{y}$. After adding positional encoding to $(\tilde{\mathbf{x}}, \tilde{\mathbf{y}})$, we further compute,

$$(\tilde{\mathbf{x}}, \tilde{\mathbf{y}}) = \text{SelfAttn}(\text{concat}(\tilde{\mathbf{x}}, \tilde{\mathbf{y}}) + (\text{PEX}, \text{PEY})), \tag{13}$$

where SelfAttn indicates a self-attention layer (Vaswani et al., 2017). Here, we can regard $(\tilde{\mathbf{x}}, \tilde{\mathbf{y}})$ before the self-attention layer as akin to context-free embeddings (Rong, 2014), because they remain unchanged when position and value are fixed. Conversely, $(\tilde{\mathbf{x}}, \tilde{\mathbf{y}})$ after the Self-Attention layer can be likened to contextual embeddings (Devlin et al., 2018), which is known as advanced embedding compared to context-free embedding, as the final representation may vary depending on alterations in value and position across other dimensions due to the interaction through the self-attention layer. Then, we employ average pooling for the $\tilde{\mathbf{x}}$ to integrate all the information across varying dimensions in $\mathbf{x}$ into a fixed-dimensional representation, i.e., $\hat{\mathbf{x}} = \text{AvgPool}(\tilde{\mathbf{x}}) \in \mathbb{R}^{d_r}$, with AvgPool representing the average pooling operation across the feature dimension.

**Handling variable number of outputs for $\mathbf{y}$.** The DAB should produce representations that can be used in the decoder later to produce outputs with varying dimensions. To achieve this, unlike for the $\mathbf{x}$, we keep the sequence $\tilde{\mathbf{y}}$ without average pooling. Instead, for each $\ell = 1, \ldots, d_y$, we concatenate $\hat{\mathbf{x}}$ and $\tilde{\mathbf{y}}^l$ and put into the decoder to get the predictive mean and variances. The dimension of the encoding for $\mathbf{y}$ from the DAB would be the same as original dimension of $\mathbf{y}$. Note that this is not like the sequential decoding in autoregressive models, and is possible because we know the dimension of $\mathbf{y}$ before we actually have to decode the representation.

### 3.3 LEARNING MORE GENERAL FEATURE UTILIZING LATENT PATH

Let $\tilde{\mathcal{D}}_j := (\hat{\mathbf{x}}_{j,k}, (\tilde{\mathbf{y}}_{j,k}^l)_{l=1}^{v(j)})_{k=1}^{n_j}$ be the representations obtained by DAB for the dataset $\mathcal{D}_j$. The next step involves computing the predictive density using the encoder and decoder structure. Here, we employ TNP (Nguyen & Grover, 2022), a variant of CNPs, as our base model structure. In TNP, Masked Transformer layers are utilized as the encoder for the deterministic path, while a simple MLP structure serves as the decoder. To improve the model's capacity to learn generally shared features across various tasks and effectively capture functional uncertainty, we introduce a new latent path comprising Transformer layers and a Self-Attention layer alongside the single deterministic path encoder. In specific, we pass the entire $\tilde{\mathcal{D}}_j$ into Masked Transformer layers to make deterministic parameter $r_j^{\text{det}}$ as follows:

$$\mathbf{z}_{j,k}^l = \text{concat}(\hat{\mathbf{x}}_{j,k}, \tilde{\mathbf{y}}_{j,k}^l) \in \mathbb{R}^{2d_r}, \tag{14}$$

$$r_j^{\text{det}} = \text{MTFL}(\text{concat}(\{\{\mathbf{z}_{j,k}^l\}_{l=1}^{v(j)}\}_{k=1}^{n_j}), M_j) \in \mathbb{R}^{v(j)n_j \times 2d_r}, \tag{15}$$

where MTFL denotes Masked Transformer layers with mask $M_j$, and $\text{concat}(\{\{\mathbf{z}_{j,k}^l\}_{l=1}^{v(j)}\}_{k=1}^{n_j})$ indicate concatenation operation which concatenate $\mathbf{z}_{j,k}^l$ for all $l \in [v(j)]$ and $k \in [n_j]$. In this context, for all $l_1, l_2 \in [v(j)]$, the mask $M_j \in \mathbb{R}^{v(j)n_j \times v(j)n_j}$ assigns a value of 1 to the index $(l_1 k_1, l_2 k_2)$ if both $k_1$ and $k_2$ are elements of $c_j$, or if $k_1$ is in $t_j$ and $k_2$ is in $c_j$; otherwise, it assigns a value of 0.

For the latent path, we only pass context set $\tilde{\mathcal{D}}_{j,c}$ through Transformer layers, followed by one self-attention and MLP operation to determine the latent parameter $r_j^{\text{lat}}$ as follows:

$$\bar{r}_j^{\text{lat}} = \text{AvgPool}(\text{SelfAttn}(\text{TL}(\text{concat}(\{\{\mathbf{z}_{j,k}^l\}_{l=1}^{v(j)}\}_{k \in c_j})))), \tag{16}$$

$$(\mathcal{D}_{j,c}, \mathbf{s}_{\mathcal{D}_{j,c}}^2) = \text{MLP}(\bar{r}_j^{\text{lat}}), \tag{17}$$

$$r_j^{\text{lat}} \sim q(r_j^{\text{lat}}|\tilde{\mathcal{D}}_{j,c}) = \mathcal{N}(r_j^{\text{lat}}|\mathcal{D}_{j,c}, \text{diag}(\mathbf{s}_{\mathcal{D}_{j,c}}^2)), \tag{18}$$

where TL denotes Transformer layers. Finally, we concatenate the deterministic parameter $r_j^{\text{det}}$ and latent parameter $r_j^{\text{lat}}$ to make the final parameter $r_j$ before forwarding them to the decoder module. Then, by utilizing a variational posterior with the latent path, our training objective transforms into

$$\mathbb{E}_{\tau_j}[\log p(\mathbf{Y}_j|\mathbf{X}_j, \mathcal{D}_{j,c})] \geq \mathbb{E}_{\tau_j}\left[\sum_{k \in [n_j]} \mathbb{E}_{q(r_j^{\text{lat}}|\mathcal{D}_j)}[\log \mathcal{N}_{j,k}] - \text{KL}[q_j\|q_{j,c}]\right], \tag{19}$$

where $q_j$ and $q_{j,c}$ denotes $q(r_j^{\text{lat}}|\mathcal{D}_j)$ and $q(r_j^{\text{lat}}|\mathcal{D}_{j,c})$, respectively. Refer to Fig. 1 to observe the contrast in the architecture between TNP and DANP.

## 4 RELATED WORKS

**Neural Processes**  The first NPs model, called CNP (Garnelo et al., 2018a), utilized straightforward MLP layers for both its encoder and decoder. Similarly, NP (Garnelo et al., 2018b) adopted MLP layers but introduced a global latent variable to capture model uncertainty, marking an early attempt to address uncertainty in NP frameworks. Conditional Attentive Neural Process (CANP) and Attentive Neural Process (ANP) (Kim et al., 2018) are notable for incorporating attention mechanisms within the encoder, enhancing the summarization of context information relevant to target points. Building on these ideas, TNP (Nguyen & Grover, 2022) employs masked transformer layers in its encoder, delivering state-of-the-art performance among NPs across multiple tasks. Louizos et al. (2019) introduced a variant that used local latent variables instead of a global latent variable to improve the model's ability to capture uncertainty. Following this, Bootstrapping Attentive Neural Process (BANP) (Lee et al., 2020) proposed the residual bootstrap method (Efron, 1992), making NPs more robust to model misspecification. Lastly, Martingale Posterior Attentive Neural Process (MPANP) (Lee et al., 2023) addressed model uncertainty with the martingale posterior (Fong et al., 2021), offering a modern alternative to traditional Bayesian inference methods. Refer to Appendix B to see a more detailed review of previous Neural Processes works.

**Neural Diffusion Process**  Similar to DANP, there are prior works (Liu et al., 2020; Kossen et al., 2021; Dutordoir et al., 2023) that utilize bi-dimensional attention blocks to facilitate more informative data feature updates or to ensure the permutation invariance property both at the data-instance level and the dimension level. Specifically, Neural Diffusion Process (NDP) (Dutordoir et al., 2023) employs bi-dimensional attention blocks to guarantee permutation invariance both at the data and dimension levels, naturally leading to dimension-agnostic properties. However, NDP has a structural limitation in that it is only partially dimension-agnostic for $x$ when $y = 1$, and is not dimension-agnostic for other combinations. This makes it difficult to use as a general regressor. Additionally, the use of diffusion-based sampling to approximate the predictive distribution leads to significantly high computational costs during inference and results in limited likelihood performance. Refer to Appendix D.1 to see the empirical comparison between DANP and NDP.

## 5 EXPERIMENTS

In this section, we carry out a series of experiments to empirically showcase the efficacy of DANP across different situations, especially in various regression tasks and Bayesian Optimization task. To establish a robust experimental foundation, we employ five distinct variations of NP, encompassing state-of-the-art model: CANP, ANP, BANP, MPANP, and TNP. For a fair comparison, we maintain an identical latent sample size in the latent path across all models, except for deterministic models such as CANP and TNP. We marked the best performance value with **boldfaced underline**, and the second-best value with underline in each column in all tables. All the performance metrics are averaged over three different seeds and we report 1-sigma error bars for all experiments. Refer to Appendix C for experimental details containing data description and model structures.

### 5.1 GP REGRESSION

To empirically verify the effectiveness of DANP, we initially conducted GP regression experiments under various conditions: *From-scratch*, *Zero-shot*, and *Fine-tuning*. In the From-scratch scenario, we compared DANP against other baselines using fixed input dimensional GP data for both training and testing. In the Zero-shot scenario, we demonstrated the ability of DANP to generalize to different dimensional input GP data without direct training on that data. Lastly, in the Fine-tuning scenario, we conducted experiments where we fine-tuned on unseen dimensional GP data, using limited training data points, utilizing pre-trained DANP alongside other baseline models.

**From-scratch**  To validate the capability of DANP to effectively learn generally shared features and capture functional uncertainty across tasks, we first compare DANP against other baseline models in the

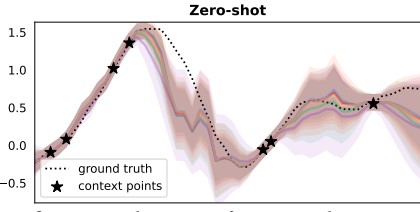 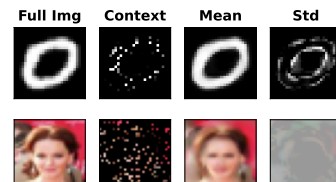

**Figure 3:** Posterior samples of DANP in (Left) the **Zero-shot** scenario with a 1-dimensional GP dataset and (Right) the **Image completion** task using the EMNIST and CelebA datasets. (Left) Black stars represent the context points and the dashed line indicates the ground truth for the target points. Each color represents different posterior samples generated from the latent path. (Right) Displays the full image, context points, predictive mean, and standard deviation of DANP for both the EMNIST and CelebA datasets. Outputs for both images are produced by a *single model*.

**Table 1:** Results of the context and target log-likelihood for the GP regression task in the From-scratch scenario. $n$D $A$ in the first row denotes the $n$-dimensional GP dataset using the $A$ kernel.

| Model | 1D RBF | | 1D Matern | | 2D RBF | | 2D Matern | |
|---|---|---|---|---|---|---|---|---|
| | context | target | context | target | context | target | context | target |
| CANP | 1.377 ±0.000 | 0.839 ±0.002 | 1.377 ±0.000 | 0.663 ±0.007 | 1.377 ±0.001 | 0.165 ±0.015 | 1.373 ±0.001 | -0.066 ±0.007 |
| ANP | 1.377 ±0.000 | 0.855 ±0.004 | 1.377 ±0.000 | 0.681 ±0.003 | 1.378 ±0.000 | 0.170 ±0.014 | 1.346 ±0.005 | -0.107 ±0.006 |
| BANP | 1.377 ±0.000 | 0.864 ±0.001 | 1.377 ±0.000 | 0.689 ±0.004 | 1.378 ±0.000 | 0.228 ±0.004 | 1.378 ±0.000 | -0.033 ±0.013 |
| MPANP | 1.376 ±0.000 | 0.856 ±0.006 | 1.376 ±0.000 | 0.679 ±0.005 | 1.378 ±0.000 | 0.242 ±0.001 | 1.376 ±0.002 | -0.029 ±0.007 |
| TNP | **1.381** ±0.000 | 0.904 ±0.003 | 1.381 ±0.000 | 0.710 ±0.001 | **1.383** ±0.000 | 0.362 ±0.001 | **1.383** ±0.000 | 0.060 ±0.002 |
| DANP (ours) | **1.381** ±0.000 | **0.921** ±0.003 | **1.382** ±0.000 | **0.723** ±0.003 | **1.383** ±0.000 | **0.373** ±0.001 | **1.383** ±0.000 | **0.068** ±0.001 |

From-scratch scenario in diverse fixed dimensional GP regression tasks. In this experiment, the meta-training datasets are produced using GP under four distinct configurations: either one-dimensional or two-dimensional input, utilizing either the RBF or Matern kernels. The results presented in Table 1 demonstrate that DANP consistently surpasses other baseline models across various settings, particularly excelling on the target dataset in terms of log-likelihood. These results prove that DANP effectively grasps common features and captures functional uncertainty, outperforming other baseline models even with various settings in fixed-dimensional GP regression tasks.

**Zero-shot**  In the Zero-shot scenario, we train a single NP model using various dimensional GP datasets and then evaluate the performance of the model on a range of GP datasets with different dimensions. Specifically, we consider two different cases: one where the training datasets include 2 and 4-dimensional GP datasets, and another where the training datasets include 2, 3, and 4-dimensional GP datasets. After training, we assess our pre-trained model on GP datasets ranging from 1 to 5 and 7 dimensions. We validate results for our model as DANP, in distinction from other baselines, is capable of simultaneously training on and inferring from datasets with diverse dimensions. Table 2a demonstrates that DANP successfully learns the shared features across different dimensional GP datasets and generalizes effectively to test datasets with the same dimensions as the training datasets. Remarkably, DANP also generalizes well to test datasets with previously unseen dimensions. For instance, the zero-shot log-likelihood for the 1-dimensional GP dataset, when DANP is trained on 2, 3, and 4-dimensional datasets, is nearly comparable to the log-likelihood of CANP with From-scratch training in Table 1. These findings suggest that DANP efficiently captures and learns general features across various tasks, allowing it to explain tasks with unseen dimensions without additional training. For further results using 2 and 3-dimensional GP datasets or different kernels during training, see Appendix D. The trends are consistent, showing that DANP generalizes well across various tasks. Refer to Fig. 3 to see the zero-shot posterior samples for the 1-dimensional GP regression task. And also refer to Appendix D.2.3 to see the results on the additional zero-shot scenarios, especially extrapolation scenarios.

**Fine-tuning**  In the fine-tuning scenario, we fine-tuned pre-trained NP models on a limited set of 160 1-dimensional GP regression tasks. For the baselines, we used pre-trained models that were trained on 2-dimensional tasks as described in the **From-scratch** experiments. For DANP, we used models pre-trained on 2, 3, and 4-dimensional tasks as mentioned in the **Zero-shot** experiments. In Table 2b, 'Full fine-tuning' refers to the process where all pre-trained neural network parameters are adjusted during fine-tuning, while 'Freeze fine-tuning' means that the shared parameters in the

**Table 2:** Log-likelihood results for the GP regression task in (a) the Zero-shot and (b) the Fine-tuning scenarios using RBF kernel. For (a), $n$D in the first column denotes the outcomes for the $n$-dimensional GP dataset. The colored cell ▉ indicates the data dimension used to pre-train DANP.

**(a) Zero-shot scenario**

| Dimension | DANP trained on 2D & 4D | | DANP trained on 2D & 3D & 4D | |
|---|---|---|---|---|
| | context | target | context | target |
| 1D RBF | 1.336 ±0.047 | 0.806 ±0.048 | **1.366** ±0.004 | **0.826** ±0.018 |
| 2D RBF | **1.383** ±0.000 | **0.340** ±0.007 | **1.383** ±0.000 | 0.335 ±0.014 |
| 3D RBF | 1.377 ±0.007 | -0.360 ±0.063 | **1.383** ±0.000 | **-0.261** ±0.025 |
| 4D RBF | 1.379 ±0.007 | -0.589 ±0.056 | **1.383** ±0.000 | **-0.568** ±0.042 |
| 5D RBF | 1.357 ±0.012 | -0.689 ±0.004 | **1.359** ±0.032 | **-0.676** ±0.004 |
| 7D RBF | 1.348 ±0.016 | -0.726 ±0.026 | **1.355** ±0.022 | **-0.723** ±0.022 |

**(b) Fine-tuning scenario**

| Method | Full fine-tuning | | Freeze fine-tuning | |
|---|---|---|---|---|
| | context | target | context | target |
| CANP | -0.305 ±0.043 | -0.495 ±0.048 | -0.061 ±0.236 | -0.386 ±0.132 |
| ANP | -0.273 ±0.121 | -0.365 ±0.093 | -0.311 ±0.034 | -0.369 ±0.037 |
| BANP | -0.292 ±0.044 | -0.379 ±0.022 | -0.131 ±0.199 | -0.193 ±0.281 |
| MPANP | -0.254 ±0.339 | -0.414 ±0.235 | -0.481 ±0.032 | -0.563 ±0.026 |
| TNP | -0.042 ±0.016 | -0.448 ±0.228 | 0.357 ±0.372 | -0.087 ±0.295 |
| DANP(ours) | **1.376** ±0.000 | **0.893** ±0.004 | **1.376** ±0.001 | **0.890** ±0.005 |

**Table 3:** Log-likelihood results for context and target values were obtained for (a) image completion tasks using the EMNIST and CelebA datasets, and (b) fine-tuning on video completion tasks. For (a), DANP was trained concurrently on both the EMNIST and CelebA datasets. For (b), † **indicates the zero-shot performance of DANP**.

**(a) Image completion**

| Model | EMNIST | | CelebA | |
|---|---|---|---|---|
| | context | target | context | target |
| CANP | 1.378 ±0.001 | 0.837 ±0.003 | 4.129 ±0.004 | 1.495 ±0.004 |
| ANP | 1.372 ±0.005 | 0.863 ±0.011 | 4.131 ±0.003 | 1.993 ±0.016 |
| BANP | 1.373 ±0.004 | 0.901 ±0.004 | 4.127 ±0.005 | **2.292** ±0.021 |
| MAPNP | 1.365 ±0.008 | 0.787 ±0.057 | 4.127 ±0.004 | 1.505 ±0.011 |
| TNP | 1.378 ±0.001 | 0.945 ±0.004 | 4.140 ±0.005 | 1.632 ±0.005 |
| DANP (ours) | **1.382** ±0.001 | **0.969** ±0.002 | **4.149** ±0.000 | 2.027 ±0.006 |

**(b) Fine-tuning on CelebA video data**

| Model | context | target |
|---|---|---|
| CANP | -1.013 ±0.116 | -1.053 ±0.076 |
| ANP | -0.498 ±0.143 | -0.517 ±0.128 |
| BANP | -0.037 ±0.334 | -0.099 ±0.273 |
| MAPNP | -1.341 ±0.132 | -1.336 ±0.136 |
| TNP | -1.574 ±0.471 | -2.747 ±0.501 |
| DANP† (ours) | 4.086 ±0.036 | 0.503 ±0.063 |
| DANP (ours) | **4.094** ±0.041 | **0.560** ±0.086 |

encoder layers remain unchanged during the fine-tuning process. The results in Table 2b clearly show that all the NP models, except for DANP, fail to achieve high generalization performance. Furthermore, the performance of DANP shows a clear improvement over the zero-shot log-likelihood result in Table 2a following the fine-tuning with the limited data. This indicates that the features from the pre-trained baselines are not effectively applied to unseen dimensional downstream datasets with a limited amount of data. In contrast, DANP is able to generalize well on these unseen dimensional downstream datasets with only a small amount of downstream data. Refer to Appendix D.2.4 to see the results on additional fine-tuning scenarios.

## 5.2 IMAGE COMPLETION AND VIDEO COMPLETION

**Image Completion** To validate our model's capability to meta-train implicit stochastic processes for varying output dimensions, we perform image completion tasks on two different datasets: EMNIST (Cohen et al., 2017) and CelebA (Liu et al., 2015). In these tasks, we randomly select some pixels as context points and use NP models to predict the selected target pixels. Here, we use the 2-dimensional position value as input and the channel value as output. Previous NP models were unable to formulate the predictive distribution for varying output dimensions, failing to learn image completion tasks with different numbers of channels. However, our DANP model can handle varying output dimensions, allowing it to simultaneously learn various image completion tasks with different numbers of channels. The experimental results for EMNIST and CelebA reported in Table 3 were measured using models trained separately for each dataset for the baselines, whereas ours were obtained using a single model trained simultaneously for both datasets. Table 3 demonstrates that DANP successfully learns both image completion tasks, validating its ability to formulate the predictive density for outputs with varying dimensions. Refer to Fig. 3 and Appendix D for the visualizations of predicted mean and standard deviation of completed images.

**Fine-tuning on Video Completion** To further validate the capability of utilizing pre-trained features in DANP for tasks with unseen dimensions, we created a simple video dataset based on the CelebA dataset. Specifically, we used the original CelebA data as the first frame at time $t = 0$ and gradually decreased the brightness by subtracting 5 from each channel for each time $t \in [9]$. This process resulted in an input dimension of 3, combining the time axis with position values. We fine-tuned pre-trained NP models on only 5 video data, simulating a scenario with limited data for the downstream task. For the baselines, we used models pre-trained on the CelebA dataset. For DANP, we used

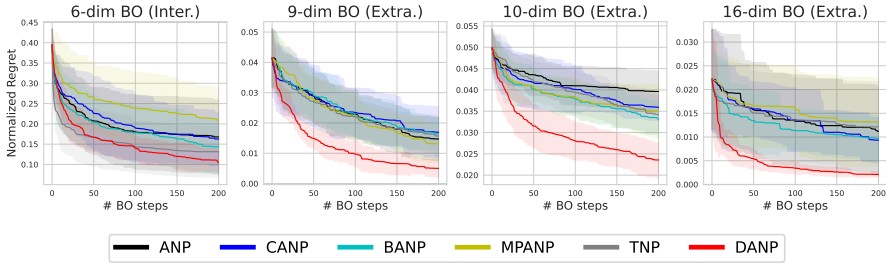

**Figure 4:** Results for Bayesian Optimization (BO) on 6-, 9-, 10-, and 16-dimensional hyperparameter tuning tasks in the HPO-B benchmark. Note that DANP is pre-trained on 2-, 3-, 8-dimensional tasks. **Table 4:** Ablation results for 1D, 2D GP regression and image completion tasks. In this table, we report the log-likelihood results for only the target dataset, excluding the context dataset.

| Model | 1D RBF | 1D Matern | 2D RBF | 2D Matern | EMNIST | CelebA |
|---|---|---|---|---|---|---|
| TNP | 0.904 ± 0.003 | 0.710 ± 0.001 | 0.362 ± 0.001 | 0.060 ± 0.002 | 0.945 ± 0.004 | 1.632 ± 0.005 |
| + DAB | 0.907 ± 0.001 | 0.713 ± 0.001 | 0.365 ± 0.001 | 0.061 ± 0.000 | 0.949 ± 0.004 | 1.645 ± 0.014 |
| + Latent | **0.923** ± 0.003 | 0.722 ± 0.001 | 0.371 ± 0.001 | 0.064 ± 0.001 | 0.967 ± 0.010 | 1.973 ± 0.023 |
| DANP | 0.921 ± 0.003 | 0.723 ± 0.003 | **0.373** ± 0.001 | **0.068** ± 0.001 | 0.969 ± 0.002 | **2.027** ± 0.006 |
| - Pos | 0.922 ± 0.002 | **0.724** ± 0.001 | -0.395 ± 0.022 | -0.446 ± 0.006 | 0.376 ± 0.012 | 0.631 ± 0.030 |
| + PMA | 0.921 ± 0.001 | 0.721 ± 0.001 | 0.372 ± 0.004 | 0.067 ± 0.002 | **0.975** ± 0.007 | 2.025 ± 0.007 |

models pre-trained on both the EMNIST and CelebA datasets. Table 3b demonstrates that, while other baseline methods fail to generalize to the increased input dimensional dataset, our method successfully learns and generalizes well with a scarce amount of training data. Refer to Appendix D for the example of video data and the predicted mean and standard deviation.

## 5.3 BAYESIAN OPTIMIZATION FOR HYPERPARAMETER TUNING

To illustrate the real-world applicability of DANP, we conducted BO (Brochu et al., 2010) experiments on 6, 9, 10, and 16-dimensional hyperparameter tuning tasks in HPO-B (Pineda-Arango et al., 2021) benchmark. HPO-B is a large-scale black-box hyperparameter optimization benchmark, which is assembled and preprocessed from the OpenML repository with 8 different hyperparameter dimensions (2, 3, 6, 8, 9, 10, 16, and 18-dim) and evaluated sparsely on 196 datasets with a total of 6.4 million hyperparameter evaluations. We target 6, 9, 10, and 16-dimensional hyperparameter tuning tasks. To do so, we pre-trained baselines on the 2-dimensional tasks in the meta-train split of the HPO-B benchmark and then fine-tune the baselines on a limited set of 4 meta-batches sampled from each target task. In contrast, for DANP, we follow the zero-shot setting, where we use a single model pre-trained on 2, 3, and 8-dimensional tasks in the meta-train split without fine-tuning on target tasks. Please see more detailed setups in Appendix C.7. We use Expected Improvement (Jones et al., 1998) as an acquisition function for all experiments and measured performance using *normalized regret*, $\frac{y_{max} - y}{y_{max} - y_{min}}$, where $y_{max}$ and $y_{min}$ denotes the global best and worst value, respectively. We run 200 iterations for all the BO experiments and report the average and standard deviation of normalized regrets over 10 different random seeds. The results in Fig. 4 demonstrate that DANP outperforms other baselines in terms of regret with the same iteration numbers. This demonstrates that DANP is capable of serving as a surrogate model for different BO-based hyperparameter tuning tasks using only a *single model* without additional training on new BO tasks, and it also effectively learns generalized shared features across a wide range of tasks. Surprisingly, the gap between DANP and baselines is even larger for the dimension extrapolation settings (9, 10, 16-dim), which empirically validates that the DANP is capable of handling unseen, varying-dimensional data, including cases where extrapolation is required. Refer to Appendix D.10 to see the results on synthetic BO tasks. The trends are similar.

## 5.4 ABLATION STUDY

To analyze the roles of each module in DANP through ablation experiments, we conducted a series of ablation experiments on various modules. The ablation experiments were categorized into three main parts: 1) the roles of the DAB module and Latent path, 2) the role of positional encoding in the DAB module, and 3) experiments replacing mean pooling with attention-based averaging in the

DAB module. The experiments were conducted on 1D GP regression, 2D GP regression, and image completion tasks. In Table 4, we analyze the log-likelihood results only for the target, excluding the context. For the full results, including the context, please refer to Appendix D.3. It can be observed that the context exhibits similar trends to the target.

**The Role of the DAB Module and Latent Path**    As mentioned in Section 3.2 and Section 3.3, the DAB module is used as a module for handling varying dimensional input, and the latent path is added to capture more accurate model uncertainty, thereby improving model performance and increasing the capacity to learn generally shared features among tasks. Table 4 show that the performance trends align well with the direction we described and intended in the Section 3. In Table 4, the DAB and Latent rows show the performance when DAB and Latent paths are added to TNP, respectively. We can observe that in all data and experiments, adding only the DAB to TNP results in a performance close to TNP, while adding the latent path results in a performance closer to DANP. This demonstrates that adding only the DAB module allows for the handling of varying dimensional data, but there are limitations in improving model performance. However, adding the latent path improves model performance but still has the issue of not being able to handle varying dimensional data.

**The Role of Positional Encoding in the DAB Module**    When treating the diverse dimensional tasks, permuting the orders of the features should not affect the result, but note that the permutation should apply simultaneously for all inputs. For instance, for a model having three features, say we permute the features to (3,1,2) for the first input and (1,3,2) for the second input. Then there is no way for the model to distinguish different feature values. Removing positional embeddings from the DAB is effectively doing this; since we treat all the features as elements in a set, it allows different permutations applied for different inputs, so the model does not distinguish different features.

We've tested the necessity of positional encoding through additional experiments, which confirmed its importance. In Table 4, "Pos" indicates the case when we extract the positional encoding from the DAB module. For the 1D GP regression tasks, because there is only one dimension for the input $x$, the existence of positional encoding does not affect the final performance. However, as seen in 2D regression tasks and image completion tasks, the existence of positional encoding is crucial for the final performance. Refer to Appendices D.5 to D.7 to see additional ablation results using Rotary Position Embedding (RoPE; Touvron et al., 2023).

**Attention-based averaging**    To verify if the mean pooling in the DAB module can be enhanced by using attention-based averaging, we employed the Pooling by Multihead Attention (PMA; Lee et al., 2019) module. This module uses a learnable token that is updated through cross-attention layers by pooling the input tokens using attention. In Table 4, the PMA row shows the results when mean pooling in the DAB is replaced with the PMA module. The results consistently indicate that mean pooling and attention-based averaging yield similar performance across nearly all tasks. Refer to Appendix D to see extensive additional empirical analysis and additional experiments.

## 6    CONCLUSION

In this paper, we present a novel NP variant that addresses the limitations of previous NP variants by incorporating a DAB block and a Transformer-based latent path. Our approach offers two key advantages: 1) the ability to directly handle diverse input and output dimensions, and 2) the capacity for learned features to be fine-tuned on new tasks with varying input and output dimensions. We empirically validate DANP across various tasks and scenarios, consistently demonstrating superior performance compared to the baselines. Conducting various experiments only with a single model can be a starting point for the utilization of NP models as general regressors for a wide range of regression tasks.

**Limitation and Future work**    In this study, DANP concentrated on the regression task, but it can naturally be extended to other tasks, such as classification. A promising direction for future work would be to pre-train the encoder, which includes the DAB module to handle diverse dimensional data, using various datasets from different tasks and then fine-tuning with a small amount of downstream data for various tasks using appropriate decoders.

**Reproducibility Statement.** We provide the architecture of our proposed model along with the architectures of other baseline models in Appendix C.1. Additionally, the hyperparameters used in the experiments, metrics, and detailed information about the data utilized in each experiment are described in Appendix C. Furthermore, additional experimental results, including ablation experiments and additional visualizations, are presented in Appendix D.

**Ethics Statement.** Our research introduces new NP variants that calculate predictive density for context and target points in tasks with varying inputs. It is unlikely that our work will have any positive or negative societal impacts. Also, we utilize openly accessible standard evaluation metrics and datasets. Furthermore, we refrain from publishing any novel datasets or models that may pose a potential risk of misuse.

**Acknowledgements** This work was partly supported by Institute of Information & communications Technology Planning & Evaluation(IITP) grant funded by the Korea government(MSIT) (No.RS-2019-II190075, Artificial Intelligence Graduate School Program(KAIST)), Institute of Information & communications Technology Planning & Evaluation(IITP) grant funded by the Korea government(MSIT) (No.RS-2024-00509279, Global AI Frontier Lab), and Institute of Information & communications Technology Planning & Evaluation(IITP) grant funded by the Korea government(MSIT) (No.RS-2022-II220713, Meta-learning Applicable to Real-world Problems).

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

## A ADDITIONAL DISCUSSION FOR THE FUTURE WORK

As demonstrated in our experiments with the MIMIC-III dataset in Appendix D.12, the ultimate goal in the Neural Processes field should be developing a general foundation regressor model capable of handling a wide range of data structures and realistic scenarios, such as diverse time series data and cases with missing features. We view the DANP research as the initial step toward achieving this ambitious objective.

A key focus of this future work direction will be to extend the model's ability to appropriately process inputs with varying dimensions, numbers of context and target points, and diverse data structures (for example there can be lots of different tasks with the same dimensional inputs such as EMNIST image completion and 2d GP regression task). Developing a model that can flexibly adapt to such variability without specific data processing based on inductive bias while providing accurate and reliable inferences across these scenarios remains a critical challenge and an exciting direction for further exploration.

## B ADDITIONAL RELATED WORKS

The first Neural Process (NP) model, the Conditional Neural Process (CNP) (Garnelo et al., 2018a), utilized straightforward MLP layers for both the encoder and decoder. Neural Process (NP) (Garnelo et al., 2018b) extended this by incorporating a global latent variable to capture model uncertainty. Enhancements followed with Attentive Neural Processes (ANP) (Kim et al., 2018) and Conditional Attentive Neural Processes (CANP), which introduced attention mechanisms in the encoder for better context summarization. Transformer Neural Processes (TNP) (Nguyen & Grover, 2022) replaced MLPs with masked transformer layers, achieving state-of-the-art performance.

Further advancements include Functional Neural Processes (Louizos et al., 2019), which employed local latent variables to improve uncertainty capture, and Bootstrapping Attentive Neural Processes (BANP) (Lee et al., 2020), which utilized a residual bootstrap approach to address model misspecification. Martingale Posterior Attentive Neural Processes (MPANP) (Lee et al., 2023) addressed uncertainty with the martingale posterior, offering a Bayesian alternative.

Recent developments have expanded NPs' scalability and expressiveness. For example, Translation Equivariant Transformer Neural Processes (TE-TNP) (Ashman et al., 2024) enhance spatio-temporal modeling with translation equivariance, which leverages symmetries in posterior predictive maps common in stationary data. Latent Bottlenecked Attentive Neural Processes (LBANP) (Feng et al., 2022a) and Mixture of Expert Neural Processes (MoE-NPs) (Wang & Van Hoof, 2022) improve latent variable modeling through bottlenecks and dynamic mixtures, improving computational efficiency and generalization across various meta-learning tasks. Autoregressive Conditional Neural Processes (AR CNPs) (Bruinsma et al., 2023) address temporal dependencies by autoregressively defining a joint predictive distribution, while Self-normalized Importance weighted Neural Process (SI-NP) (Wang et al., 2023) refine inference through iterative optimization.

Other contributions include Constant Memory Attentive Neural Processes (CMANPs) (Feng et al., 2023), which reduce memory usage with constant memory attention blocks, and Gaussian Neural Processes (GNPs) (Markou et al., 2022), focusing on tractable dependent predictions by modeling the covariance matrix of a Gaussian predictive process. Efficient Queries Transformer Neural Processes (EQTNPs) (Feng et al., 2022b) improve TNPs by applying self-attention only to the context points, retrieving information for target points through cross-attention. Together, these advancements address key limitations in uncertainty modeling, inference, and computational efficiency, forming the basis for further progress in NP research.

## C EXPERIMENTAL DETAILS

To ensure reproducibility, we have included our experiment code in the supplementary material. Our code builds upon the official implementation[1] of TNP (Nguyen & Grover, 2022). We utilize PyTorch (Ansel et al., 2024) for all experiments, and BayesO (Kim & Choi, 2023), BoTorch (Balandat

---

[1]https://github.com/tung-nd/TNP-pytorch.git

et al., 2020), and GPyTorch (Gardner et al., 2018) packages for Bayesian Optimization experiments. All experiments were conducted on either a single NVIDIA GeForce RTX 3090 GPU or an NVIDIA RTX A6000 GPU. For optimization, we used the Adam optimizer (Kingma & Ba, 2015) with cosine learning rate scheduler. Unless otherwise specified, we selected the base learning rate, weight decay, and batch size from the following grids: $\{5\times10^{-5}, 7\times10^{-5}, 9\times10^{-5}, 1\times10^{-4}, 3\times10^{-4}, 5\times10^{-4}\}$ for learning rate, $\{0, 1\times10^{-5}\}$ for weight decay, and $\{16, 32\}$ for batch size, based on validation task log-likelihood.

## C.1 DETAILS OF MODEL STRUCTURES

**Table 5:** Model structure details of CANP

| CATEGORY | DETAILS |
|---|---|
| **MODEL SPECIFICATIONS** | |
| DETERMINISTIC PATH HIDDEN DIMENSION | 128 |
| MLP DEPTH FOR VALUE IN CROSS ATTENTION LAYER | 4 |
| MLP DEPTH FOR KEY AND QUERY IN CROSS ATTENTION LAYER | 2 |
| MLP DEPTH FOR SELF-ATTENTION INPUT LAYER | 4 |
| MLP DEPTH FOR SELF-ATTENTION OUTPUT LAYER | 2 |
| DECODER DEPTH | 3 |
| NUMBER OF PARAMETERS FOR 1D GP REGRESSION | 331906 |

**Table 6:** Model structure details of ANP

| CATEGORY | DETAILS |
|---|---|
| **MODEL SPECIFICATIONS** | |
| DETERMINISTIC PATH HIDDEN DIMENSION | 128 |
| LATENT PATH HIDDEN DIMENSION | 128 |
| MLP DEPTH FOR VALUE IN CROSS ATTENTION LAYER | 4 |
| MLP DEPTH FOR KEY AND QUERY IN CROSS ATTENTION LAYER | 2 |
| MLP DEPTH FOR SELF-ATTENTION INPUT LAYER | 4 |
| MLP DEPTH FOR SELF-ATTENTION OUTPUT LAYER | 2 |
| DECODER DEPTH | 3 |
| NUMBER OF PARAMETERS FOR 1D GP REGRESSION | 348418 |

In this section, we summarize the structural details of CANP, ANP, BANP, MPANP, TNP, and DANP. It is important to note that while we report the number of parameters for the baseline models in a 1-dimensional GP regression scenario, their parameter counts increase as the input and output dimensions increase. In contrast, the number of parameters in our model remains constant regardless of the input and output dimension combinations. This proves that our model is structurally efficient compared to other baseline models. Also, following Lee et al. (2023), we model MPANP without the Self-Attention layer in the deterministic path.

**Table 7:** Model structure details of BANP

| CATEGORY | DETAILS |
|---|---|
| **MODEL SPECIFICATIONS** | |
| DETERMINISTIC PATH HIDDEN DIMENSION | 128 |
| MLP DEPTH FOR VALUE IN CROSS ATTENTION LAYER | 4 |
| MLP DEPTH FOR KEY AND QUERY IN CROSS ATTENTION LAYER | 2 |
| MLP DEPTH FOR SELF-ATTENTION INPUT LAYER | 4 |
| MLP DEPTH FOR SELF-ATTENTION OUTPUT LAYER | 2 |
| DECODER DEPTH | 3 |
| NUMBER OF PARAMETERS FOR 1D GP REGRESSION | 364674 |

**Table 8:** Model structure details of MPANP

| CATEGORY | DETAILS |
|---|---|
| **MODEL SPECIFICATIONS** | |
| DETERMINISTIC PATH HIDDEN DIMENSION | 128 |
| HIDDEN DIMENSION FOR EXCHANGEABLE GENERATIVE MODEL | 128 |
| DEPTH FOR EXCHANGEABLE GENERATIVE MODEL | 1 |
| MLP DEPTH FOR VALUE IN CROSS ATTENTION LAYER | 4 |
| MLP DEPTH FOR KEY AND QUERY IN CROSS ATTENTION LAYER | 2 |
| MLP DEPTH FOR SELF-ATTENTION INPUT LAYER | 4 |
| MLP DEPTH FOR SELF-ATTENTION OUTPUT LAYER | 2 |
| DECODER DEPTH | 3 |
| NUMBER OF PARAMETERS FOR 1D GP REGRESSION | 892418 |

## C.2 EVALUATION METRIC FOR THE TASKS

Following Le et al. (2018), we used the normalized predictive log-likelihood:

$$\frac{1}{|o|} \sum_{k \in o} \log p(\mathbf{y}_{j,k} | \mathbf{x}_{j,k}, \mathcal{D}_{j,c}) \tag{20}$$

for the CNP variants CANP and TNP, where $o \in \{c_j, t_j\}$ denotes context or target points. For the other models, we approximated the normalized predictive log-likelihood as follows:

$$\frac{1}{|o|} \sum_{k \in o} \log p(y_{j,k} | x_{j,k}, \mathcal{D}_{j,c}) \approx \frac{1}{|o|} \sum_{k \in o} \log \frac{1}{K} \sum_{k=1}^{K} p(y_{j,k} | x_{j,k}, \theta_j^{(k)}), \tag{21}$$

where $\theta_j^{(k)}$ are independent samples drawn from $q(\theta_j | \mathcal{D}_{j,c})$ for $k \in [K]$. Again, $o \in \{c_j, t_j\}$ indicates context or target points.

**Table 9:** Model structure details of TNP

| CATEGORY | DETAILS |
|---|---|
| **MODEL SPECIFICATIONS** | |
| HIDDEN DIMENSION FOR EMBEDDING LAYERS | 64 |
| NUMBER OF LAYERS FOR EMBEDDING LAYERS | 4 |
| HIDDEN DIMENSION FOR MASKED TRANSFORMER LAYERS | 128 |
| NUMBER OF LAYERS FOR MASKED TRANSFORMER LAYERS | 6 |
| NUMBER OF HEADS FOR MASKED TRANSFORMER LAYERS | 4 |
| DECODER DEPTH | 2 |
| NUMBER OF PARAMETERS FOR 1D GP REGRESSION | 222082 |

**Table 10:** Model structure details of DANP

| CATEGORY | DETAILS |
|---|---|
| **MODEL SPECIFICATIONS** | |
| HIDDEN DIMENSION FOR LINEAR PROJECTION IN DAB | 32 |
| HIDDEN DIMENSION FOR SELF-ATTENTION IN DAB | 32 |
| HIDDEN DIMENSION FOR TRANSFORMER LAYERS IN LATENT PATH | 64 |
| NUMBER OF LAYERS FOR TRANSFORMER LAYERS IN LATENT PATH | 2 |
| HIDDEN DIMENSION FOR SELF-ATTENTION IN LATENT PATH | 64 |
| HIDDEN DIMENSION FOR MLP LAYERS IN LATENT PATH | 128 |
| NUMBER OF LAYERS FOR MLP LAYERS IN LATENT PATH | 2 |
| HIDDEN DIMENSION FOR MASKED TRANSFORMER LAYERS | 128 |
| NUMBER OF LAYERS FOR MASKED TRANSFORMER LAYERS | 6 |
| NUMBER OF HEADS FOR MASKED TRANSFORMER LAYERS | 4 |
| DECODER DEPTH | 2 |
| NUMBER OF PARAMETERS FOR 1D GP REGRESSION | 334562 |

### C.3  DATASET DETAILS OF n-DIMENSIONAL GP REGRESSION TASK

In an $n$-dimensional Gaussian Process (GP) regression task, we start by sampling the context and target inputs. Specifically, we first determine the number of context points $|c|$ by drawing from a uniform distribution $\text{Unif}(n^2 \times 5, n^2 \times 50 - n^2 \times 5)$. The interval for this uniform distribution is scaled by $n^2$ to ensure that as the input dimension increases, the number of context points also increases, which is necessary for constructing an accurate predictive density for the target points.

Next, we sample the number of target points $|t|$ from a uniform distribution $\text{Unif}(n^2 \times 5, n^2 \times 50 - |c|)$ to keep the total number of points within a manageable range. After determining the number of

context and target points, we sample the input $\mathbf{x}$ for each context and target point from the uniform distribution $\text{Unif}(-2, 2)$ independently for each dimension $i$ in $[n]$.

We then generate the outputs $\mathbf{y}$ using the corresponding kernel functions. We employ the RBF kernel $k(\mathbf{x}, \mathbf{x}') = s^2 \exp\left(\frac{-||\mathbf{x}-\mathbf{x}'||^2}{2\ell^2}\right)$ and the Matern $5/2$ kernel $k(\mathbf{x}, \mathbf{x}') = s^2 \left(1 + \frac{\sqrt{5}d}{\ell} + \frac{5d^2}{3\ell^2}\right)$, where $d = ||\mathbf{x} - \mathbf{x}'||$. For these kernels, the parameters are sampled as follows: $s \sim \text{Unif}(0.1, 1.0)$, $\ell \sim \text{Unif}(0.1, 0.6)$, and $p \sim \text{Unif}(0.1, 0.5)$.

### C.4 EMNIST DATASET

We employed the EMNIST [2] Balanced dataset (Cohen et al., 2017), which consists of 112,800 training samples and 18,800 test samples. This dataset encompasses 47 distinct classes, from which we selected 11 classes for our use. Consequently, our training and test datasets comprise 26,400 and 4,400 samples, respectively. Each image is represented by a $28 \times 28$ grid with a single channel. We mapped the pixel coordinates to a range from -0.5 to 0.5 and normalized the pixel values to lie within [-0.5, 0.5]. We sample the number of context points $|c| \sim \text{Unif}(5, 45)$ and the number of target points $|t| \sim \text{Unif}(5, 50 - |c|)$.

### C.5 CELEBA DATASET

We utilized the CelebA [3] dataset (Liu et al., 2015), which includes 162,770 training samples, 19,867 validation samples, and 19,962 test samples. The images were center-cropped to 32x32 pixels, resulting in a $32 \times 32$ grid with 3 RGB channels. As with the EMNIST dataset, we scaled the pixel coordinates to a range of -0.5 to 0.5 and normalized each pixel value within [-0.5, 0.5]. We sampled the number of context points $|c|$ from $\text{Unif}(5, 45)$ and the number of target points $|t|$ from $\text{Unif}(5, 50 - |c|)$.

### C.6 CELEBA VIDEO DATASET

For the CelebA Video dataset, we generated a simple video dataset using the CelebA image dataset. Specifically, after normalizing the pixel coordinates and values according to the pre-processing steps for the CelebA dataset, we set the original CelebA data as the initial frame at time $t = 0$. We then gradually decreased the brightness by subtracting $5/255$ from each channel for each time step $t \in [9]$, concatenating each generated image to the previous ones. This resulted in a simple video with a $32 \times 32$ grid, 3 RGB channels, and 10 frames. Consequently, the input dimension was 3, combining the time axis with pixel coordinates. As with the CelebA dataset, we sampled the number of context points $|c|$ from $\text{Unif}(5, 45)$ and the number of target points $|t|$ from $\text{Unif}(5, 50 - |c|)$.

### C.7 BAYESIAN OPTIMIZATION

Except for the BO experiments on HPO-B benchmark, we adjust the objective function to have the domain of $[-2.0, 2.0]$. We evaluated our method using various benchmark datasets and real-world scenarios. Below, we provide details of these experiments.

**Hyperparameter Tuning on HPO-B benchmark** We utility the HPO-B benchmark (Pineda-Arango et al., 2021), which consists of 176 search spaces (algorithms) evaluated sparsely on 196 datasets with a total of 6.4 million hyperparameter evaluations. In this benchmark, the continuous hyperparameters normalized in $[0, 1]$, and the categorical hyperparameters are one-hot encoded. We use all the search space except for the 18-dimensional space, i.e., 2-, 3-, 6-, 8-, 9-, 10-, and 16-dimensional search spaces are used for the experiments. To construct meta-batch from each task, we sample the number of context points $|c|$ from $\text{Unif}(5, 50)$ and the number of target points $|t|$ from $\text{Unif}(5, 50 - |c|)$; therefore, we exclude tasks which lesser than $100\,(= 50 + 50)$ hyperparameter evaluations. For baselines, we first pre-train them on all the tasks collected from all the 2-dimensional search spaces of meta-train split. We then fine-tune them on 4 meta-batches randomly sampled from each target task (6-, 9-, 10-, 16-dim). To prevent an overfitting on the limited data, we early-stop the

---

[2] https://www.nist.gov/itl/products-and-services/emnist-dataset
[3] https://mmlab.ie.cuhk.edu.hk/projects/CelebA.html

training with respect to the likelihood on meta-validation split of target task. For DANP, we pre-train it on all the tasks collected from 2-, 3-, and 8-dimensional search spaces of meta-train split. We then evaluate it without fine-tuning on target tasks, which corresponds to zero-shot setting. For BO, we run 200 iterations for each hyperparameter tuning task in meta-test split; therefore, we exclude tasks which have lesser than 210 hyperparameter evaluations. Furthermore, the order of hyperparameter dimensions is randomly shuffled to make the task diverse.

**1 dimensional BO with GP generated objective functions** To evaluate basic BO performance when using each model as a surrogate for the black-box objective function, we first create an oracle sample using a GP with an RBF kernel and evaluate how well each model approximates these samples. We conducted 1D BO for 100 iterations across 100 tasks using the expected improvement acquisition function.

**2 and 3 dimensional BO benchmarks** We utilize three benchmark objective functions:

**Ackley (Back, 1996) function**

$$f(x) = -a \exp\left(-b\sqrt{\frac{1}{d}\sum_{i=1}^{d} x_i^2}\right) - \exp\left(\frac{1}{d}\sum_{i=1}^{d} \cos(cx_i)\right) + a + \exp(1) \tag{22}$$

where $x_i \in [-32.768, 32.768]$, for all $i = 1, \cdots, d$ and global minimum is $x^* \approx (0, \cdots, 0)$.

**Cosine function**

$$f(x) = \sum_{i=1}^{d} \cos(x_i)\left(\frac{0.1}{2\pi}|x_i| - 1\right) \tag{23}$$

where $x_i \in [-2\pi, 2\pi]$, for all $i = 1, \cdots, d$ and global minimum is $x^* \approx (0, \cdots, 0)$.

**Rastrigin (Rastrigin, 1974) function**

$$f(x) = 10d + \sum_{i=1}^{d}[x_i^2 - 10\cos(2\pi x_i)] \tag{24}$$

where $x_i \in [-5.12, 5.12]$, for all $i = 1, \cdots, d$ and global minimum is $x^* \approx (0, \cdots, 0)$.

To evaluate the models in multi-dimensional scenarios, we conduct experiments with cases of $d = 1$ and $d = 2$ for all the aforementioned benchmark functions. We perform evaluations over 100 iterations for each of the 100 tasks, utilizing the expected improvement acquisition function.

**CNN BO** For evaluation in a real-world BO scenario, we utilized the CIFAR-10 dataset (Krizhevsky et al., 2009) and a Convolutional Neural Network (CNN) (LeCun et al., 1989). The CIFAR-10 dataset consists of 50,000 training samples and 10,000 test samples across 10 classes. In this setting, we generated 1,000 samples by creating combinations of weight decay, learning rate, and batch size, and trained the model for 20 epochs using the Adam optimizer (Kingma & Ba, 2015). The range for each hyperparameter is $[1e - 05, 1e - 01]$ for learning rate, $[1e - 04, 1e - 01]$ for weight decay, and $[128, 256]$ for batch size, with 10 values selected uniformly within each range. These 1,000 samples were pre-generated, and we evaluated each BO task with 1 initial sample and conducted 50 iterations for each of the 10 tasks using the expected improvement acquisition function.

## D  ADDITIONAL EXPERIMENTS

### D.1  COMPARISON BETWEEN NEURAL DIFFUSION PROCESS AND DIMENSION AGNOSTIC NEURAL PROCESSES

As we discussed in Section 4, NDP has two major issues: 1) it has a structural limitation, being only partially dimension-agnostic for $x$ when $y = 1$, and not dimension-agnostic for other combinations, and 2) its reliance on diffusion-based sampling to approximate the predictive distribution results in

**Table 11:** Additional results of the context and target log-likelihood for the GP regression task in the From-scratch scenario on NDP and DANP.

| Model | 1D RBF | | 1D Matern | | 2D RBF | | 2D Matern | |
|---|---|---|---|---|---|---|---|---|
| | context | target | context | target | context | target | context | target |
| NDP | **5.914** ±0.097 | -0.376 ±0.077 | **5.924** ±0.046 | -0.503 ±0.016 | **5.945** ±0.044 | -0.570 ±0.006 | **6.079** ±0.114 | -0.704 ±0.021 |
| DANP (ours) | 1.381 ±0.000 | **0.921** ±0.003 | 1.382 ±0.000 | **0.723** ±0.003 | 1.383 ±0.000 | **0.373** ±0.001 | 1.383 ±0.000 | **0.068** ±0.001 |

**Table 12:** Additional results of zero-shot scenario. The colored cell ■ indicates the data dimension used to pre-train DANP and NDP.

| Dimension | NDP trained on 2D & 3D & 4D | | DANP trained on 2D & 3D & 4D | |
|---|---|---|---|---|
| | context | target | context | target |
| 1D RBF | 5.5664 ±0.001 | -0.5665 ±0.097 | 1.366 ±0.004 | 0.826 ±0.018 |
| 2D RBF | 5.9409 ±0.002 | -1.5654 ±0.092 | 1.383 ±0.000 | 0.335 ±0.014 |
| 3D RBF | 5.5935 ±0.001 | -4.5919 ±0.098 | 1.383 ±0.000 | -0.261 ±0.025 |
| 4D RBF | 5.9792 ±0.005 | -7.8666 ±0.095 | 1.383 ±0.000 | -0.568 ±0.042 |
| 5D RBF | 5.3512 ±0.008 | -8.4127 ±0.103 | 1.359 ±0.032 | -0.676 ±0.004 |
| 7D RBF | 5.4938 ±0.009 | -14.6106 ±0.101 | 1.355 ±0.022 | -0.723 ±0.022 |

significantly high computational costs during inference and limited likelihood performance. Table 11 and Table 12 clearly show that while NDP outperforms DANP in terms of context likelihood, it significantly underperforms in target likelihood. This discrepancy arises because NDP relies on a diffusion-based sampling method to generate possible outputs and then calculates the empirical posterior distribution from the gathered samples. This approach leads the model to predict the context points with high accuracy and low variance, thus achieving high context likelihood. However, for target points, the model struggles to accurately predict the distribution, resulting in a lower target likelihood. Moreover, in most of our tasks, it is more important to achieve high likelihood predictions for unseen target points rather than focusing on the observed context points. Therefore, having a higher target point likelihood is more crucial than having a high context likelihood. Specifically, as shown in Table 12, NDP struggles to simultaneously learn across diverse dimensional inputs, demonstrating that it cannot function effectively as a general regressor for unseen dimensions.

## D.2 GP REGRESSION TASK

### D.2.1 OTHER METRICS

**Table 13:** Additional evaluation of CRPS metric and confidence interval coverage on the 1D GP regression task with RBF kernel. For the CRPS metric, a smaller value indicates better performance.

| Model | context CI | target CI | context CRPS ($\downarrow$) | target CRPS ($\downarrow$) |
|---|---|---|---|---|
| ANP | 0.999 ± 0.000 | 0.889 ± 0.014 | **0.024** ± 0.000 | 0.075 ± 0.004 |
| BANP | 0.999 ± 0.000 | 0.904 ± 0.001 | **0.024** ± 0.000 | 0.075 ± 0.000 |
| CANP | 0.999 ± 0.000 | 0.899 ± 0.001 | **0.024** ± 0.000 | 0.076 ± 0.000 |
| MPANP | 0.999 ± 0.000 | 0.898 ± 0.001 | **0.024** ± 0.000 | 0.075 ± 0.000 |
| TNP | 0.999 ± 0.000 | 0.909 ± 0.001 | **0.024** ± 0.001 | 0.071 ± 0.000 |
| DANP | 0.999 ± 0.000 | 0.912 ± 0.002 | **0.024** ± 0.000 | **0.068** ± 0.000 |

**CRPS and Empirical Confidence interval coverage**   Here, we further evaluate DANP and other baselines using additional metrics like continuous ranked probability score (CRPS) and empirical confidence interval coverage. We have measured these metrics for the 1D and 2D GP regression tasks. For the fair comparison, we used the checkpoints from the **From-scratch** experiment in Section 5.1 for all models. The results are presented in Table 13.

First, regarding the confidence interval coverage, it is observed that the context confidence interval tends to be too wide for all models. This issue arises because Neural Process models set a minimum standard deviation of 0.1 during inference to account for training stability and data noise. Additionally, the value of 0.1 is simply a conventional choice when designing Neural Process models. Therefore, setting this value lower during model construction can help ensure an appropriate confidence interval. On the other hand, the target confidence interval tends to be relatively narrow, with the DANP model showing the best results. Additionally, when looking at CRPS scores, it is clear that for context points, the models generally perform at a similar level, but for target points, DANP shows better scores compared to the other models.

**Table 14:** Results on additional metrics containing MAE, RMSE, $R^2$, RMSCE, MACE, and MA on 1d GP regression task. Except for $R^2$, lower values for all these metrics indicate better alignment with the target and improved calibration performance.

| Model | MAE ($\downarrow$) | RMSE ($\downarrow$) | $R^2$ ($\uparrow$) | RMSCE ($\downarrow$) | MACE ($\downarrow$) | MA ($\uparrow$) |
|---|---|---|---|---|---|---|
| ANP | $0.126 \pm 0.001$ | $0.176 \pm 0.003$ | $0.788 \pm 0.012$ | $0.273 \pm 0.001$ | $0.238 \pm 0.003$ | $0.240 \pm 0.003$ |
| BANP | $0.125 \pm 0.001$ | $0.175 \pm 0.003$ | $\underline{0.811} \pm 0.001$ | $0.273 \pm 0.007$ | $\underline{0.237} \pm 0.001$ | $0.239 \pm 0.002$ |
| CANP | $0.127 \pm 0.001$ | $0.178 \pm 0.002$ | $0.801 \pm 0.005$ | $0.267 \pm 0.008$ | $0.239 \pm 0.002$ | $\underline{0.237} \pm 0.005$ |
| MPANP | $0.124 \pm 0.001$ | $\underline{0.173} \pm 0.003$ | $0.807 \pm 0.005$ | $0.274 \pm 0.014$ | $0.242 \pm 0.007$ | $0.244 \pm 0.008$ |
| TNP | $\underline{0.122} \pm 0.002$ | $\underline{0.173} \pm 0.001$ | $0.808 \pm 0.002$ | $0.287 \pm 0.003$ | $0.251 \pm 0.005$ | $0.253 \pm 0.006$ |
| DANP | $\mathbf{0.120} \pm 0.001$ | $\mathbf{0.165} \pm 0.002$ | $\mathbf{0.816} \pm 0.002$ | $\mathbf{0.259} \pm 0.000$ | $\mathbf{0.230} \pm 0.003$ | $\mathbf{0.228} \pm 0.002$ |

**Metrics related to calibration**  We additionally measure some other metrics related to calibration. We measured and reported the following 6 additional metrics: 1) Mean Absolute Error (MAE), 2) Root Mean Square Error (RMSE), 3) Coefficient of Determination ($R^2$), 4) Root Mean Square Calibration Error (RMSCE), 5) Mean Absolute Calibration Error (MACE), 6) Miscalibration Area (MA). Except for $R^2$, lower values for all these metrics indicate better alignment with the target and improved calibration performance. We conducted the evaluation using models trained on a 1d GP regression task, comparing our method with the baselines. The results, summarized in Table 14, demonstrate that DANP achieves the best performance across a range of metrics. This observation reaffirms that DANP not only outperforms in terms of NLL but also achieves improved performance in calibration-related metrics compared to the baselines. These additional evaluations highlight the robustness of our method across diverse aspects of model performance.

### D.2.2 FINE-GRAINED EVALUATION ON THE 1D GP REGRESSION TASKS

**Table 15:** Additional fine-grained evaluation on the 1D GP regression task. Here, we evaluate for the less context and more context scenarios.

| Model | Less context | | More context | |
|---|---|---|---|---|
| | context | target | context | target |
| ANP | $1.380 \pm 0.000$ | $0.323 \pm 0.006$ | $1.375 \pm 0.000$ | $1.165 \pm 0.002$ |
| BANP | $1.380 \pm 0.000$ | $0.334 \pm 0.002$ | $1.375 \pm 0.000$ | $1.172 \pm 0.001$ |
| CANP | $1.380 \pm 0.001$ | $0.291 \pm 0.005$ | $1.374 \pm 0.000$ | $1.156 \pm 0.001$ |
| MPANP | $1.380 \pm 0.000$ | $0.317 \pm 0.007$ | $1.374 \pm 0.001$ | $1.165 \pm 0.003$ |
| TNP | $\underline{1.382} \pm 0.000$ | $\underline{0.363} \pm 0.005$ | $\underline{1.379} \pm 0.001$ | $\underline{1.209} \pm 0.004$ |
| DANP | $\mathbf{1.383} \pm 0.000$ | $\mathbf{0.396} \pm 0.003$ | $\mathbf{1.380} \pm 0.000$ | $\mathbf{1.214} \pm 0.002$ |

Here, we evaluate how uncertainty behavior changes under various conditions for each method. To explore these changes, we considered three settings in GP regression tasks: 1) scenarios with a small number of context points versus a large number, 2) situations with high noise scale, and 3) cases where the training kernel differs from the evaluation kernel. The experimental results can be found in Table 15 and Table 16.

First, in the 1D GP regression experiment reported in Section 5.1, the number of context points ranged randomly from a minimum of 5 to a maximum of 45 for evaluation. For the first setting, 'small

**Table 16:** Additional fine-grained evaluation on the 1D GP regression task. Here, we evaluate the increased noise scale and kernel change scenarios.

| Model | Noise | | Kernel change | |
|---|---|---|---|---|
| | context | target | context | target |
| ANP | $1.377 \pm 0.000$ | $0.855 \pm 0.004$ | $1.373 \pm 0.000$ | $0.667 \pm 0.003$ |
| BANP | $1.377 \pm 0.000$ | $0.864 \pm 0.001$ | $1.373 \pm 0.000$ | $0.688 \pm 0.003$ |
| CANP | $1.377 \pm 0.000$ | $0.837 \pm 0.003$ | $1.372 \pm 0.000$ | $0.641 \pm 0.003$ |
| MPANP | $1.376 \pm 0.001$ | $0.856 \pm 0.005$ | $1.372 \pm 0.001$ | $0.663 \pm 0.005$ |
| TNP | $\underline{1.381} \pm 0.000$ | $\underline{0.906} \pm 0.005$ | $\underline{1.380} \pm 0.000$ | $\underline{0.697} \pm 0.003$ |
| DANP | $\underline{1.381} \pm 0.001$ | $\mathbf{0.922} \pm 0.002$ | $\mathbf{1.381} \pm 0.000$ | $\mathbf{0.717} \pm 0.008$ |

**Table 17:** Additional experimental results showing the context and target log-likelihoods for the GP regression task in **Zero-shot** scenarios. The first column labeled $n$D represents the outcomes for the $n$-dimensional GP dataset with an RBF kernel. Cells highlighted in ▮ indicate the data dimension used to pre-train DANP. The initial context and target log-likelihood results are derived from the DANP model trained on 2 and 3-dimensional GP datasets with an RBF kernel. The subsequent results are obtained from the DANP model trained on 2, 3, and 4-dimensional GP datasets with both RBF and Matern kernels.

| Dimension | 2D & 3D with RBF | | 2D & 3D & 4D with RBF and Matern | |
|---|---|---|---|---|
| | context | target | context | target |
| 1D RBF | $1.354 \pm 0.023$ | $0.826 \pm 0.048$ | $1.360 \pm 0.022$ | $0.830 \pm 0.030$ |
| 2D RBF | $1.383 \pm 0.000$ | $0.341 \pm 0.008$ | $1.383 \pm 0.001$ | $0.318 \pm 0.021$ |
| 3D RBF | $1.383 \pm 0.001$ | $-0.251 \pm 0.008$ | $1.383 \pm 0.001$ | $-0.296 \pm 0.050$ |
| 4D RBF | $1.368 \pm 0.003$ | $-0.661 \pm 0.012$ | $1.383 \pm 0.001$ | $-0.601 \pm 0.063$ |
| 5D RBF | $1.293 \pm 0.050$ | $-0.711 \pm 0.012$ | $1.378 \pm 0.003$ | $-0.679 \pm 0.005$ |

context' refers to using 5 to 15 context points, while 'large context' involves 30 to 45 context points for evaluation. In the second setting, the variance of the Gaussian noise used was increased to 2.5 times the original value, and the model was evaluated using this adjusted evaluation set. Lastly, for the third setting, we evaluated the model, trained on an RBF kernel, with an evaluation set generated using a Matern 5/2 kernel.

As shown in Table 15, DANP clearly outperforms other baselines in both small and large context scenarios. Notably, DANP demonstrates superior performance compared to the other baselines in the small context scenario, indicating its ability to accurately predict the predictive distribution with a limited number of observations. Moreover, as illustrated in Table 16, DANP excels in both the increased noise scale and differing kernel scenarios. These results confirm that DANP can effectively adapt to unseen tasks by learning generally shared features across different tasks.

### D.2.3 ADDITIONAL RESULTS FOR THE ZERO-SHOT SCENARIO

In the **Zero-shot** scenario for the GP regression task, we conducted two additional experiments utilizing DANP pre-trained with: 1) 2 and 3-dimensional GP datasets using the RBF kernel, and 2) 2, 3, and 4-dimensional GP datasets using both RBF and Matern kernels. The log-likelihood results in Table 17 for the first experiment indicate that DANP can effectively predict the density of unseen dimensional GP datasets, though performance slightly declines for higher-dimensional datasets that are farther from the trained dimensions, as compared to the results in Table 2a. This demonstrates that while DANP can perform zero-shot inference on unseen dimensional datasets, training on a diverse range of dimensions enhances predictive performance.

In the second experiment, the log-likelihood results show that DANP can be trained on diverse tasks with different kernels. Notably, DANP was able to simultaneously train on 2, 3, and 4-dimensional GP

**Table 18:** Additional experimental results showing the context and target log-likelihoods for the GP regression in **Fine-tuning** scenarios. Here, we utilize 1-dimensional GP regression task with the Matern kernel as a downstream task.

| Method | Full fine-tuning | | Freeze fine-tuning | |
|---|---|---|---|---|
| | context | target | context | target |
| CANP | -0.352 ±0.053 | -0.512 ±0.034 | -0.233 ±0.108 | -0.408 ±0.084 |
| ANP | -0.280 ±0.094 | -0.348 ±0.080 | -0.343 ±0.044 | -0.422 ±0.023 |
| BANP | -0.320 ±0.073 | -0.401 ±0.037 | -0.025 ±0.145 | -0.207 ±0.154 |
| MPANP | -0.128 ±0.246 | -0.259 ±0.125 | -0.260 ±0.092 | -0.490 ±0.142 |
| TNP | -0.086 ±0.024 | -0.476 ±0.139 | 0.336 ±0.128 | -0.243 ±0.162 |
| DANP(ours) | **1.372** ±0.001 | **0.689** ±0.004 | **1.372** ±0.001 | **0.684** ±0.004 |

**Table 19:** Ablation results for 1D, 2D GP regression and image completion tasks.

| Model | 1D RBF | | 1D Matern | | 2D RBF | | 2D Matern | | EMNIST | | CelebA | |
|---|---|---|---|---|---|---|---|---|---|---|---|---|
| | context | target | context | target | context | target | context | target | context | target | context | target |
| TNP | **1.381** ± 0.000 | 0.904 ± 0.003 | 1.381 ± 0.000 | 0.710 ± 0.001 | **1.383** ± 0.000 | 0.362 ± 0.001 | **1.383** ± 0.000 | 0.060 ± 0.002 | 1.378 ± 0.001 | 0.945 ± 0.004 | 4.140 ± 0.005 | 1.632 ± 0.005 |
| + DAB | **1.381** ± 0.000 | 0.907 ± 0.001 | **1.382** ± 0.000 | 0.713 ± 0.001 | **1.383** ± 0.000 | 0.365 ± 0.001 | **1.383** ± 0.000 | 0.061 ± 0.000 | 1.378 ± 0.001 | 0.949 ± 0.004 | 4.146 ± 0.001 | 1.645 ± 0.014 |
| + Latent | **1.381** ± 0.000 | **0.923** ± 0.003 | **1.382** ± 0.000 | 0.722 ± 0.001 | **1.383** ± 0.000 | 0.371 ± 0.001 | **1.383** ± 0.000 | 0.064 ± 0.001 | 1.379 ± 0.001 | 0.967 ± 0.010 | 4.140 ± 0.002 | 1.973 ± 0.023 |
| DANP | **1.381** ± 0.000 | 0.921 ± 0.003 | **1.382** ± 0.000 | 0.723 ± 0.003 | **1.383** ± 0.000 | **0.373** ± 0.001 | **1.383** ± 0.000 | **0.068** ± 0.001 | **1.382** ± 0.001 | 0.969 ± 0.002 | 4.149 ± 0.000 | **2.027** ± 0.006 |
| - Pos | **1.381** ± 0.000 | 0.922 ± 0.002 | **1.382** ± 0.000 | **0.724** ± 0.001 | 1.381 ± 0.000 | -0.395 ± 0.022 | 1.381 ± 0.001 | -0.446 ± 0.006 | 1.279 ± 0.009 | 0.376 ± 0.012 | 3.117 ± 0.005 | 0.631 ± 0.030 |
| + PMA | **1.381** ± 0.000 | 0.921 ± 0.001 | **1.382** ± 0.000 | 0.721 ± 0.001 | **1.383** ± 0.000 | 0.372 ± 0.004 | **1.383** ± 0.000 | 0.067 ± 0.002 | 1.381 ± 0.000 | **0.975** ± 0.007 | **4.150** ± 0.001 | 2.025 ± 0.007 |

datasets with both RBF and Matern kernels without increasing model size. By increasing the model size to accommodate more features and additional structural layers, DANP can generalize to a wider variety of tasks with different generating processes.

### D.2.4 ADDITIONAL RESULTS FOR THE FINE-TUNING SCENARIO WITH DIFFERENT KERNEL

In the **Fine-tuning** scenario for the GP regression task, we fine-tuned on 160 1-dimensional GP regression tasks using the Matern kernel. For baselines, we utilized pre-trained models that had been trained on 2-dimensional GP regression tasks with the RBF kernel, as detailed in Section 5.1. For DANP, we used models pre-trained on 2, 3, and 4-dimensional GP regression tasks with the RBF kernel, also as described in Section 5.1. The results in Table 18 clearly demonstrate that while the baselines fail to generalize, DANP can generalize to the 1-dimensional GP regression task with the Matern kernel almost as effectively as the **From-scratch** results in Table 1 with only a few datasets. This indicates that DANP not only generalize well to unseen dimensional GP tasks with a known kernel but also to unseen dimensional GP tasks with an unknown kernel, compared to other baselines.

### D.3 FULL EXPERIMENTAL RESULTS FOR THE SECTION 5.4

In this subsection, we report the full log-likelihood results from the ablation study on both the context and target datasets. In Table 19, it can be easily observed that the context exhibits similar trends to the target as we discussed in Section 5.4.

### D.4 ABLATION RESULTS ON DIFFERENT OBJECTIVES

As highlighted in Foong et al. (2020), the ELBO loss we used for DANP does not provide the exact ELBO for the $-\log p_\theta(y_t|x_t, D_c)$, because we use $q(\theta|D_c)$ instead of $p(\theta|D_c)$. More precisely, the Maximum Likelihood Loss is a biased estimator of $-\log p_\theta(y_t|x_t, D_c)$, and the ELBO we used is a lower bound of the same quantity. Therefore, both losses still share the same issue, and the effectiveness of each loss depends on the model.

Typically, the maximum likelihood loss tends to exhibit larger variance compared to variational inference, so, given our model's need to handle multiple varying dimensional tasks simultaneously, we opted for variational inference to ensure stability. However, it is worth experimenting with other loss functions. Therefore, we conducted additional experiments and included the results from training with the maximum likelihood loss as well.

**Table 20:** Comparison of zero-shot performance between DANP trained with the variational loss and the maximum likelihood loss. Here, each method trained with 2 and 4D GP datasets with RBF kernel while performing inference on the 1, 2, 3, 4, and 5D GP datasets with RBF kernel

| Method | Variational Inference | | Marginal Likelihood | |
|---|---|---|---|---|
| | context | target | context | target |
| 1D RBF | $1.336 \pm 0.047$ | $\mathbf{\underline{0.806}} \pm 0.048$ | $\mathbf{\underline{1.340}} \pm 0.025$ | $0.790 \pm 0.008$ |
| 2D RBF | $\mathbf{\underline{1.383}} \pm 0.000$ | $\mathbf{\underline{0.340}} \pm 0.007$ | $\mathbf{\underline{1.383}} \pm 0.000$ | $0.330 \pm 0.012$ |
| 3D RBF | $1.377 \pm 0.007$ | $\mathbf{\underline{-0.360}} \pm 0.063$ | $\mathbf{\underline{1.381}} \pm 0.001$ | $-0.420 \pm 0.112$ |
| 4D RBF | $1.379 \pm 0.007$ | $\mathbf{\underline{-0.589}} \pm 0.056$ | $\mathbf{\underline{1.383}} \pm 0.000$ | $-0.614 \pm 0.045$ |
| 5D RBF | $\mathbf{\underline{1.357}} \pm 0.012$ | $\mathbf{\underline{-0.689}} \pm 0.004$ | $1.356 \pm 0.040$ | $-0.701 \pm 0.023$ |

**Table 21:** Comparison of zero-shot performance between DANP trained with the variational loss and the maximum likelihood loss. Here, each method trained with 2, 3, and 4D GP datasets with RBF kernel while performing inference on the 1, 2, 3, 4, and 5D GP datasets with RBF kernel

| Method | Variational Inference | | Marginal Likelihood | |
|---|---|---|---|---|
| | context | target | context | target |
| 1D RBF | $\mathbf{\underline{1.366}} \pm 0.004$ | $\mathbf{\underline{0.826}} \pm 0.018$ | $1.360 \pm 0.006$ | $0.805 \pm 0.021$ |
| 2D RBF | $\mathbf{\underline{1.383}} \pm 0.000$ | $\mathbf{\underline{0.355}} \pm 0.014$ | $1.382 \pm 0.000$ | $0.285 \pm 0.012$ |
| 3D RBF | $\mathbf{\underline{1.383}} \pm 0.000$ | $\mathbf{\underline{-0.261}} \pm 0.025$ | $\mathbf{\underline{1.383}} \pm 0.001$ | $-0.320 \pm 0.044$ |
| 4D RBF | $1.383 \pm 0.000$ | $\mathbf{\underline{-0.568}} \pm 0.042$ | $1.381 \pm 0.002$ | $-0.658 \pm 0.039$ |
| 5D RBF | $1.359 \pm 0.032$ | $\mathbf{\underline{-0.676}} \pm 0.004$ | $\mathbf{\underline{1.364}} \pm 0.021$ | $-0.742 \pm 0.006$ |

We conducted ablation experiments on the ML loss and VI loss using DANP trained on 2 and 4d GP data, as well as DANP trained on 2d, 3d, and 4d GP data. These experiments were performed in a zero-shot scenario by inferring on 1, 2, 3, 4, and 5d GP regression data. The results, presented in Table 20 and Table 21, show that while ML loss occasionally yields better log-likelihoods for context points, the VI loss consistently provides superior performance for the target points, which are of greater interest during inference. This trend is particularly evident in experiments trained on 2, 3, and 4d GP data. These findings demonstrate that using the VI loss for training DANP is generally more beneficial for improving generalization compared to the ML loss.

### D.5 ABLATION EXPERIMENTS ON POSITIONAL EMBEDDING IN DAB MODULE

Many previous works have shown that sinusoidal positional encoding tends to perform poorly (Press et al., 2021) in terms of generalization when extrapolating to longer sequence lengths for Large Language Models. In response to this, approaches like Rotary Position Embedding (RoPE; Touvron et al., 2023; Su et al., 2024) have been proposed and used to address these limitations. While sinusoidal positional encoding successfully handled interpolation and extrapolation in our experimental settings, RoPE could potentially improve this performance. Therefore, we conducted additional experiments using a modified RoPE-based encoding tailored for the DAB module.

In our implementation, we retained the basic formulation of RoPE while ensuring different positional encodings for the **each $x$ and $y$ dimensions**, similar to the approach we used with DAB. Specifically, we distinguished the embeddings added to queries and keys from $x$ and $y$ by alternating the cosine and sine multiplications for each. For example, if for x we calculate $q_{1x} \cdot \cos(\text{pos}) + q_{2x} \cdot \sin(\text{pos})$, then for y, we compute $q_{1y} \cdot \sin(\text{pos}) + q_{2y} \cdot \cos(\text{pos})$.

Using this modified positional encoding, we conduct additional experiments on the zero-shot and the fine-tune scenario in Gaussian Process regression tasks using the same settings in the main paper to evaluate the impact of RoPE on the performance of our model.

We conducted ablation experiments on sinusoidal PE and RoPE in a zero-shot scenario by inferring on 1D, 2D, 3D, 4D, and 5D GP regression data using DANP models trained on 2D and 4D GP

**Table 22:** Comparison of zero-shot performance between DANP trained with the sinusoidal positional embedding and RoPE. Here, each method trained with 2, and 4D GP dataset with RBF kernel while performing inference on the 1, 2, 3, 4, and 5D GP dataset with RBF kernel

| Positional Embedding | sinusoidal PE | | RoPE | |
|---|---|---|---|---|
| | context | target | context | target |
| 1D RBF | 1.336 ±0.047 | **0.806** ±0.048 | **1.352** ±0.012 | 0.777 ±0.035 |
| 2D RBF | **1.383** ±0.000 | **0.340** ±0.007 | **1.383** ±0.000 | 0.348 ±0.003 |
| 3D RBF | 1.377 ±0.007 | **-0.360** ±0.063 | **1.381** ±0.001 | -0.360 ±0.013 |
| 4D RBF | 1.379 ±0.007 | **-0.589** ±0.056 | **1.383** ±0.000 | -0.577 ±0.008 |
| 5D RBF | **1.357** ±0.012 | **-0.689** ±0.004 | 1.351 ±0.024 | -0.704 ±0.019 |

**Table 23:** Comparison of zero-shot performance between DANP trained with the sinusoidal positional embedding and RoPE. Here, each method trained with 2, 3, and 4D GP dataset with RBF kernel while performing inference on the 1, 2, 3, 4, and 5D GP dataset with RBF kernel

| Positional Embedding | sinusoidal PE | | RoPE | |
|---|---|---|---|---|
| | context | target | context | target |
| 1D RBF | 1.366 ±0.004 | **0.826** ±0.018 | **1.367** ±0.002 | 0.787 ±0.021 |
| 2D RBF | **1.383** ±0.000 | **0.355** ±0.014 | 1.382 ±0.000 | 0.334 ±0.007 |
| 3D RBF | **1.383** ±0.000 | -0.261 ±0.025 | **1.383** ±0.001 | **-0.256** ±0.006 |
| 4D RBF | **1.383** ±0.000 | **-0.568** ±0.042 | **1.383** ±0.002 | -0.576 ±0.036 |
| 5D RBF | 1.359 ±0.032 | **-0.676** ±0.004 | **1.367** ±0.014 | -0.679 ±0.007 |

regression data, as well as on 2D, 3D, and 4D GP regression data. The results, presented in Table 22 and Table 23, indicate that while sinusoidal PE consistently outperforms RoPE in the 1D case, their performance is largely similar across other dimensions. This suggests that for these scenarios, both sinusoidal PE and RoPE exhibit comparable interpolation and extrapolation capabilities.

We also conducted experiments using the trained models to perform few-shot learning on 1D GP regression, following the setup in the main paper. As shown in Table 24, while there were some performance differences in the zero-shot setting for the 1D GP regression task, these differences largely disappeared after few-shot fine-tuning. This indicates that the choice of positional embedding—whether sinusoidal PE or RoPE—has minimal impact on performance once the model is fine-tuned.

### D.6 ABLATION ON GP REGRESSION SETUP AND ZERO-SHOT EVALUATION

Because most of the models have trouble with extrapolation rather than interpolation, it is important to analyze our method's extrapolation capabilities as compared to its performance in interpolation settings. To address this, we conducted additional experiments by training on the $\{1, 2\}$, and $\{3, 4\}$ dimensional cases, then evaluating the results on $\{1, 2, 3, 4, 5\}$ dimensional test data.

Here, we train DANP utilizing both sinusoidal PE and RoPE to further analyze their generalization ability. Table 25 and Table 26 present the performance of DANP when trained on data from $\{1, 2\}$ dimensions and $\{3, 4\}$ dimensions, respectively.

From Table 25, we observe that when trained on the limited range of $\{1, 2\}$ dimensions, both positional embedding methods fail to learn sufficient general features, leading to lower generalization performance compared to training on $\{2, 4\}$ or $\{2, 3, 4\}$ dimensions. This result emphasizes the importance of training on higher-dimensional data to capture general features that enable better generalization to unseen dimensions. A similar pattern is evident in Table 26.

However, a distinct trend emerges in Table 25 compared to Table 22 and Table 23. While both sinusoidal PE and RoPE performed similarly when sufficient general features could be learned from more diverse training dimensions, RoPE demonstrates noticeably weaker generalization ability than

**Table 24:** Comparison of fine-tune performance between DANP trained with the sinusoidal PE and the RoPE. Here, each method trained with 2, and 4D GP datasets or 2, 3, and 4D GP datasets with RBF kernel while performing few-shot training on the 1D GP dataset with RBF kernel. Here, we report the performance for both the full fine-tuning and freeze finetuning

| Positional Embedding | Full finetuning | | Freeze finetuning | |
|---|---|---|---|---|
| | context | target | context | target |
| 2,4D sinusoidal PE | 1.375 ±0.001 | 0.890 ±0.004 | 1.375 ±0.001 | 0.889 ±0.002 |
| 2,3,4D sinusoidal PE | 1.375 ±0.000 | **0.893** ±0.004 | **1.376** ±0.001 | **0.890** ±0.005 |
| 2,4D RoPE | 1.375 ±0.001 | 0.886 ±0.020 | 1.374 ±0.001 | 0.884 ±0.015 |
| 2,3,4D RoPE | **1.376** ±0.000 | 0.882 ±0.006 | **1.376** ±0.001 | 0.882 ±0.007 |

**Table 25:** Comparison of zero-shot performance between DANP trained with the sinusoidal PE and the RoPE. Here, each method trained with 1, and 2D GP datasets with RBF kernel while performing inference on the 1, 2, 3, 4, and 5D GP datasets with RBF kernel.

| Positional Embedding | sinusoidal PE | | RoPE | |
|---|---|---|---|---|
| | context | target | context | target |
| 1D RBF | **1.381** ±0.000 | **0.916** ±0.003 | **1.381** ±0.012 | **0.916** ±0.002 |
| 2D RBF | **1.383** ±0.000 | 0.346 ±0.001 | **1.383** ±0.000 | **0.350** ±0.006 |
| 3D RBF | **1.307** ±0.004 | **-0.633** ±0.030 | 1.056 ±0.204 | -0.919 ±0.172 |
| 4D RBF | **1.138** ±0.012 | **-0.817** ±0.005 | 0.101 ±0.676 | -1.685 ±0.416 |
| 5D RBF | **0.885** ±0.022 | **-0.961** ±0.069 | -1.223 ±0.758 | -2.899 ±0.360 |

sinusoidal PE when the training data is limited to the narrow dimensional range of $\{1, 2\}$. This result highlights the dependency of RoPE on richer training data which contains richer general features to achieve high generalization ability.

### D.7 Additional extrapolation results for the fine-tuning scenario

We conducted additional fine-tuning experiments on 5 d GP regression data to analyze the extrapolation ability of our method. In this experiment, we aim to compare not only the performance of a single DANP model against the baselines but also evaluate and compare multiple variants of DANP trained on different dimensional GP data. Specifically, we include DANP models trained on $\{1, 2\}$, $\{3, 4\}$, $\{2, 4\}$, and $\{2, 3, 4\}$ dimensional GP data, as well as the corresponding DANP models where sinusoidal PE is replaced with RoPE.

The results in Table 27 clearly demonstrate that DANP outperforms the baselines in extrapolation few-shot scenarios, showcasing its robustness in handling these challenging tasks. Additionally, we observe that the DANP trained with 1,2d RoPE shows a notable improvement in generalization performance when provided with a few-shot setting. However, despite this improvement, its performance on the target data remains inferior compared to other DANP training settings, such as those utilizing higher-dimensional data ($\{3, 4\}$, $\{2, 4\}$, or $\{2, 3, 4\}$) or sinusoidal PE.

### D.8 Training both GP regression and Image completion

To further demonstrate the ability of DANP to learn various tasks simultaneously, we conducted an experiment involving both GP regression tasks and image completion tasks. Specifically, we trained our model on 2 and 3-dimensional GP regression tasks with the RBF kernel, as well as on EMNIST and CelebA image completion tasks. We then evaluated our model using an additional 1-dimensional GP regression task. As shown in Table 28, although the performance slightly decreased compared to training each task separately, DANP successfully learned all training tasks and generalized well to the unseen task. This demonstrates that DANP is capable of simultaneously training on diverse

**Table 26:** Comparison of zero-shot performance between DANP trained with the sinusoidal PE and the RoPE. Here, each method trained with 3, and 4D GP datasets with RBF kernel while performing inference on the 1, 2, 3, 4, and 5D GP datasets with RBF kernel.

| Positional Embedding | sinusoidal PE | | RoPE | |
|---|---|---|---|---|
| | context | target | context | target |
| 1D RBF | 1.130 ±0.042 | **0.501** ±0.016 | **1.239** ±0.021 | 0.472 ±0.019 |
| 2D RBF | 1.301 ±0.008 | 0.178 ±0.010 | **1.369** ±0.001 | 0.248 ±0.012 |
| 3D RBF | **1.383** ±0.000 | -0.278 ±0.005 | **1.383** ±0.001 | **-0.265** ±0.002 |
| 4D RBF | **1.383** ±0.000 | -0.582 ±0.014 | **1.383** ±0.000 | **-0.556** ±0.006 |
| 5D RBF | **1.359** ±0.012 | **-0.701** ±0.015 | 1.242 ±0.024 | -0.726 ±0.044 |

**Table 27:** Comparison of fine-tuning performance between DANP with various settings and the baselines. Here, we use the few-shot 5d GP regression task with RBF kernel for the evaluation. We also compare the performances for both full finetuning and freeze finetuning for all models.

| Method | Full fine-tuning | | Freeze fine-tuning | |
|---|---|---|---|---|
| | context | target | context | target |
| ANP | -0.851 ±0.017 | -0.852 ±0.016 | -0.837 ±0.024 | -0.837 ±0.025 |
| BANP | -0.817 ±0.012 | -0.813 ±0.011 | -0.830 ±0.013 | -0.828 ±0.016 |
| CANP | -0.854 ±0.026 | -0.856 ±0.022 | -0.847 ±0.057 | -0.851 ±0.050 |
| MPANP | -0.862 ±0.081 | -0.863 ±0.083 | -0.910 ±0.016 | -0.911 ±0.015 |
| TNP | -0.825 ±0.081 | -0.831 ±0.083 | -0.830 ±0.021 | -0.831 ±0.023 |
| 2,4D sinusoidal PE | **1.382** ±0.005 | -0.674 ±0.003 | **1.382** ±0.001 | -0.674 ±0.003 |
| 2,3,4D sinusoidal PE | **1.382** ±0.001 | **-0.672** ±0.004 | **1.382** ±0.001 | **-0.671** ±0.006 |
| 1,2D sinusoidal PE | 1.301 ±0.020 | -0.772 ±0.034 | 1.300 ±0.021 | -0.774 ±0.030 |
| 3,4D sinusoidal PE | 1.377 ±0.006 | -0.683 ±0.004 | 1.377 ±0.006 | -0.684 ±0.004 |
| 2,4D RoPE | 1.381 ±0.001 | **-0.672** ±0.001 | **1.382** ±0.001 | -0.672 ±0.001 |
| 2,3,4D RoPE | 1.382 ±0.000 | **-0.672** ±0.003 | **1.382** ±0.001 | -0.672 ±0.004 |
| 1,2D RoPE | **1.126** ±0.010 | -0.901 ±0.006 | 1.124 ±0.009 | -0.903 ±0.005 |
| 3,4D RoPE | 1.371 ±0.009 | -0.693 ±0.023 | 1.374 ±0.006 | -0.691 ±0.021 |

tasks and generalizing across different tasks. The model's performance could be further improved by increasing its capacity, either by expanding the feature space or adding more layers.

## D.9 ADDITIONAL EXTRAPOLATION EXPERIMENTS FOR THE IMAGE COMPLETION TASK

We conducted an additional experiment on the CelebA landmark (Liu et al., 2015) task to further demonstrate the capabilities of our method. In the standard CelebA landmark task, the goal is to predict the locations of five facial landmarks: left eye, right eye, left mouth corner, right mouth corner, and nose, based on a single image. However, since Neural Processes predict a distribution over the target points using a given context, we adapted the CelebA landmark task to better fit this approach. We modified the task by combining the image's RGB values with the corresponding coordinates for each landmark, creating a 5-dimensional input. The output was restructured as a 5-dimensional label representing which of the five facial regions the prediction corresponds to. This setup allowed us to train and evaluate the model in a way that aligns with the predictive distribution framework of Neural Processes.

For the experiment, we used pre-trained models for the baselines, specifically the CelebA image completion models, while we trained DANP on both the EMNIST dataset and CelebA image completion tasks. This approach allowed us to assess the performance of DANP under a slightly modified but challenging setup, testing its ability to generalize across different types of tasks. Table 29 validates that DANP still performs well on the different types of tasks compared to other baselines.

**Table 28:** Additional results for training both GP regression tasks and image completion tasks. We trained DANP with 2 and 3-dimensional GP dataset and EMNIST and CelebA image completion tasks.

| Dataset | context | target |
|---|---|---|
| 1D RBF | 1.299 ±0.023 | 0.710 ±0.032 |
| 2D RBF | 1.381 ±0.000 | 0.294 ±0.005 |
| 3D RBF | 1.381 ±0.000 | -0.313 ±0.020 |
| EMNIST | 1.382 ±0.000 | 0.888 ±0.004 |
| CelebA | 4.148 ±0.000 | 1.895 ±0.024 |

**Table 29:** Experimental results on the modified CelebA landmark task. Here, we fine-tuned baselines with 100-shot CelebA landmark dataset.

| Method | Full fine-tuning | | Freeze fine-tuning | |
|---|---|---|---|---|
| | context | target | context | target |
| ANP | 0.572 ±0.024 | 0.557 ±0.027 | 0.568 ±0.022 | 0.554 ±0.027 |
| BANP | 0.636 ±0.031 | 0.574 ±0.020 | 0.628 ±0.027 | 0.568 ±0.023 |
| CANP | 0.525 ±0.030 | 0.506 ±0.028 | 0.523 ±0.031 | 0.504 ±0.028 |
| MPANP | 0.536 ±0.036 | 0.485 ±0.023 | 0.535 ±0.034 | 0.487 ±0.024 |
| TNP | 0.658 ±0.020 | 0.557 ±0.035 | 0.653 ±0.021 | 0.554 ±0.033 |
| DANP(ours) | **1.354** ±0.001 | **0.674** ±0.007 | **1.340** ±0.002 | **0.672** ±0.005 |

For the zero-shot scenario, DANP achieves $1.171 \pm 0.020$ for the context dataset and $0.252 \pm 0.003$ for the target dataset. These results demonstrate that although the target likelihood of zero-shot DANP is lower compared to that of fine-tuned baselines—primarily due to variations in both input and output dimensions from the training data—DANP quickly surpasses other baselines after fine-tuning. This highlights DANP's robust ability to generalize effectively in challenging zero-shot scenarios while rapidly improving with minimal fine-tuning.

## D.10 BAYESIAN OPTIMIZATION

To illustrate the wide-ranging applicability of DANP, we conducted BO (Brochu et al., 2010) experiments across various scenarios: 1) a 1-dimensional BO experiment using objective functions derived from GPs with an RBF kernel, 2) 2 and 3-dimensional BO benchmarks, and 3) hyperparameter tuning for a 3-layer CNN (LeCun et al., 1989) on the CIFAR-10 (Krizhevsky et al., 2009) classification task. Following Nguyen & Grover (2022), we utilized Ackley, Cosine, and Rastrigin benchmark functions as the objective functions for the 2 and 3-dimensional BO experiments. For the hyperparameter tuning of the 3-layer CNN, we initially trained 1000 CNN models with varying hyperparameters, including learning rate, batch size, and weight decay, and then identified the optimal hyperparameter combination using NP models. We measured performance using *best simple regret*, which measures the difference between the current best value and the global best value. And, we run 50 iterations for all the BO experiments. For detailed information about the objective functions in the 2 and 3-dimensional BO and CNN training, see Appendix C. As baselines, we employed pre-trained models for each $n$-dimensional GP regression task corresponding to the $n$-dimensional BO tasks. In contrast, for DANP, we used a single model pre-trained with 2, 3, and 4-dimensional GP regression tasks in the **Zero-shot** scenario. The results in Fig. 5 demonstrate that DANP outperforms other baselines in terms of regret with same iteration numbers. This demonstrates that DANP is capable of serving as a surrogate model for different BO tasks using only a *single model* without additional training using BO datasets. In Fig. 5, we only report BO results with 2-dimensional Cosine and 3-dimensional Ackley objective function among various 2 and 3-dimensional BO benchmarks.

**Full results for the synthetic Bayesian Optimization**  Here, we present the comprehensive experimental results for 2 and 3-dimensional BO benchmark objective functions, including Ackley, Cosine,

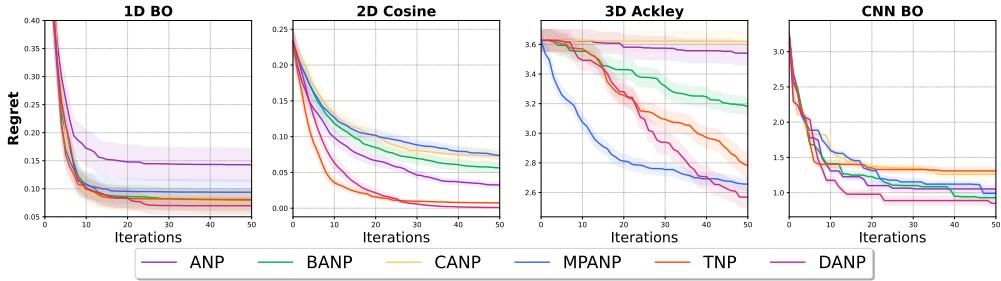

**Figure 5:** Results for BO with various BO tasks. These four figures, from left to right, show the regret results for 1-dimensional GP with RBF kernel, 2-dimensional cosine, 3-dimensional Ackley, and the CNN BO experiments.

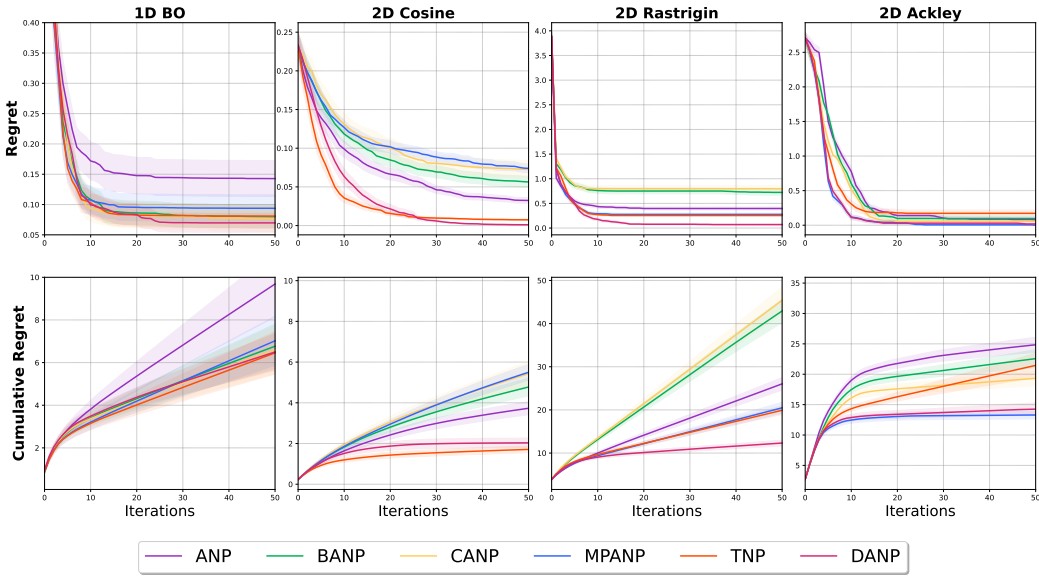

**Figure 6:** Full Results for BO with 1-dimensional GP generated BO tasks and 2-dimensional benchmark BO tasks. Here, we present cumulative regret results in addition to regret results.

and Rastrigin. Additionally, we report cumulative regret results alongside regret results for all BO experiments. Similar to the BO experiments outlined in Appendix D.10, we employed pre-trained models for each $n$-dimensional GP regression task corresponding to the $n$-dimensional BO tasks as baselines. In contrast, for DANP, we utilized a single model pre-trained with 2, 3, and 4-dimensional GP regression tasks in the **Zero-shot** scenario. The results depicted in Fig. 6 and Fig. 7 demonstrate that DANP is proficient in serving as a surrogate model for various BO tasks using only a *single model*, without requiring additional training on BO datasets.

### D.11 IMAGE COMPLETION AND VIDEO COMPLETION

**Additional visualization examples for the Image completion and Video completion** In this section, we provide additional visualization examples for the image completion task using the EMNIST and CelebA datasets, as well as for the video completion task with the CelebA video datasets. First, in Fig. 8, we display true video examples generated using the process described in Appendix C. It is visually apparent that the images gradually become darker over time. Next, we visualize 10 example images from the EMNIST and CelebA datasets. In Fig. 9 and Fig. 10, the full images are shown in the first column and the context points in the second column. Following that, we sequentially visualize the predicted mean and variance of CANP, ANP, BANP, MPANP, TNP, and DANP. And finally, in Fig. 11, we report predictive mean and variance of video trained DANP.

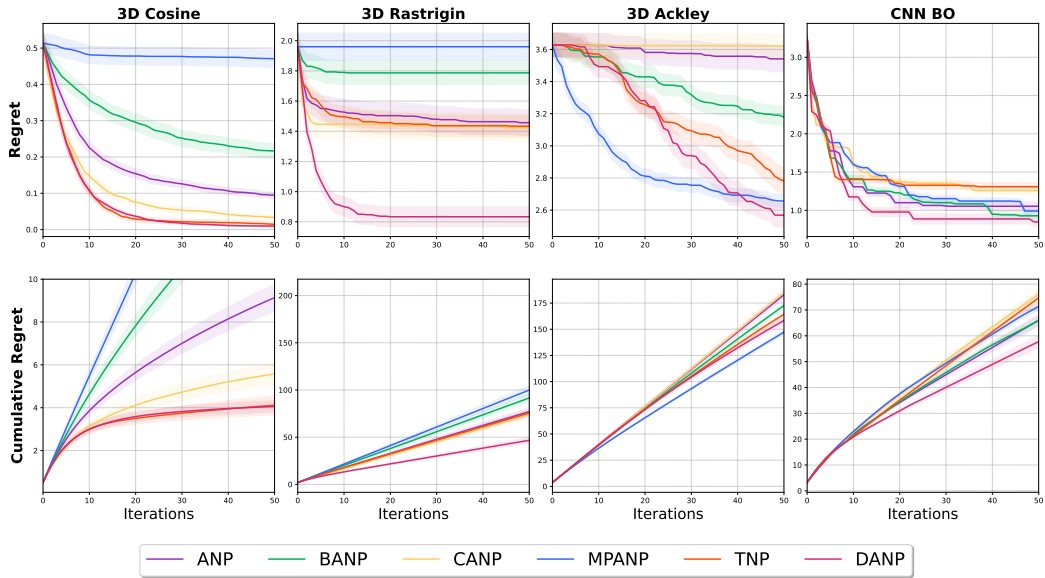

**Figure 7:** Full Results for BO with 3-dimensional benchmark BO tasks and CNN BO tasks. Here, we present cumulative regret results in addition to regret results.

**Table 30:** Empirical results on the time series blood pressure estimation task.

| Method | Full fine-tuning | | Freeze fine-tuning | |
| --- | --- | --- | --- | --- |
| | context | target | context | target |
| CANP | 0.964 ±0.030 | 0.875 ±0.024 | 0.962 ±0.031 | 0.870 ±0.022 |
| ANP | 1.037 ±0.021 | 0.950 ±0.017 | 1.035 ±0.021 | 0.947 ±0.019 |
| BANP | 1.104 ±0.018 | 0.968 ±0.011 | 1.100 ±0.017 | 0.966 ±0.012 |
| MPANP | 1.012 ±0.016 | 0.938 ±0.018 | 1.010 ±0.014 | 0.930 ±0.010 |
| TNP | 1.165 ±0.020 | 0.987 ±0.013 | 1.160 ±0.022 | 0.986 ±0.011 |
| DANP(ours) | **1.235** ±0.001 | **1.184** ±0.006 | **1.230** ±0.002 | **1.180** ±0.005 |

### D.12 TIME-SERIES EXPERIMENT

To further validate the practicality, we conducted additional experiments on time series data using the blood pressure estimation task from the MIMIC-III dataset (Johnson et al., 2016). Specifically, we assumed real-world scenarios where certain features from patient data might be missing, or entirely different sets of features could be collected. Under this assumption, we trained the model using only a subset of features from the MIMIC-III dataset and evaluated its performance when additional or different sets of features became available.

Specifically, we considered five features: T, Heart Rate, Respiratory Rate, SpO2, and Temperature. For pre-training, we utilized T and Heart Rate features, while for the fine-tuning scenario, we assumed only Respiratory Rate, SpO2, and Temperature features were available (this scenario can happen if we assume that we trained our model with data from hospital A and want to evaluate on the data in hospital B). We pre-trained the models with 32,000 training samples and fine-tuned them with only 320 samples. And here, we considered observations from time $0, ..., t$ as context points and $t+1, ..., T$ as target points. As shown in Table 30, our DANP achieved strong performance in the time series blood pressure estimation task, demonstrating robustness and adaptability in this real-world scenario. These results are consistent with the findings presented in the main paper, further validating DANP's effectiveness in handling diverse and practical challenges.

|   | t=0 | t=1 | t=2 | t=3 | t=4 | t=5 | t=6 | t=7 | t=8 | t=9 |

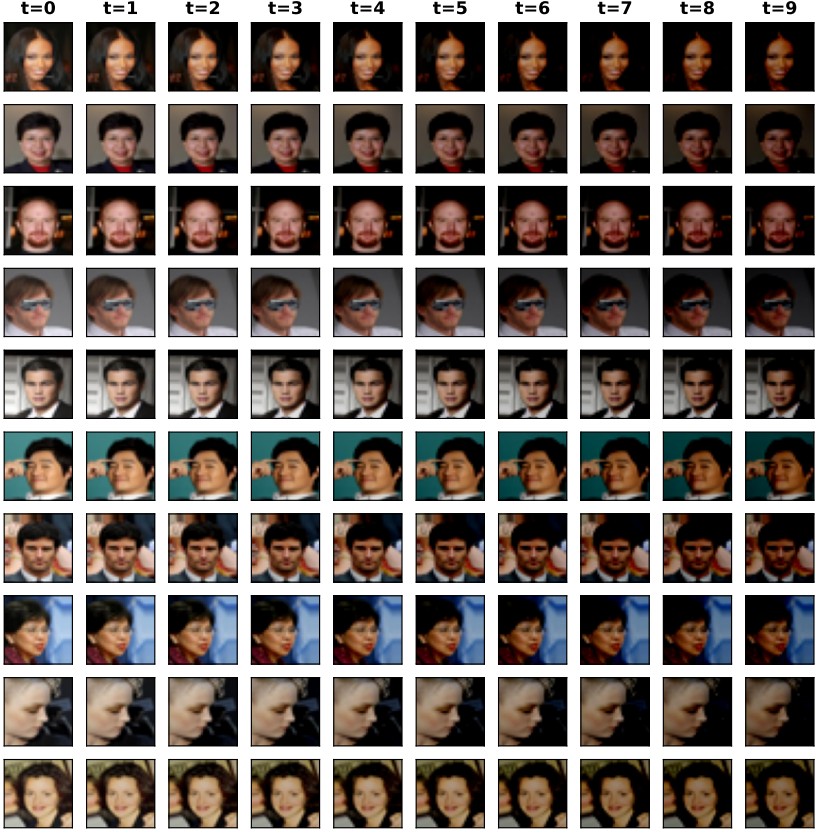

**Figure 8:** Examples of video data constructed following Appendix C.

**Table 31:** Wall clock time evaluation for the TNP and DANP in various settings. Here, we utilize RTX 3090 GPU for the evaluation.

| Model | 1D regression | 2D regression | EMNIST | CelebA |
|---|---|---|---|---|
| TNP | 1 min 30 sec | 1 min 50 sec | 1 min | 1 min 20 sec |
| DANP | 1 min 50 sec | 2 min 40 sec | 1 min 20 sec | 1 min 40 sec |

D.13 DISCUSSION ON THE RESOURCE REQUIREMENTS

Here, we will analyze the time complexity compared to the TNP both theoretically and practically.

First theoretically, let us denote $B$, $N$, $d_x$, $d_y$, $d_r$, $L_d$, and $L_l$ denote the batch size, the number of data points (union of context and target), the dimension of input $x$, the dimension of output $y$, the representation dimension, the number of layers in the deterministic path, and the number of layers in the latent path, respectively. The additional computational cost for the DAB module is $O(BN(d_x + d_y)^2)d_r)$, and for the latent path, it is $O(L_l BN^2 d_r)$. Since the computational cost for TNP is $O(L_d BN^2 d_r)$, the overall computational cost of DANP can be expressed as $O((L_l +$

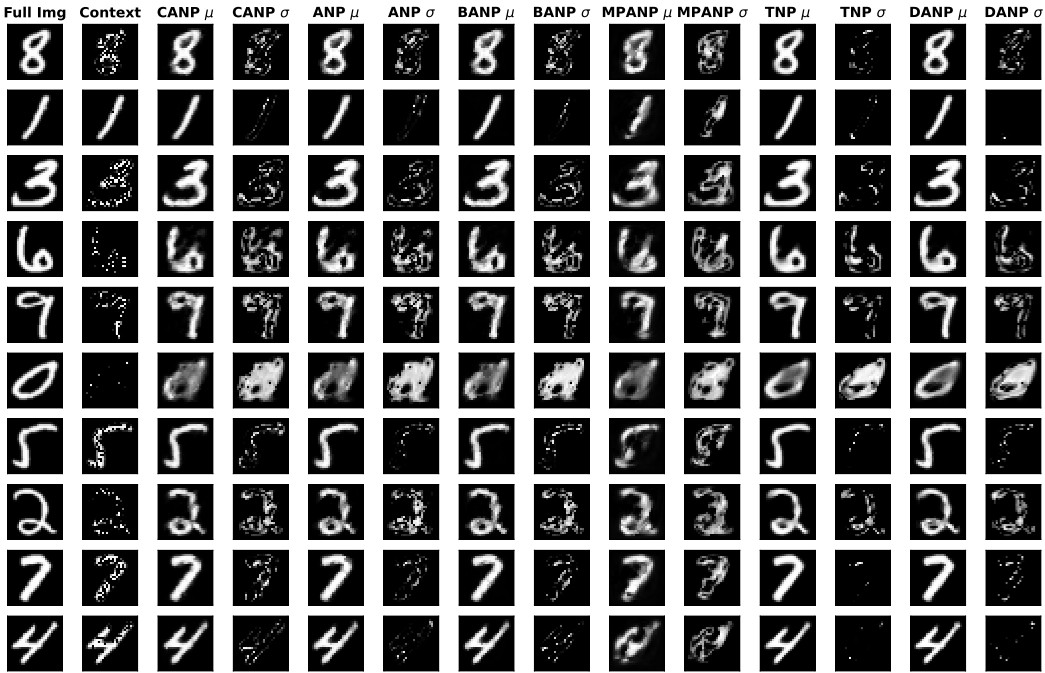

**Figure 9:** Predicted mean and variance of EMNIST dataset with baselines and DANP.

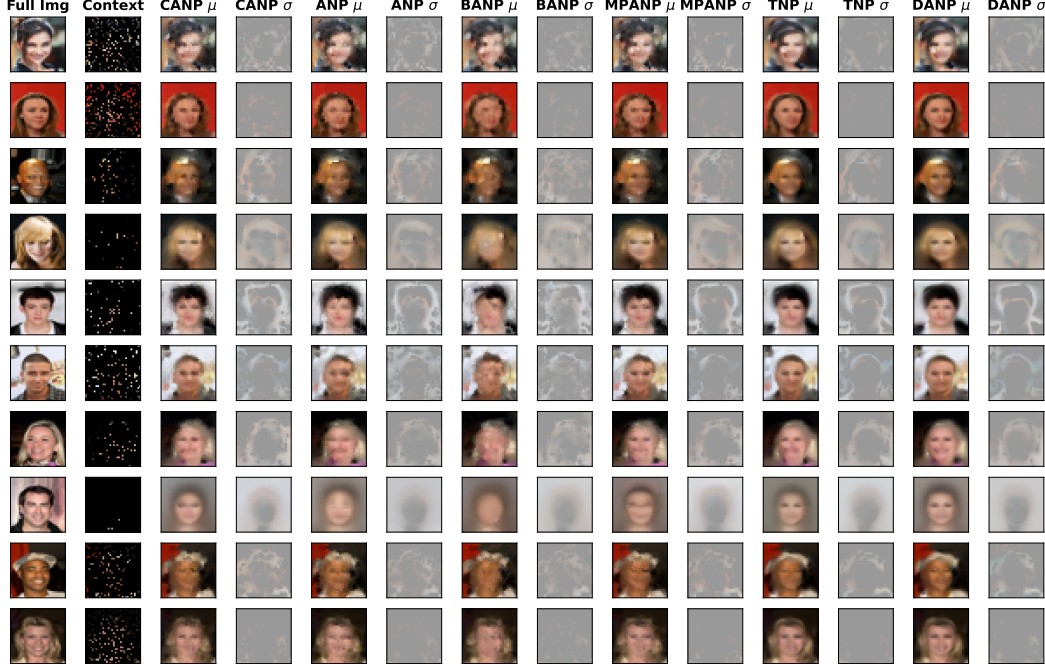

**Figure 10:** Predicted mean and variance of CelebA dataset with baselines and DANP.

$L_d)BN^2d_r) + O(BN(d_x + dy)^2d_r)$. Generally, since $N >> (d_x + d_y)^2$ holds, the dominant term in the computational cost can be approximated as $O((L_l + L_d)BN^2d_r)$.

For the practical time cost, we measure the time cost to train 5000 steps for the GP regression tasks and image completion tasks for TNP and DANP. The results are shown in Table 31.

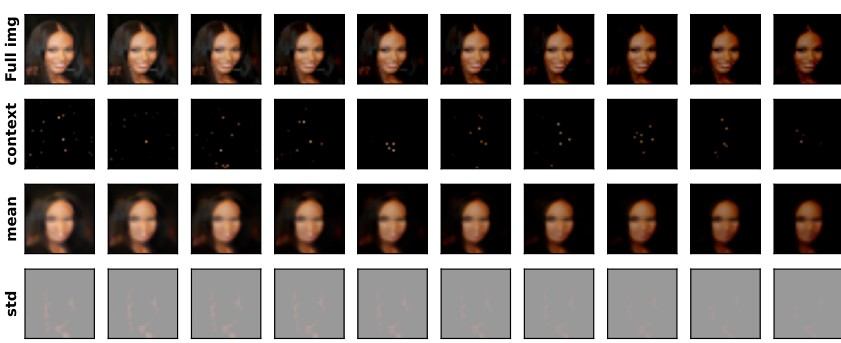

**Figure 11:** Predicted mean and variance of video data with DANP when training with video dataset.

