# OpenReview forum: "Dimension Agnostic Neural Processes"
_ICLR.cc/2025/Conference — ICLR 2025 Poster_

### Official Review · Reviewer_1CSd · 2024-10-28

**Soundness:** 3
**Presentation:** 3
**Contribution:** 2
**Rating:** 6
**Confidence:** 3

**Summary:**

The paper describes the existing challenges of Neural Processes (NP) when using a variable number of input dimensions and learned features. The authors proposed to tackle this problem by extending the Transformer Neural Process architecture with a Dimension Aggregator Block, using Positional Embeddings to take into account the different input dimensions before transforming the features into a fixed dimensional space. The paper tests the performance of the proposed method on zero-shot and fine-tuning settings using synthetic regression tasks and image completion datasets well-known in the NP literature.

**Strengths:**

- The paper has a good and concise summary of the Neural Processes setting and it highlights clearly the problem of fixed dimensions, as opposed to variable ones.
- The authors submitted the source-code to reproduce their experiments, which is a neat way to connect the ideas exposed on the main body with practical details of the implementation. Great work here.
- The idea of leveraging positional embeddings on the dimensions axis is I believe novel, and interesting on itself.

**Weaknesses:**

Post rebuttal: The authors have addressed many of the weaknesses I've listed below, added new ablations that show case the limitations of extrapolation and RoPE vs Positional Ebeddings among others. Hence I've increased my score. Reviewer utkY still has some methodological and novelty concerns, which I think are valid points, but I still think the empirical value of this work is enough to be accepted. Thanks to the authors for engaging on the rebuttal process.

**Limitations of positional embeddings on the proposed DAB module:**

Positional Embeddings is a well-studied part of the transformer architecture, specially in Language Models. Generally speaking, while there has been progress since the original Transformer, the community agrees that length extrapolation is an open-problem and positional embeddings do not extrapolate on further sequence lengths, see [1] for example, where the sinusoidal embeddings used by DAB are the worst in extrapolation. In LLMs (arguably the biggest current application of transformers and positional embeddings) the community has largely moved on from Sinusoidal embeddings and into other approaches such as RoPE [2] [3]. Thereby, while the setting here is very different I see the following weaknesses with the current paper given that positional embeddings is a crucial part for the DAB module to be dimension-agnostic:

- Concretely, Positional Embeddings have a hard time in transformers when extrapolating from small to large context lengths. This warrants a discussion on whether they’re applicable to the setting on this paper.
- There has been a substantial amount of work on newer and better positional embeddings, I believe this warrants at least a discussion and acknowledgement of recent work, and a stronger argument for using Sinusoidal Embeddings.
- At best this warrants an ablation which justifies the choice in the architecture.
- I believe highlighting the failure modes of using Positional Embeddings (sinusoidal or otherwise) do not diminish the contributions of the paper, quite on the contrary. But in my view, it’s important to highlight the limitations of the proposed approach, and if there’s evidence that these failure modes do not exist on the setting exposed here, it makes even an stronger paper.

**Experimental weaknesses (dimension generalisation):**

Connecting to the previous section, I believe that the experiments need to be more robust to analyse more carefully the failure modes of the proposed approach. In some cases, this is already in the appendix, but unfortunately this is not properly referenced from the main body. I urge the authors to present the failure modes of the approach in the main body more clearly, and when pushing results to the appendix, to reference them from the main body **explicitly,** calling out from the main body what are the strengths or weaknesses of the appendix results.

- On the 0-shot scenario of GP regression (section 5.1), the model is either trained on {2,4} or {2,3,4} dimensions and evaluated on {1,2,3,4,5,7}. Given the above discussion about positional embeddings, I think that this setting benefits evaluating on dimensions {1,3} since they’re covered by interpolation. A better experimental setting would involve fully non-overlapping dimensions in train and validation, such as {1,2,3} and {4,5,6}. Even better, would be to try different configurations and report the threshold where the performance breaks down or when the performance is better. I hypothesise this model is better when it is evaluated during interpolation. At the very least, there should be an open discussion about this in the paper; however, as this is core to the contribution of this work, I believe it’s important to address these experimental weaknesses. Some of this is already in Appendix B.2 and in Table 16, however it is not clear if the kernels are the same (names are present in table 16 but not in table 2a); irrespective of this, given the arguments above, I strongly believe this discussion on the limitation of the approach belongs to the main body and should be **clearly stated on the conclusions** **and contributions sections** since it paints a full picture of the failure modes. The whole appendix B is barely referenced (it’s very big with very different subsections), and it’s very easy to miss the discussion of table 16.
- For the fine-tuning section of GP regression, I have similar concerns. In the positional embedding LLMS literature, typically what you do to extend to longer contexts is to finetune on context lengths which are larger. Here, however, the finetuning happens on 1-dimensional tasks, after being pretrained on 2,3,4. Again, a likely failure mode is that when fine tuning on dimensions which are bigger than in pretraining the model won’t extrapolate very well. Unfortunately, as opposed to the zero-shot section, the appendix does not have experiments which reflect this setting.

**Experimental weaknesses (practical regression tasks)**

- While I understand the engineering challenges of video datasets, I believe the authors should tone down section 5.2. The proposed adaptation of CelebA does not make it a Video dataset — hence I believe it’s wrong to call this a Video Completion task, I suggest this is rephrased throughout the paper to reflect that this is just an image completion task with a synthetically generated extra dimension. Alternatively, it’d be really interesting to test this on an actual short video dataset, such as [4], however I understand this can be fully out of scope. Perhaps it’s useful to reference it as future work.
- I believe the extra dimension added to CelebA is a rather weak task, since it’s a very simple subtraction over the brightness. I wonder if using off-the-shelf dimensions such as CelebA landmarks [5] [6] would be a better task (there are 5 different landmarks locations for each face, tagged as 2d coordinates). This has the benefit of being an already established benchmark in the literature, and arguably a more real-world and practical regression task. If the proposed method does not work well on these harder tasks, it’s still interesting to highlight the limitations of the model.
- The core results of section 5.2 rely on pre-training on both the CelebA and EMNIST datasets, while interesting to see some positive transfer, after reading the framing of the paper and the GP regression section, I would have expected some zero-shot results on image tasks (for both pretraining and validation), which arguably is more relevant to meta-learning than fine-tuning. A Video Completion experiment in a zero-shot setting would make a stronger contribution, which is more consistent with the GP regressions section.

**Experimental weaknesses (other)**

- The log-likelihood is likely an incomplete picture of the results. As done in [7] I believe a stronger analysis should include other metrics such as calibration error.
- The context provided about confidence interval coverage is worth mentioning on the main body, instead of being in the appendix where it’s hardly accessible and referenced directly from the main body.
- In general, Appendix B is too broad and many interesting conclusions that should be in the main body are there without a direct reference.
- In my opinion, the details that have been mentioned to be in the appendix, deserve to be in the main body to have a better complete picture of the strengths and limitations of the proposed approach. While I understand the page-limit is a limitation, I’d argue that what I have mentioned is of more relevance compared to the mostly positive results of section 5.4 — these can be in the appendix and mentioned briefly with a direct reference for interested readers. Another alternative is to leave the tables in the appendix but still mention the relevant conclusion in the main body.
- NDP is mentioned as a relevant baseline and briefly tried on Appendix B - Table 11/12. Since this is the most relevant baseline according to the authors, I believe results with this relevant baseline from prior work should be in the main body as much as possible (it’s clear that it’s only comparable with dimension-agnostic methods for x when y=1) with a discussion about the quantitative results, not just qualitative as in section 4. There’s clearly at least two settings where it’s possible to do this, but they’re in the appendix.

**Misc:**

- The Dimension Aggregator Block implementation uses a linear projection **with bias [8].** This renders equation 10 incorrect since the bias term is missing there. I believe this is a common blind spot because the bias is set by default to True in Pytorch, so I urge the authors to revise all linear layers and either update the manuscript or the code/results. Note that I do not expect this to change the results much, but it’s better to address this for reproducibility and clarity.
- Table 2a does not have the 2nd or 1st best performance marked with underline or boldfaced underline. This makes it harder to read.
- Table 2a does not state which kernels were used, as compared to table 1 and table 16.
- It’s hard to read table 1 and table 2 separately, and as the authors mentioned, it’s helpful to read them in context.
- While I appreciate the detail problem setting on section 2, if there are concerns of page limits, I’d suggest the authors to prioritise space for experimental results and discussion as suggested in the previous section — this can be either added as an Appendix and referenced directly from the main body, or just cite from previous work as appropriate.

[1] https://openreview.net/forum?id=R8sQPpGCv0

[2] https://arxiv.org/abs/2104.09864

[3] https://arxiv.org/abs/2302.13971

[4] https://svdbase.github.io/

[5] https://mmlab.ie.cuhk.edu.hk/projects/CelebA.html

[6] https://www.kaggle.com/code/danmoller/inspecting-the-celeba-dataset-face-landmarks

[7] https://arxiv.org/abs/2207.04179

[8] https://pytorch.org/docs/stable/generated/torch.nn.Linear.html#torch.nn.Linear

**Questions:**

- In the 0-shot evaluation on GP regression (section 5.1), why is dimension 6 not included in the validation? It seems straightforward to do and would paint a complete picture. Can this experiment be included? If it’s a matter of space in the manuscript or computational resources, I believe dimension 6 is better than dimension 7.
- What are your thoughts on the inherent limitations of Positional Embeddings extrapolation? Is the lack of extrapolation a concern in the dimension-agnostic setting?
- Did you experiment with Zero-Shot on image tasks? Given that you had results for fine-tuning, I imagine it is possible to get zero-shot results, unless I’m missing something.

---

> ### Author Response · Authors · 2024-11-18
> **Rebuttal by Authors**
>
> **[Q1] Positional Embedding Extrapolation**
>
> Thank you for the constructive and informative comment regarding the limitations of sinusoidal positional encoding. As you pointed out, many previous works have shown that sinusoidal positional encoding tends to perform poorly in terms of generalization when extrapolating to longer sequence lengths for Large Language Models. In response to this, approaches like RoPE have been proposed and used to address these limitations. While sinusoidal positional encoding successfully handled interpolation and extrapolation in our experimental settings, as you suggested, RoPE could potentially improve this performance. Therefore, we conducted additional experiments using a modified RoPE-based encoding tailored for the DAB module.
>
> In our implementation, we retained the basic formulation of RoPE while ensuring different positional encodings for the x and y dimensions, similar to the approach we used with DAB. Specifically, we distinguished the embeddings added to queries and keys from x and y by alternating the cosine and sine multiplications for each. For example, if for x we calculate $ q_{1x} \cdot \cos(\text{pos}) + q_{2x} \cdot \sin(\text{pos}) $, then for y, we compute $ q_{1y} \cdot \sin(\text{pos}) + q_{2y} \cdot \cos(\text{pos}) $.
>
> Using this modified positional encoding, we conduct additional experiments on the zero-shot and the fine-tune scenario in Gaussian Process regression tasks using the same settings in the main paper to evaluate RoPE’s impact on our model’s performance.
>
> We conducted ablation experiments on sinusoidal PE and RoPE in a zero-shot scenario by inferring on 1D, 2D, 3D, 4D, and 5D GP regression data using DANP models trained on 2D and 4D GP regression data, as well as on 2D, 3D, and 4D GP regression data. The results, presented in Tables R.9 and R.10, indicate that while sinusoidal PE consistently outperforms RoPE in the 1D case, their performance is largely similar across other dimensions. This suggests that for these scenarios, both sinusoidal PE and RoPE exhibit comparable interpolation and extrapolation capabilities.
>
> We also conducted experiments using the trained models to perform few-shot learning on 1D GP regression, following the setup in the main paper. As shown in Table R.11, while there were some performance differences in the zero-shot setting for the 1D GP regression task, these differences largely disappeared after few-shot fine-tuning. This indicates that the choice of positional embedding—whether sinusoidal PE or RoPE—has minimal impact on performance once the model is fine-tuned.
>
> In the existing experimental settings presented in the main paper, we did not observe significant differences in performance or generalization ability between sinusoidal PE and RoPE. However, in experiments related to Q2, we identified certain scenarios where differences emerged. For further details and analysis, please refer to our response to Q2.

---

> > ### Comment · Reviewer_1CSd · 2024-11-19
> > **Question about RoPE formulation**
> >
> > Are you rotating the hidden states of x and y for each of the i-th dimensions? That is kind of the core of RoPE which makes it different than other methods (and why the base -- the theta -- hyper-parameter matters).

---

> ### Author Response · Authors · 2024-11-18
> **Rebuttal by Authors**
>
> __Table R.9__ Comparison of zero-shot performance between DANP trained with the sinusoidal PE and the RoPE. Here, each method trained with {2, 4}d GP dataset with RBF kernel while performing inference on the {1, 2, 3, 4, 5}d GP dataset with RBF kernel.
>
> |Positional Embedding|   sinusoidal PE         |   |              RoPE     |     |
> | :-                 | :-          | :-           | :-          | :-           |
> | MODEL              | CTX         |  TAR         | CTX         |  TAR         |
> | :-                 | :-          |      :-      | :-          | :-           |
> | 1d               | 1.336 ± 0.047 |**0.806**± 0.048 |**1.352**± 0.012 | 0.777 ± 0.035 |
> | 2d             | **1.383**± 0.000 | 0.340 ± 0.007 |**1.383**± 0.000 |**0.348**± 0.003 |
> | 3d             | 1.377 ± 0.007 |**-0.360**± 0.063|**1.381**± 0.001 |**-0.360**± 0.013 |
> | 4d              | 1.379 ± 0.007 | -0.589 ± 0.056|**1.383**± 0.000 |**-0.577**± 0.008 |
> | 5d              |**1.357**± 0.012 |**-0.689**± 0.004| 1.351 ± 0.024 |-0.704 ± 0.019 |
>
> __Table R.10__ Comparison of zero-shot performance between DANP trained with the sinusoidal PE and the RoPE. Here, each method trained with {2, 3, 4}d GP dataset with RBF kernel while performing inference on the {1, 2, 3, 4, 5}d GP dataset with RBF kernel.
>
> |Positional Embedding|   sinusoidal PE    |        |              RoPE|          |
> | :-                 | :-          | :-           | :-          | :-           |
> | MODEL              | CTX         |  TAR         | CTX         |  TAR         |
> | :-                 | :-          |      :-      | :-          | :-           |
> | 1d             | 1.366 ± 0.004 |**0.826**± 0.018 |**1.367**± 0.002 | 0.787 ± 0.003 |
> | 2d               |**1.383**± 0.000 |**0.335**± 0.014 | 1.382 ± 0.000 | 0.334 ± 0.007 |
> | 3d             |**1.383**± 0.000 | -0.261 ± 0.025|**1.383**± 0.001|**-0.256**± 0.006 |
> | 4d          |**1.383**± 0.000 |**-0.568**± 0.042|**1.383**± 0.002 |-0.576 ± 0.036 |
> | 5d                | 1.359 ± 0.032 |**-0.676**± 0.004|**1.367**± 0.014 |-0.679 ± 0.007 |
>
> __Table R.11__ Comparison of fine-tune performance between DANP trained with the sinusoidal PE and the RoPE. Here, each method trained with {2, 4}d GP dataset or {2, 3, 4}d GP dataset with RBF kernel while performing few-shot training on the 1d GP dataset with RBF kernel. Here, we report the performance for both the full finetuning and free finetuning.
>
> |Finetuning Method   |   Full finetuning    |      |   Freeze finetuning   |     |
> | :-                 | :-          | :-           | :-          | :-           |
> | MODEL              | CTX         |  TAR         | CTX         |  TAR         |
> | :-                 | :-          |      :-      | :-          | :-           |
> | 2,4d sinusoidal PE | 1.375 ± 0.001 | 0.890 ± 0.003 | 1.375 ± 0.001 | 0.889 ± 0.002 |
> |2,3,4d sinusoidal PE| 1.375 ± 0.000 |**0.893**± 0.004|**1.376**± 0.001|**0.890**± 0.005|
> | 2,4d RoPE          | 1.375 ± 0.001 | 0.886 ± 0.020  | 1.374 ± 0.001 | 0.884 ± 0.015 |
> | 2,3,4d RoPE        |**1.376**± 0.000 | 0.882 ± 0.006 |**1.376**± 0.001 | 0.882 ± 0.007 |

---

> ### Author Response · Authors · 2024-11-18
> **Rebuttal by Authors**
>
> **[Q2] GP Regression Setup and Zero-Shot Evaluation**
>
> Thank you for your insightful comment. The reason we set our experimental configurations to {2,4} and {2,3,4} was to comprehensively demonstrate our model’s performance in interpolation and both increasing and decreasing extrapolation scenarios.
>
> Specifically, training on {2,4} was chosen to first show that the model works well on the interpolative dimension of 3. Additionally, we aimed to evaluate its ability to generalize to the decreasing extrapolation (e.g., 1-dimensional) case, as well as the increasing extrapolation to 5 dimensions and further extrapolation to 7 dimensions. Although some might consider 1-dimensional inference as learned due to our training on {2,4} dimensions, it remains an extrapolation task because our attention mechanism must update the representation space using only the first dimension’s positional encoding without higher-dimensional support.
>
> While we could explore even larger extrapolative dimensions, GP data requires an exponentially growing number of context points as dimensions increase, making inference challenging. Therefore, 7 dimensions provided a balance for feasible inference. The subsequent training on {2,3,4} dimensions was included to show a clear performance gain on both interpolative and extrapolative tasks, underscoring how additional dimensions during training can boost performance across nearby dimensions.
>
> However, we clearly understand the importance of analyzing our method's extrapolation capabilities as compared to its performance in interpolation settings. To address this, we followed your advice and conducted additional experiments by training on the {1,2} and {3,4} dimensional cases, then evaluating the results on {1,2,3,4,5} dimensional test data.
>
> Here, we train DANP utilizing both sinusoidal PE and RoPE to further analyze their generalization ability. Tables R.12 and R.13 present the performance of DANP when trained on data from {1,2} dimensions and {3,4} dimensions, respectively.
>
> From Table R.12, we observe that when trained on the limited range of {1,2} dimensions, both positional embedding methods fail to learn sufficient general features, leading to lower generalization performance compared to training on {2,4} or {2,3,4} dimensions. This result emphasizes the importance of training on higher-dimensional data to capture general features that enable better generalization to unseen dimensions. A similar pattern is evident in Table R.13.
>
> However, a distinct trend emerges in Table R.12 compared to Tables R.9 and R.10. While both sinusoidal PE and RoPE performed similarly when sufficient general features could be learned from more diverse training dimensions, RoPE demonstrates noticeably weaker generalization ability than sinusoidal PE when the training data is limited to the narrow dimensional range of {1,2}. This result highlights the dependency of RoPE on richer training data which contains richer general features to achieve high generalization ability.
>
> In the final manuscript, we will add this discussion in the main paper's conclusion and clearly reference and analyze the related experimental results from the appendix within the main experiment section to provide a more comprehensive analysis.

---

> ### Author Response · Authors · 2024-11-18
> **Rebuttal by Authors**
>
> __Table R.12__ Comparison of zero-shot performance between DANP trained with the sinusoidal PE and the RoPE. Here, each method trained with {1, 2}d GP dataset with RBF kernel while performing inference on the {1, 2, 3, 4, 5}d GP dataset with RBF kernel.
>
> |Positional Embedding|   sinusoidal PE  |          |              RoPE    |      |
> | :-                 | :-          | :-           | :-          | :-           |
> | MODEL              | CTX         |  TAR         | CTX         |  TAR         |
> | :-                 | :-          |      :-      | :-          | :-           |
> | 1d             |**1.381**± 0.000 |**0.916**± 0.003 |**1.381**± 0.000|**0.916**± 0.002 |
> | 2d                 |**1.383**± 0.000 |0.346 ± 0.001|**1.383**± 0.000|**0.350**± 0.006 |
> | 3d                 |**1.307**± 0.004 |**-0.633**± 0.030|1.056 ± 0.204 |-0.919 ± 0.172 |
> | 4d                 |**1.138**± 0.012|**-0.817**± 0.005| 0.101 ± 0.676 |-1.685 ± 0.416 |
> | 5d                |**0.885**± 0.022 |**-0.961**± 0.069|-1.223 ± 0.758 |-2.899 ± 0.360 |
>
> __Table R.13__ Comparison of zero-shot performance between DANP trained with the sinusoidal PE and the RoPE. Here, each method trained with {3, 4}d GP dataset with RBF kernel while performing inference on the {1, 2, 3, 4, 5}d GP dataset with RBF kernel.
>
> |Positional Embedding|   sinusoidal PE     |       |              RoPE        |  |
> | :-                 | :-          | :-           | :-          | :-           |
> | MODEL              | CTX         |  TAR         | CTX         |  TAR         |
> | :-                 | :-          |      :-      | :-          | :-           |
> | 1d                 | 1.130 ± 0.042 |**0.501**± 0.016|**1.239**± 0.021| 0.472 ± 0.019 |
> | 2d                 | 1.301 ± 0.008 | 0.178 ± 0.010|**1.369**± 0.000 |**0.248**± 0.012 |
> | 3d                 |**1.383**± 0.000|-0.278± 0.005|**1.383**± 0.001|**-0.265**± 0.002 |
> | 4d                |**1.383**± 0.000|-0.582 ± 0.014|**1.383**± 0.000|**-0.556**± 0.006 |
> | 5d                 |**1.359**± 0.012 |**-0.701**± 0.015|1.242 ± 0.024 |-0.726 ± 0.044 |

---

> > ### Comment · Reviewer_1CSd · 2024-11-19
> > **Tables R.12 and R.13**
> >
> > Thanks for running these extra ablations! I think they'll help to make a case for future work on top of this in the NP literature.

---

> ### Author Response · Authors · 2024-11-18
> **Rebuttal by Authors**
>
> **[Q3] Fine-Tuning Issues and Extrapolation**
>
> Following your suggestion, we conducted additional fine-tuning experiments on 5 d GP regression data to analyze the extrapolation ability of our method. In this experiment, we aim to compare not only the performance of a single DANP model against the baselines but also evaluate and compare multiple variants of DANP trained on different dimensional GP data. Specifically, we include DANP models trained on {1,2}, {3,4}, {2,4}, and {2,3,4} dimensional GP data, as well as the corresponding DANP models where sinusoidal PE is replaced with RoPE.
>
> The results in Table R.14 clearly demonstrate that DANP outperforms the baselines in extrapolation few-shot scenarios, showcasing its robustness in handling these challenging tasks. Additionally, we observe that the DANP trained with 1,2d RoPE shows a notable improvement in generalization performance when provided with a few-shot setting. However, despite this improvement, its performance on the target data remains inferior compared to other DANP training settings, such as those utilizing higher-dimensional data ({3,4}, {2,4}, or {2,3,4}) or sinusoidal PE.
>
> __Table R.14__ Comparison of fine-tuning performance between DANP with various settings and the baselines. Here, we use the few-shot 5d GP regression task with RBF kernel for the evaluation. We also compare the performances for both full finetuning and freeze finetuning for all models.
>
> |Finetuning Method   |   Full finetuning|          |   Freeze finetuning   |     |
> | :-                 | :-          | :-           | :-          | :-           |
> | MODEL              | CTX         |  TAR         | CTX         |  TAR         |
> | :-                 | :-          |      :-      | :-          | :-           |
> | anp                |-0.851 ± 0.017 |-0.852 ± 0.016 |-0.837 ± 0.024 |-0.837 ± 0.025 |
> |banp                |-0.817 ± 0.012 |-0.813 ± 0.011 |-0.830 ± 0.013 |-0.828 ± 0.016 |
> | canp               |-0.854 ± 0.026 |-0.856 ± 0.022  |-0.847 ± 0.057 |-0.851 ± 0.050  |
> | mpanp              |-0.862 ± 0.081 |-0.863 ± 0.083  |-0.910 ± 0.016 |-0.911 ± 0.015 |
> |tnpd                |-0.825 ± 0.081 |-0.831 ± 0.083  |-0.830 ± 0.021 |-0.831 ± 0.023 |
> |--------------------|-------------|-------------|--------------|--------------|
> | 2,4d sinusoidal PE|**1.382**± 0.005|-0.674 ± 0.003 |**1.382**± 0.001 |-0.674 ± 0.003 |
> |2,3,4d sinusoidal PE|**1.382**± 0.001|**-0.672**± 0.004|**1.382**± 0.001|**-0.671**± 0.006|
> | 1,2d sinusoidal PE | 1.301 ± 0.020 |-0.772 ± 0.034  | 1.300 ± 0.021 |-0.774 ± 0.030 |
> | 3,4d sinusoidal PE | 1.377 ± 0.006 |-0.683 ± 0.004  | 1.377 ± 0.006 |-0.684 ± 0.004 |
> |--------------------|-------------|-------------|--------------|--------------|
> | 2,4d RoPE          | 1.381 ± 0.001 |-0.672 ± 0.001 |**1.382**± 0.001 |-0.672 ± 0.001 |
> |2,3,4d RoPE        |**1.382**± 0.000|**-0.672**± 0.003|**1.382**± 0.001|-0.672 ± 0.004 |
> | 1,2d RoPE          | 1.126 ± 0.010 |-0.901 ± 0.006  | 1.124 ± 0.009 |-0.903 ± 0.005 |
> | 3,4d RoPE          | 1.371 ± 0.009 |-0.693 ± 0.023  | 1.374 ± 0.006 |-0.691 ± 0.021 |

---

> ### Author Response · Authors · 2024-11-18
> **Rebuttal by Authors**
>
> **[Q4] Video Completion Task**
>
> Thank you for suggesting the evaluation of zero-shot performance on the video completion task. We agree that reporting the zero-shot performance of DANP in this task provides a stronger contribution and maintains consistency with the GP regression experiments. Based on that actually, we already included zero-shot results in the main paper's Table 3 (b), indicated with a dagger symbol.
>
> The results clearly demonstrate that even without fine-tuning, DANP achieves performance that surpasses the few-shot fine-tuning results of other models. This highlights DANP's robust generalization capabilities and further validates its effectiveness in handling complex tasks like video completion, aligning well with the strengths observed in our GP regression experiments.
>
> Also, based on your suggestion, we conducted an additional experiment on the CelebA landmark task to further demonstrate the capabilities of our method. In the standard CelebA landmark task, the goal is to predict the locations of five facial landmarks: left eye, right eye, left mouth corner, right mouth corner, and nose, based on a single image. However, since Neural Processes predict a distribution over the target points using a given context, we adapted the CelebA landmark task to better fit this approach. We modified the task by combining the image's RGB values with the corresponding coordinates for each landmark, creating a 5-dimensional input. The output was restructured as a 5-dimensional label representing which of the five facial regions the prediction corresponds to. This setup allowed us to train and evaluate the model in a way that aligns with the predictive distribution framework of Neural Processes.
>
> For the experiment, we used pre-trained models for the baselines, specifically the CelebA image completion models, while we trained DANP on both the EMNIST dataset and CelebA image completion tasks. This approach allowed us to assess the performance of DANP under a slightly modified but challenging setup, testing its ability to generalize across different types of tasks. Table R.15 validates that DANP still performs well on the different types of tasks compared to other baselines. And for the zero-shot scenario, DANP achieves 1.171±0.020 for the context dataset and 0.252±0.003 for the target dataset. These results demonstrate that although the target likelihood of zero-shot DANP is lower compared to that of fine-tuned baselines—primarily due to variations in both input and output dimensions from the training data—DANP quickly surpasses other baselines after fine-tuning. This highlights DANP's robust ability to generalize effectively in challenging zero-shot scenarios while rapidly improving with minimal fine-tuning.
>
> __Table R.15__ Experimental results on the modified CelebA landmark task. Here, we fine-tuned baselines with 100-shot CelebA landmark dataset.
>
> |Finetuning Method   |   Full finetuning     |     |   Freeze finetuning   |     |
> | :-                 | :-          | :-           | :-          | :-           |
> | MODEL              | CTX         |  TAR         | CTX         |  TAR         |
> | :-                 | :-          |      :-      | :-          | :-           |
> |  anp               | 0.572 ± 0.024 | 0.557 ± 0.027 | 0.568 ± 0.022 | 0.554 ± 0.027 |
> |  banp              | 0.636 ± 0.031 | 0.574 ± 0.020 | 0.628 ± 0.027 | 0.568 ± 0.023 |
> |  canp              | 0.525 ± 0.030 | 0.506 ± 0.028  | 0.523 ± 0.031 | 0.504 ± 0.028 |
> |  mpanp             | 0.536 ± 0.036 | 0.485 ± 0.023  | 0.535 ± 0.034 | 0.487 ± 0.024 |
> |  tnpd              | 0.658 ± 0.020 | 0.557 ± 0.035  | 0.653 ± 0.021 | 0.554 ± 0.033 |
> |  DANP             |**1.345**± 0.001|**0.674**± 0.007|**1.340**± 0.002|**0.672**± 0.005|

---

> > ### Comment · Reviewer_1CSd · 2024-11-19
> > **RE: Video Completion Task**
> >
> > Oh sorry I totally missed the dagger symbol on the table. I suggest to try and see if you can make that more visible, perhaps mentioning this directly on section 5.2
> >
> > Thanks for running the celeba attributes ablation, it's an interesting formulation.

---

> ### Author Response · Authors · 2024-11-18
> **Rebuttal by Authors**
>
> **[Q5] Additional Metrics for Model Evaluation**
>
> Thank you for your question regarding additional model evaluation. In response to your suggestion, we measured and report the following 6 additional metrics:  1) Mean Absolute Error (MAE)**, 2) **Root Mean Square Error (RMSE)**, 3) **Coefficient of Determination ($R^2$)**, 4) **Root Mean Square Calibration Error (RMSCE)**, 5) **Mean Absolute Calibration Error (MACE)**, 6) **Miscalibration Area (MA)**. Except for $R^2$, lower values for all these metrics indicate better alignment with the target and improved calibration performance.  We conducted the evaluation using models trained on a 1d GP regression task, comparing our method with the baselines. The results, summarized in Table R.16, demonstrate that DANP achieves the best performance across a range of metrics. This observation reaffirms that DANP not only outperforms in terms of NLL but also achieves improved performance in calibration-related metrics compared to the baselines. These additional evaluations highlight the robustness of our method across diverse aspects of model performance. We will include this analysis in our revised manuscript to provide a more comprehensive evaluation of our approach.
>
> __Table R.16__ Results on additional metrics containing MAE, RMSE, $R^2$, RMSCE, MACE, and MA on 1d GP regression task. Except for $R^2$, lower values for all these metrics indicate better alignment with target and improved calibration performance.
>
> |Metric|MAE        |RMSE      |$R^2$      |RMSCE   |MACE      |MA           |
> |--------|-------------|--------------|-------------|--------------|--------------|-------------|
> |anp     |0.126±0.001|0.176±0.003 |0.788±0.012|0.273±0.007|0.238±0.003|0.240±0.003|
> |banp    |0.125±0.001|0.175±0.003|0.811±0.001|0.273±0.001|0.237±0.001|0.239±0.002|
> |canp   |0.127±0.001|0.178±0.002|0.801±0.005|0.267±0.008|0.239±0.002|0.237±0.005|
> |mpanp|0.124±0.001|0.173±0.003|0.807±0.005|0.274±0.014|0.242±0.007|0.244±0.008|
> |tnpd   |0.122±0.002|0.173±0.001|0.808±0.002|0.287±0.003|0.251±0.005|0.253±0.006|
> |danp  |**0.120**±0.001|**0.165**±0.002|**0.816**±0.002|**0.259**±0.002|**0.230**±0.003|**0.228**±0.002|
>
>
> **[Q6] Misc**
>
> Thank you for pointing out the errors in the code and tables. We appreciate your feedback, and we will make sure to address these issues. Specifically, we will ensure that the main paper accurately reflects the additional experimental results and provides a thorough analysis as per your suggestions. We are committed to refining the final manuscript accordingly and will make the necessary revisions to ensure clarity and consistency throughout.

---

> ### Author Response · Authors · 2024-11-20
> **Regarding question about RoPE formulation**
>
> Thank you for reading our response and asking follow-up questions. Actually, yes, we rotated the hidden states of $ x $ and $ y $ for each $ i $-th dimension to properly apply RoPE in the DAB module. And we'll make dagger more visible following your suggestion. Please feel free to let us know if you have any further questions or would like to discuss anything!

---

> > ### Comment · Reviewer_1CSd · 2024-11-21
> > **rope response**
> >
> > thanks! it'd be good to put those details in the manuscript too, since rope can be tricky to interpret sometimes

---

> > > ### Author Response · Authors · 2024-11-22
> > > **Response from the Authors**
> > >
> > > Thank you for the various discussions that helped improve our paper. We will make sure to reflect all the discussions and suggestions we had in our final manuscript. If there are no further questions, could you revise your score based on our discussion?

---

> > > > ### Comment · Reviewer_1CSd · 2024-11-22
> > > > **Update**
> > > >
> > > > I have updated my score. Thank you for engaging on the rebuttal.

---

> ### Author Response · Authors · 2024-11-22
>
> Thank you for your positive review and the constructive comments that suggested directions to better understand and improve our paper. We will structure the final manuscript carefully to make it more readable and self-contained.

---

### Official Review · Reviewer_utkY · 2024-10-31

**Soundness:** 2
**Presentation:** 2
**Contribution:** 2
**Rating:** 5
**Confidence:** 5

**Summary:**

This paper works on neural processes and studies the case when there exist diverse input dimensions and learned features. Tot this end, the Dimension Agnostic Block (DANP) is developed to transform input features into a fixed embedding and then combined with neural process modules. It conducts experiments in zero-shot and few-shot scenarios.

--post rebuttal--

After attending the discussion, I pose some revision suggestions on the notation system:

(1) Both NP and CNP employ a **global latent variable (in NP, $z$ is parameterized with an encoder to obtain the Gaussian distribution by inputting $D_c$, in CNP $z$ is a deterministic embedding of $D_c$)**, and this means **marginalization the global latent variable $z$ out is standard instead of local latent variables** in NPs' family. In CNP, there is no marginalization out operation due to the deterministic global one.

(2) Let's use the author's notation $\theta_j$ as the latent variable for the encoder, it is right to use $p(\theta\vert D_c)$ rather than the data-point specific $\theta_j$ as the latter means the local latent variable.

I accept this work conditionally on revising the notation system to make it misleading.

**Strengths:**

I can easily follow this work and the layout is clear. However, there are severe writing and examination issues.

**Weaknesses:**

(1) Overall, this work includes several engineering tricks to handle diverse input dimension cases and lacks theoretical analysis to examine the proposal.

(2) It seems the motivation of this work also considers the uncertainty, however, I did not see sufficient results to illustrate this part when the dimension of the output is high.

(3) In the related work [1], it has been demonstrated the Eq. (19) is not a valid ELBO. Hence, Line 266-269 should be revised. In line279, I disagree that NP is the earliest to address uncertainty as CNP can also achieve this in experiments. It seems several related works [2-9] are not discussed in the literature.

(4) In line 316 and the following experiments, I am afraid that the zero-shot and finetune scenarios are not appropriate in evaluation as NP families require the context and amortize the few-shot adaptation without gradient updates. I am not convinced by the meaning of fine-tune or zero-shot in the task concept.

(5) The computational complexity towards the modules and other ablations are required in examining the performance. Meanwhile it seems a lot of results in the context are nearly the same in scales in Table2-3. Details about the number of shots are missing in experiments, further weakening the results.

---

Reference:

[1] Foong A, Bruinsma W, Gordon J, et al. Meta-learning stationary stochastic process prediction with convolutional neural processes[J]. Advances in Neural Information Processing Systems, 2020, 33: 8284-8295.

[2] Ashman, Matthew, et al. "Translation Equivariant Transformer Neural Processes." arXiv preprint arXiv:2406.12409 (2024).

[3] Feng, Leo, et al. "Latent bottlenecked attentive neural processes." arXiv preprint arXiv:2211.08458 (2022).

[4] Bruinsma, Wessel P., et al. "Autoregressive conditional neural processes." arXiv preprint arXiv:2303.14468 (2023).

[5] Wang Q, Federici M, van Hoof H. Bridge the inference gaps of neural processes via expectation maximization[C]//The Eleventh International Conference on Learning Representations. 2023.

[6] Tailor D, Khan M E, Nalisnick E. Exploiting inferential structure in neural processes[C]//Uncertainty in Artificial Intelligence. PMLR, 2023: 2089-2098.

[7] Markou S, Requeima J, Bruinsma W P, et al. Practical conditional neural processes via tractable dependent predictions[J]. arXiv preprint arXiv:2203.08775, 2022.

[8] Wang Q, Van Hoof H. Learning expressive meta-representations with mixture of expert neural processes[J]. Advances in neural information processing systems, 2022, 35: 26242-26255.

[9] Feng, Leo, et al. "Memory efficient neural processes via constant memory attention block." arXiv preprint arXiv:2305.14567 (2023).

[10] Feng L, Hajimirsadeghi H, Bengio Y, et al. Efficient Queries Transformer Neural Processes[C]//Sixth Workshop on Meta-Learning at the Conference on Neural Information Processing Systems.

**Questions:**

See the above

---

> ### Author Response · Authors · 2024-11-18
> **Rebuttal by Authors**
>
> **[W4] The zero-shot and fine-tune experiments are not appropriate in evaluation as NP families require the context and amortize the few-shot adaptation without gradient updates. I am not convinced by the meaning of fine-tune or zero-shot in the task concept**
>
> Thank you for the constructive comment. Although this is not the reviewer utkY's first question, we would like to address this point first to ensure there is no misunderstanding about the zero-shot and fine-tuning scenarios before moving on to discuss other questions. It seems there was a misunderstanding regarding the experimental setup in our paper. By "zero-shot" and "fine-tuning" in our experiments, we did not mean performing zero-shot inference without a context set or fine-tuning on a given context before inferring on the target set. Instead, we refer to the process where a model trained on tasks consisting of a context set and target set is then used to perform inference on unseen tasks, which involve context and target sets with input dimensions that were not part of the original training data.
>
> For example, in Table 2, the model trained on tasks sampled from 2d, 3d, and 4d dimensional GPs was used to infer on unseen dimensional tasks, such as 1d, 5d, and 7d dimensional GPs. These unseen tasks still contain context and target sets, and inference is performed based on the given context. In this context, "zero-shot" refers to directly using DANP to infer on unseen dimensional tasks without additional training, which is a capability that other models cannot achieve. On the other hand, "fine-tuning" refers to experiments where a small number of training samples from the unseen task are used to further train both the baseline models and DANP to observe how performance improves.
>
> **[W1] Lacks theoretical analysis**
>
> While our proposed DANP model lacks theoretical analysis, we have provided extensive empirical analysis demonstrating that DANP performs more robustly in zero-shot and few-shot scenarios on varying-dimensional tasks compared to other baseline models. Specifically, in the GP regression task, we show that DANP achieves performance comparable to that of fully trained baselines solely through zero-shot inference, even without being trained on 1d regression data. This result demonstrates DANP's strong capability to generalize and capture shared features across previously unseen tasks with different dimensions.
>
> However, as you suggested, incorporating theoretical analysis would strengthen our paper further, and we see it as a valuable direction for future work.

---

> ### Author Response · Authors · 2024-11-18
> **Rebuttal by Authors**
>
> **[W2] Did not see sufficient results to illustrate uncertainty when the dimension of the output is high**
>
> Thank you for the constructive comment. As you noted, demonstrating DANP's ability to learn and handle tasks with varying output dimensions concurrently is crucial, especially as it highlights the model's robustness in high-dimensional outputs. To show this, we conducted experiments on image and video completion tasks in the main paper Table 3, training on both EMNIST data (output dimension 1) and CelebA data (output dimension 3) simultaneously. These experiments verified that our model can effectively learn and infer across different output dimensions. Furthermore, in the video completion task, we empirically demonstrated that DANP is capable of handling and making inferences on video data (output dimension 3) with only minimal task-specific finetuning, unlike other models. This reinforces the versatility and strength of our model in diverse scenarios.
>
> **[W3-1] It has been demonstrated that the Eq (19) is not a valid ELBO. Hence Line 266-269 should be revised.**
>
> Thank you for the insightful comment. As you pointed out, and as highlighted in [1], the ELBO loss we used does not provide the exact ELBO for the $- \log p_\theta(y_t|x_t, D_c)$, because we use $q(z|D_c)$ instead of $p(z|D_c)$. More precisely, the Maximum Likelihood Loss is a biased estimator of $- \log p_\theta(y_t|x_t, D_c)$, and the ELBO we used is a lower bound of the same quantity [2]. Therefore, both losses still share the same issue, and the effectiveness of each loss depends on the model.
>
> Typically, the maximum likelihood loss tends to exhibit larger variance compared to variational inference, so, given our model's need to handle multiple varying dimensional tasks simultaneously, we opted for variational inference to ensure stability. However, as you mentioned, it is worth experimenting with other loss functions. Therefore, we plan to include the results from training with the maximum likelihood loss as well.
>
> We conducted ablation experiments on the ML loss and VI loss using DANP trained on 2 and 4d GP data, as well as DANP trained on 2d, 3d, and 4d GP data. These experiments were performed in a zero-shot scenario by inferring on 1, 2, 3, 4, and 5d GP regression data. The results, presented in Tables R.6 and R.7, show that while ML loss occasionally yields better log-likelihoods for context points, the VI loss consistently provides superior performance for the target points, which are of greater interest during inference. This trend is particularly evident in experiments trained on 2, 3, and 4d GP data. These findings demonstrate that using the VI loss for training DANP is generally more beneficial for improving generalization compared to the ML loss.
>
> __Table R.6__ Comparison of zero-shot performance between DANP trained with the variational loss and the maximum likelihood loss. Here, each method trained with {2, 4}d GP dataset with RBF kernel while performing inference on the {1, 2, 3, 4, 5}d GP dataset with RBF kernel.
>
> |	Training loss	 |   VI                |       |                ML     |     |
> | :-                 | :-          | :-           | :-          | :-           |
> | MODEL              | CTX         |  TAR         | CTX         |  TAR         |
> | :-                 | :-          |      :-      | :-          | :-           |
> | 1d              | 1.336 ± 0.047 |**0.806**± 0.048 |**1.340**± 0.025 | 0.790 ± 0.008 |
> | 2d             |**1.383**± 0.000|**0.340**± 0.007 |**1.383**± 0.000 | 0.330 ± 0.012 |
> | 3d                 | 1.377 ± 0.007 |**-0.360**± 0.063|**1.381**± 0.001 |-0.420 ± 0.112 |
> | 4d                | 1.379 ± 0.007 |**-0.589**± 0.056|**1.383**± 0.000 |-0.614 ± 0.045 |
> | 5d                |**1.357**± 0.012 |**-0.689**± 0.004| 1.356 ± 0.040 |-0.701 ± 0.023 |
>
> __Table R.7__ Comparison of zero-shot performance between DANP trained with the variational loss and the maximum likelihood loss. Here, each method trained with {2, 3, 4}d GP dataset with RBF kernel while performing inference on the {1, 2, 3, 4, 5}d GP dataset with RBF kernel.
>
> |	Training loss	 |   VI            |           |                ML|          |
> | :-                 | :-          | :-           | :-          | :-           |
> | MODEL              | CTX         |  TAR         | CTX         |  TAR         |
> | :-                 | :-          |      :-      | :-          | :-           |
> | 1d                |**1.366**± 0.004 |**0.826**± 0.018 | 1.360 ± 0.006 | 0.805 ± 0.021 |
> | 2d                |**1.383**± 0.000 |**0.335**± 0.014 | 1.382 ± 0.000 | 0.285 ± 0.035 |
> | 3d              |**1.383**± 0.000 |**-0.261**± 0.025|**1.383**± 0.001 |-0.320 ± 0.044 |
> | 4d               |**1.383**± 0.000 |**-0.568**± 0.042| 1.381 ± 0.002 |-0.658 ± 0.039 |
> | 5d               | 1.359 ± 0.032 |**-0.676**± 0.004| **1.364**± 0.021 |-0.742 ± 0.006 |

---

> ### Author Response · Authors · 2024-11-18
> **Rebuttal by Authors**
>
> **[W3-2] Disagree that NP is the earliest to address uncertainty as CNP can also achieve this in experiments**
>
> Thank you for the insightful comment. It seems there was a misunderstanding in the message. As you correctly pointed out, CNP models are indeed designed to capture data (aleatoric) uncertainty. However, what we intended to convey in that sentence is that the Neural Processes paper is the first to attempt capturing model  (epistemic) uncertainty (uncertainty in the underlying function f) using a global latent path. We will make this clear in the revision.
>
> **[W3-3] It seems several works [2-9] are not discussed in the literature**
>
> Thank you for your constructive suggestion on improving our manuscript. Following your request, we will include a new, dedicated related work section in the appendix of the final manuscript. This section will provide a broader and more detailed discussion of related works to make the paper more accessible and comprehensible for readers. We appreciate your recommendation and believe this addition will enhance the overall quality and clarity of our work.
>
> **[W5-1] The computational complexity towards the modules and other ablations are required in examining the performance.**
>
> We appreciate your insightful question regarding the time complexity. We will analyze the time complexity compared to the Transformer Neural Processes(TNP) both in theoretically and practically.
>
> First in theoretically, let us denote $B$, $N$, $d_x$, $d_y$, $d_r$, $L_d$, and $L_l$ denote the batch size, the number of data points (union of context and target), the dimension of input $x$, the dimension of output $y$, the representation dimension, the number of layers in the deterministic path, and the number of layers in the latent path, respectively. The additional computational cost for the DAB module is $O(BN(d_x+d_y)^2)d_r)$, and for the latent path, it is $O(L_lBN^2d_r)$. Since the computational cost for TNP is $O(L_dBN^2d_r)$, the overall computational cost of DANP can be expressed as $O((L_l+L_d)BN^2d_r)+O(BN(d_x+dy)^2d_r$. Generally, since $N >> (d_x+d_y)^2$ holds, the dominant term in the computational cost can be approximated as $O((L_l+L_d)BN^2d_r).
>
> For the practical time cost, we measure the time cost to train 5000 steps for the GP regression tasks and image completion tasks for TNP and DANP. The results are shown in Table R.8.
>
> Also, regarding the ablation study, as you suggested, we believe that conducting an ablation study to examine how the DAB module and latent path affect model predictions provides a deeper understanding of each module's role. Consequently, in the main paper, Section 5.4 under the paragraph titled **The Role of the DAB Module and Latent Path**, we analyzed each module's function through ablation experiments.
>
> In summary of the results in Table 4, across various experimental settings, we observed that adding only the DAB module allows the model to handle data of varying dimensions, but it does not provide additional performance gains. In contrast, including the latent path results in performance improvements (indicating an enhanced capacity to learn shared features across tasks) but lacks the ability to manage varying dimensional data. These results demonstrate that the DAB module enables DANP to handle data with varying dimensions, while the latent path enhances the model’s ability to learn generally shared features across tasks of different dimensions.
>
> __Table R.8.__ Wall clock time evaluation for the TNP and DANP in various settings. Here, we utilize RTX 3090 GPU for the evaluation.
> | MODEL | 1D regression  | 2D regression  | EMNIST      | CelebA     |
> |-------|----------------|----------------|---------------|---------------|
> | TNP   | 1 min 30 sec | 1 min 50 sec | 1 min |1 min 20 sec |
> | DANP  | 1 min 50 sec | 2 min 40 sec | 1 min 20 sec | 1 min 40 sec |

---

> ### Author Response · Authors · 2024-11-18
> **Rebuttal by Authors**
>
> **[W5-2] Meanwhile it seems a lot of results in the context are nearly the same in scales in Table 2-3. Details about the number of shots are missing in experiments, further weakening the results.**
>
> Thank you for the feedback on the experimental results. To clarify, the similar likelihoods for the context in our experiments were due to our approach of modeling the standard deviation for all models' decoders' Gaussian distribution prediction as \(0.1 + 0.9 \cdot \text{softplus}\). This was done to ensure fair comparisons between models. By modeling the standard deviation in this way, we prevent the performance from being biased by differences in the way the standard deviation is handled across models, ensuring that the output form is consistent for both DANP and all baseline models.
>
> For the 1-dimensional output case, the performance of context point prediction reaching a value of 1.38 indicates a nearly perfect reconstruction. This result shows that most models were able to reconstruct the context points well in the given model setting. The similar values observed are consistent with other works, like in [3].
>
> Furthermore, we have provided all the information about the context and target set sizes used in the experiments in Appendix A.3. The term "shot" that seemed confusing refers to the information needed when experimenting with unseen tasks, and we have made sure to include this information either in the main text or Appendix A.3 for clarity.
>
> **References**
>
> [1] Foong A, Bruinsma W, Gordon J, et al. Meta-learning stationary stochastic process prediction with convolutional neural processes[J]. Advances in Neural Information Processing Systems, 2020, 33: 8284-8295.
>
> [2] Yann Dubois, Jonathan Gordon, and Andrew YK Foong, Neural Process Family. http://yanndubs.github.io/Neural-Process-Family/
>
> [3] Hyungi Lee, Eunggu Yun, Giung Nam, Edwin Fong, and Juho Lee. Martingale posterior neural processes. In International Conference on Learning Representations (ICLR), 2023.

---

> ### Author Response · Authors · 2024-11-22
>
> Thank you for your dedication and interest in our paper. As the author and reviewer discussion period approaches its end, we are curious to know your thoughts on our rebuttal and whether you have any additional questions.

---

> > ### Comment · Reviewer_utkY · 2024-11-22
> > **Still concerned about the contribution and evaluation**
> >
> > I thank the reviewer's detailed response. After reading the rebuttal and double-checking the manuscript, I am still concerned about the novelty and evaluation of this work.
> >
> > **Novelty**: it seems simply introducing the dimension aggregation module is not sufficient in contribution even though authors conduct extensive experiments (however, setups and evaluations seem problematic to some extent). In this case, theoretical analysis is necessary.
> >
> > **Evaluation**: the fine-tuning few-shot learner makes sense to me. But the zero-shot testing cannot convince me which contradicts with the original setup of neural process family. Meanwhile several evaluation and analysis parts require carefully revision.
> >
> > Given the above two points, I keep my score as reject. And thanks for the author's rebuttal.

---

> ### Author Response · Authors · 2024-11-22
>
> Thank you for your response to our rebuttal. However, we are curious why you believe the zero-shot setting contradicts the original setting of Neural Processes. Neural Processes are fundamentally designed to learn from various tasks and amortize the acquired knowledge to solve new, unseen tasks.
>
> Aligned with this goal, while different from our paper they do not focus on creating models agnostic to unseen dimensions, studies like [1] extend Transformer Neural Processes for in-context learning, solving new tasks with the same dimensions (e.g., training on CelebA and testing on CIFAR10 image completion tasks). In this case, these models also tackle tasks in a zero-shot manner without fine-tuning on new tasks, instead relying on in-context learning with a few data points.
>
> Thus, we are keen to understand which aspects of the Neural Process framework you believe are unsuitable for such experiments and research directions.
>
> References
>
> [1] Ashman, M., Diaconu, C., Weller, A., & Turner, R. E. (2024). In-Context In-Context Learning with Transformer Neural Processes. arXiv preprint arXiv:2406.13493.

---

> ### Comment · Reviewer_utkY · 2024-11-22
> **Misconception**
>
> Thanks for the feedback. I ever read the mentioned paper "In-Context In-Context Learning with Transformer Neural Processes. " However, I'm afraid this work has some misunderstanding about the zero-shot learning or testing.
>
> Actually, **in-context learning is also kind of few-shot learning, which depends on the context information**. This is not typical zero-shot testing.

---

> ### Author Response · Authors · 2024-11-22
>
> Thank you for your prompt response.
>
> You may know that we are conducting in-context learning at the task level but not the data point level in our paper and [1]. This means that, without updating the weights, the Neural Process utilizes its pre-amortized knowledge to infer a new given task. In the ICL setting, inference may initially be inaccurate for the new task when relying solely on previously learned knowledge. However, by providing examples from new tasks, the model adjusts its inference, enabling more accurate predictions.
>
> In our paper, we conducted purely zero-shot experiments at the task level to demonstrate that when the model learns tasks of a similar nature (for example different dimensional GP tasks), it can still achieve strong performance even in purely zero-shot inference at the task level without additional examples, unlike the ICL setting.
>
> We are curious to hear your thoughts on this experimental setup and empirical validation. Additionally, we would like to understand which aspects of the Neural Process framework you believe this approach might conflict with.

---

### Official Review · Reviewer_P7fT · 2024-11-03

**Soundness:** 3
**Presentation:** 3
**Contribution:** 3
**Rating:** 6
**Confidence:** 2

**Summary:**

This work introduces a novel approach to meta-learning, specifically addressing the challenges of accommodating diverse input dimensions and learned features in Neural Process (NP) methods.
- The authors propose the Dimension Agnostic Neural Process (DANP), which incorporates a Dimension Aggregator Block (DAB) to transform input features into a fixed-dimensional space, enhancing the model's ability to handle varied datasets.
- By leveraging the Transformer architecture and latent encoding layers, DANP is capable of learning a broader range of features that are generalizable across different tasks.
- Through extensive experimentation on various synthetic and practical regression tasks, the authors demonstrate that DANP outperforms previous NP variations, effectively overcoming the limitations of traditional NP models and showcasing its potential for broader applicability in diverse regression scenarios.

**Strengths:**

- DANP is a novel extension of NP, that addresses the limitations of existing NP methods in handling diverse input dimensions and learned features.
- This work not only points out the shortcomings of current NP methods but also proposes a robust solution through the DAB and the integration of Transformer architecture.
- The paper is clear in its structure and presentation.

**Weaknesses:**

- The paper focuses on regression tasks, but its applicability to other tasks such as classification is not thoroughly explored. It could benefit from additional experiments or a theoretical discussion on how DANP might perform in non-regression tasks.
- While DANP shows promising results, the paper lacks a detailed discussion on the model's interpretability. The paper should include an analysis or discussion on how the components of DANP contribute to its predictions, especially given its complex architecture involving the DAB and Transformer-based latent path.

**Questions:**

- How does the authors' proposed DANP model perform in tasks outside of regression, such as classification or time-series forecasting? Are there any modifications needed for effective application in these domains?
- Are there any plans to conduct longitudinal studies to evaluate the long-term performance and stability of DANP in dynamic environments?

---

> ### Author Response · Authors · 2024-11-18
> **Rebuttal by Authors**
>
> **[W1, Q1] It could be benefit from additional experiments or a theoretical discussion on how DANP might perform in non-regression tasks. Are there any modifications needed for effective application in these domains?**
>
> Following the reviewer's insightful suggestion, we conducted additional experiments on time series data using the blood pressure estimation task from the MIMIC-III dataset. Specifically, we assumed real-world scenarios where certain features from patient data might be missing, or entirely different sets of features could be collected. Under this assumption, we trained the model using only a subset of features from the MIMIC-III dataset and evaluated its performance when additional or different sets of features became available.
>
> Specifically, we considered five features: T, Heart Rate, Respiratory Rate, SpO2, and Temperature. For pre-training, we utilized T and Heart Rate features, while for the fine-tuning scenario, we assumed only Respiratory Rate, SpO2, and Temperature features were available (this scenario can happen if we assume that we trained our model with data from hospital A and want to evaluate on the data in hospital B). We pre-trained the models with 32,000 training samples and fine-tuned them with only 320 samples. And here, we considered observations from time $0,...,t$ as context points and $t+1,...,T$ as target points. As shown in Table R.5, our DANP achieved strong performance in the time series blood pressure estimation task, demonstrating robustness and adaptability in this real-world scenario. These results are consistent with the findings presented in the main paper, further validating DANP's effectiveness in handling diverse and practical challenges.
>
> And also our model can, of course, be applied to classification problems as well. However, since output predictions in classification are typically better modeled with likelihoods other than a Gaussian distribution, it would be beneficial to adapt the decoder model structure accordingly. By redesigning the decoder, our approach can effectively handle classification tasks.
>
> __Table R.5.__ Empirical results on the time series blood pressure estimation task.
> |Finetuning Method   |   Full finetuning     |     |   Freeze finetuning       | |
> | :-                 | :-          | :-           | :-          | :-           |
> | MODEL              | CTX         |  TAR         | CTX         |  TAR         |
> | :-                 | :-          |      :-      | :-          | :-           |
> |  anp               | 1.037 ± 0.021 | 0.950 ± 0.017 | 1.035 ± 0.021 | 0.947 ± 0.019 |
> |  banp              | 1.104 ± 0.018 | 0.968 ± 0.011 | 1.100 ± 0.017 | 0.966 ± 0.012 |
> |  canp              | 0.964 ± 0.030 | 0.875 ± 0.024  | 0.962 ± 0.031 | 0.870 ± 0.022 |
> |  mpanp             | 1.012 ± 0.016 | 0.938 ± 0.018  | 1.010 ± 0.014 | 0.930 ± 0.010 |
> |  tnpd              | 1.165 ± 0.020 | 0.987 ± 0.013  | 1.160 ± 0.022 | 0.986 ± 0.011 |
> |  DANP             |**1.235**± 0.001|**1.184**± 0.006|**1.230**± 0.002|**1.180**± 0.005|

---

> ### Author Response · Authors · 2024-11-18
> **Rebuttal by Authors**
>
> **[W2]  The paper should include an analysis or discussion on how the components of DANP contribute to its predictions, especially given its complex architecture involving the DAB and Transformer-based latent path.**
>
> Thank you for the insightful comment. As you suggested, we believe that conducting an ablation study to examine how the DAB module and latent path affect model predictions provides a deeper understanding of each module's role. Consequently, in the main paper, Section 5.4 under the paragraph titled **The Role of the DAB Module and Latent Path**, we analyzed each module's function through ablation experiments.
>
> In summary, across various experimental settings, we observed that adding only the DAB module allows the model to handle data of varying dimensions, but it does not provide additional performance gains. In contrast, including the latent path results in performance improvements (indicating an enhanced capacity to learn shared features across tasks) but lacks the ability to manage varying dimensional data. These results demonstrate that the DAB module enables DANP to handle data with varying dimensions, while the latent path enhances the model’s ability to learn generally shared features across tasks of different dimensions.
>
>
> **[Q2] Are there any plans to conduct longitudinal studies to evaluate the long-term performance and stability of DANP in dynamic environments?**
>
> Thank you for inquiring about our future work plans. As demonstrated in our experiments with the MIMIC-III dataset, our ultimate goal is to develop a general foundation regressor model capable of handling a wide range of data structures and realistic scenarios, such as diverse time series data and cases with missing features. We view the DANP research as the initial step toward achieving this ambitious objective.
>
> A key focus of our future work will be to extend the model’s ability to appropriately process inputs with varying dimensions, numbers of context and target points, and diverse data structures (for example there can be lots of different tasks with the same dimensional inputs such as EMNIST image completion and 2d GP regression task). Developing a model that can flexibly adapt to such variability without specific data processing based on inductive bias while providing accurate and reliable inferences across these scenarios remains a critical challenge and an exciting direction for further exploration.

---

> ### Author Response · Authors · 2024-11-22
>
> Thank you for your dedication and interest in our paper. As the author and reviewer discussion period approaches its end, we are curious to know your thoughts on our rebuttal and whether you have any additional questions.

---

> > ### Comment · Reviewer_P7fT · 2024-11-26
> > **Reply for the authors**
> >
> > Thanks for your positive reply!
> > Additional experiments on the time series blood pressure estimation task better validate the proposed method.
> > Although the task setting introduces complex scenarios (feature missing, cross-hospital data distribution changes), the goal of blood pressure estimation determines that this is a regression task rather than a non-regression task.
> > Therefore, I still use the original score.

---

> ### Author Response · Authors · 2024-11-26
>
> Thank you for the positive comments on our paper. We promise to properly add the additional experiments and discussions conducted in the discussion period to the final manuscript.

---

### Official Review · Reviewer_8LUr · 2024-11-04

**Soundness:** 3
**Presentation:** 3
**Contribution:** 3
**Rating:** 6
**Confidence:** 3

**Summary:**

This paper introduces a new model, Dimension Agnostic Neural Process (DANP), which addresses the limitations of current Neural Processes (NP) in handling inputs with varying dimensions. DANP includes a Dimension Aggregator Block (DAB) that converts input features into a fixed-dimensional space, while also incorporating Transformer architecture and latent encoding layers to enhance adaptability across different tasks. Experimental results show that DANP performs well on multiple regression tasks, highlighting its potential as a versatile, general-purpose regressor.

**Strengths:**

1. Introduces the DAB module, enabling the model to handle inputs and outputs of varying dimensions, adding flexibility.
2. Covers multiple tasks and scenarios, demonstrating the model's stability across different conditions.
3. Performs well in regression, hyperparameter tuning, and other tasks, showing promise for broad applications.

**Weaknesses:**

1. Model Complexity. The design is complex, making replication and understanding challenging.
2. Lacks Analysis of Computational Costs. There’s no discussion of the model's time and resource requirements, impacting assessments for practical use.
3. Limited Application Scope. Primarily validated on regression tasks, with little exploration of classification or other tasks.

**Questions:**

I mainly have concerns about the practical aspects of this work. It would be helpful if the authors could provide more concrete examples and relate them to more complex, real-world applications.

---

> ### Author Response · Authors · 2024-11-18
> **Rebuttal by Authors**
>
> **[W1] Design is complex, making replication and understanding challenging**
>
> We appreciate the valuable feedback aimed at making our paper clearer and more accessible for readers. Based on the discussions during the rebuttal period, we plan to revise the content in the final manuscript to improve readability and clarity, ensuring it is organized in a way that facilitates better understanding.
>
> **[W2] There’s no discussion of the model’s time and resource requirements, impacting assessments for practical use.**
>
> We appreciate your insightful question regarding the time complexity. We will analyze the time complexity compared to the Transformer Neural Processes(TNP) both in theoretically and practically.
>
> First in theoretically, let us denote $B$, $N$, $d_x$, $d_y$, $d_r$, $L_d$, and $L_l$ denote the batch size, the number of data points (union of context and target), the dimension of input $x$, the dimension of output $y$, the representation dimension, the number of layers in the deterministic path, and the number of layers in the latent path, respectively. The additional computational cost for the DAB module is $O(BN(d_x+d_y)^2)d_r)$, and for the latent path, it is $O(L_lBN^2d_r)$. Since the computational cost for TNP is $O(L_dBN^2d_r)$, the overall computational cost of DANP can be expressed as $O((L_l+L_d)BN^2d_r)+O(BN(d_x+dy)^2d_r$. Generally, since $N >> (d_x+d_y)^2$ holds, the dominant term in the computational cost can be approximated as $O((L_l+L_d)BN^2d_r).
>
> For the practical time cost, we measure the time cost to train 5000 steps for the GP regression tasks and image completion tasks for TNP and DANP. The results are shown in Table R.3.
>
> __Table R.3.__ Wall clock time evaluation for the TNP and DANP in various settings. Here, we utilize RTX 3090 GPU for the evaluation.
> | MODEL | 1D regression  | 2D regression  | EMNIST      | CelebA     |
> |-------|----------------|----------------|---------------|---------------|
> | TNP   | 1 min 30 sec | 1 min 50 sec | 1 min |1 min 20 sec |
> | DANP  | 1 min 50 sec | 2 min 40 sec | 1 min 20 sec | 1 min 40 sec |

---

> ### Author Response · Authors · 2024-11-18
> **Rebuttal by Authors**
>
> **[W3, Q1] It would be helpful if the authors could provide more concrete examples and relate them to more complex, real-world applications**
>
> We appreciate your constructive suggestion regarding demonstrating DANP's applicability in additional practical scenarios in real-world applications. While the hyperparameter optimization task explored in our main paper is indeed a critical and highly relevant application, we agree that showcasing DANP's performance in other practical settings could further strengthen the contributions of our work. To validate the practicality, we conducted additional experiments on time series data using the blood pressure estimation task from the MIMIC-III dataset. Specifically, we assumed real-world scenarios where certain features from patient data might be missing, or entirely different sets of features could be collected. Under this assumption, we trained the model using only a subset of features from the MIMIC-III dataset and evaluated its performance when additional or different sets of features became available.
>
> Specifically, we considered five features: T, Heart Rate, Respiratory Rate, SpO2, and Temperature. For pre-training, we utilized T and Heart Rate features, while for the fine-tuning scenario, we assumed only Respiratory Rate, SpO2, and Temperature features were available (this scenario can happen if we assume that we trained our model with data from hospital A and want to evaluate on the data in hospital B). We pre-trained the models with 32,000 training samples and fine-tuned them with only 320 samples. And here, we considered observations from time $0,...,t$ as context points and $t+1,...,T$ as target points. As shown in Table R.4, our DANP achieved strong performance in the time series blood pressure estimation task, demonstrating robustness and adaptability in this real-world scenario. These results are consistent with the findings presented in the main paper, further validating DANP's effectiveness in handling diverse and practical challenges.
>
> __Table R.4.__ Empirical results on the time series blood pressure estimation task.
>
> |Finetuning Method   |   Full finetuning |         |   Freeze finetuning|        |
> | :-                 | :-          | :-           | :-          | :-           |
> | MODEL              | CTX         |  TAR         | CTX         |  TAR         |
> |  anp               | 1.037 ± 0.021 | 0.950 ± 0.017 | 1.035 ± 0.021 | 0.947 ± 0.019 |
> |  banp              | 1.104 ± 0.018 | 0.968 ± 0.011 | 1.100 ± 0.017 | 0.966 ± 0.012 |
> |  canp              | 0.964 ± 0.030 | 0.875 ± 0.024  | 0.962 ± 0.031 | 0.870 ± 0.022 |
> |  mpanp             | 1.012 ± 0.016 | 0.938 ± 0.018  | 1.010 ± 0.014 | 0.930 ± 0.010 |
> |  tnpd              | 1.165 ± 0.020 | 0.987 ± 0.013  | 1.160 ± 0.022 | 0.986 ± 0.011 |
> |  DANP             |**1.235**± 0.001|**1.184**± 0.006|**1.230**± 0.002|**1.180**± 0.005|

---

> ### Author Response · Authors · 2024-11-22
>
> Thank you for your dedication and interest in our paper. As the author and reviewer discussion period approaches its end, we are curious to know your thoughts on our rebuttal and whether you have any additional questions.

---

### Official Review · Reviewer_LwkF · 2024-11-08

**Soundness:** 3
**Presentation:** 3
**Contribution:** 3
**Rating:** 8
**Confidence:** 3

**Summary:**

The authors introduce Dimension Agnostic Neural Process (DANP) that incorporates Dimension Aggregator Block (DAB) to transform input features into a fixed-dimensional space, in an attempt to enhance the model's ability to handle diverse datasets. Leveraging a transformer architecture and latent encoding layers, the proposed approach learns a wide range of features, generalizable across various tasks.

To evaluate the approach the authors present a comprehensive evaluation, including synthetic and practical regression tasks. The empirical results, consisting of comparisons with exiting state-of-the-art methods and ablations show that the effectiveness of the proposed approach.  The authors outperforms past existing Neural Process methods, demonstrating advantages and improvements on GP regression, Image and Video Completion and Bayesian Optimization tasks.

**Strengths:**

Originality
 - The paper seem it has an evident level of novelty, tackling the diverse input and output dimensions challenge in the uncertainty aware meta-learning methods such as neural processes. Two novelties seems to be the case here, the dimension aggregation block and the latent path, in a transformer-like arhitecture.

Quality
- The paper is well motivated, structured and presented, the problem is well introduces and connected to existing work. The writing is good. There is extensive and diverse evaluation over synthetic and publicly available data sets for the GP regression, Image and Video Completion, and Bayesian Optimization tasks. The Ablation study is also useful.

Clarity
- The proposed approach is nicely presented and explained.

Significance
- The approach shows effectiveness of such methods under evaluated dataset and seems to hold potential for applicability in diverse regression scenarios

**Weaknesses:**

Presentation of the tasks/problems that the method addresses
- The presentation is sufficiently clear. I find that more on the actual task considered here can help to appreciate more the significance and the benefits of this approach. In particular related how the GP regression and the image completion tasks benchmarks help to validate the broad applicability of the approach?

Evaluations
- Non consistent result and seem to be marginal improvements in the GP Regression (from-scratch case) and the image completion task.

**Questions:**

In Table 1, it seems that the proposed approach has only marginal advantage over the proposed methods, in the GP Regression in the from-scratch case. Since, I'm able to see and it appears that between TNP and DANP (proposed approach) across 1D RBF, 1D Matern, 2D RBF and 2D Matern at the target column the only difference (improvement) is at the second (third) decimal.

What is the reason for that?

I would have expected higher improvement (e.g. on the first decimal). I'm not sure I can consider this to be statistically significant. Can I ask also whether confidence intervals are available and would it be possible to share them?

Similar performance is for the image completion task.

---

> ### Author Response · Authors · 2024-11-18
> **Rebuttal by Authors**
>
> **[W1] I find that more on the actual task considered here can help to appreciate more the significance and the benefits of this approach. In particular related how the GP regression and the image completion tasks benchmarks help to validate the broad applicability of the approach?**
>
> We appreciate your constructive suggestion regarding demonstrating DANP's applicability in additional practical scenarios in real-world application. While the hyperparameter optimization task explored in our main paper is indeed a critical and highly relevant application, we agree that showcasing DANP's performance in other practical settings could further strengthen the contributions of our work. To validate the practicality, we conducted additional experiments on time series data using the blood pressure estimation task from the MIMIC-III dataset. Specifically, we assumed real-world scenarios where certain features from patient data might be missing, or entirely different sets of features could be collected. Under this assumption, we trained the model using only a subset of features from the MIMIC-III dataset and evaluated its performance when additional or different sets of features became available.
>
> Specifically, we considered five features: T, Heart Rate, Respiratory Rate, SpO2, and Temperature. For pre-training, we utilized T and Heart Rate features, while for the fine-tuning scenario, we assumed only Respiratory Rate, SpO2, and Temperature features were available (this scenario can happen if we assume that we trained our model with data from hospital A and want to evaluate on the data in hospital B). We pre-trained the models with 32,000 training samples and fine-tuned them with only 320 samples. And here, we considered observations from time $0,...,t$ as context points and $t+1,...,T$ as target points. As shown in Table R.1, our DANP achieved strong performance in the time series blood pressure estimation task, demonstrating robustness and adaptability in this real-world scenario. These results are consistent with the findings presented in the main paper, further validating DANP's effectiveness in handling diverse and practical challenges.
>
> __Table R.1.__ Empirical results on the time series blood pressure estimation task.
>
> |Finetuning Method   |   Full finetuning     |     |   Freeze finetuning |        |
> | :-                 | :-          | :-           | :-          | :-           |
> | MODEL              | CTX         |  TAR         | CTX         |  TAR         |
> |  anp               | 1.037 ± 0.021 | 0.950 ± 0.017 | 1.035 ± 0.021 | 0.947 ± 0.019 |
> |  banp              | 1.104 ± 0.018 | 0.968 ± 0.011 | 1.100 ± 0.017 | 0.966 ± 0.012 |
> |  canp              | 0.964 ± 0.030 | 0.875 ± 0.024  | 0.962 ± 0.031 | 0.870 ± 0.022 |
> |  mpanp             | 1.012 ± 0.016 | 0.938 ± 0.018  | 1.010 ± 0.014 | 0.930 ± 0.010 |
> |  tnpd              | 1.165 ± 0.020 | 0.987 ± 0.013  | 1.160 ± 0.022 | 0.986 ± 0.011 |
> |  DANP             |**1.235**± 0.001|**1.184**± 0.006|**1.230**± 0.002|**1.180**± 0.005|

---

> ### Author Response · Authors · 2024-11-18
> **Rebuttal by Authors**
>
> **[W2, Q] The performance seems marginal compared to TNP in some cases. Share the confidence interval**
>
> Thank you for your comment regarding the performance and the confidence interval. We would like to first point out that our contribution is not about doing better than TNP for individual tasks, but showing that DANP is capable of processing variable dimension data seamleassely. Still, In response to your comment, while the performance figures in Table 1 (from-scratch GP regression) and Table 3 (image completion) may appear similar between DANP and TNP, it is important to note that DANP achieves differences in the second decimal place with the third decimal place standard deviations, which can be considered a significant improvement. More critically, we want to highlight that the DANP demonstrates substantial advantages in scenarios where zero-shot inference on new dimensional data or few-shot learning is required before inference. These strengths are clearly reflected in results such as Table 2, Table 3 (b), and the realistic Bayesian Optimization benchmark experiments shown in Figure 4.
>
> We also agree with the reviewer's suggestion that additional evaluation through various metrics could further validate the effectiveness of DANP. As such, we have conducted further analyses with additional metrics, which are detailed in Appendix B.2. To further compare DANP with other baseline models using alternative metrics, we provide results for two additional metrics: the Continuous Ranked Probability Score (CRPS) and Empirical Confidence Interval Coverage. These results can be found in Table 13 in Appendix B.2. In summary, for a target confidence interval, DANP demonstrates relatively narrower intervals, delivering the best results among the models. Moreover, when comparing CRPS scores, DANP achieves better scores than the other models.
>
> Also, we additionally measure some other metrics related to calibration. We measured and report the following 6 additional metrics:  1) Mean Absolute Error (MAE)**, 2) **Root Mean Square Error (RMSE)**, 3) **Coefficient of Determination ($R^2$)**, 4) **Root Mean Square Calibration Error (RMSCE)**, 5) **Mean Absolute Calibration Error (MACE)**, 6) **Miscalibration Area (MA)**. Except for $R^2$, lower values for all these metrics indicate better alignment with the target and improved calibration performance.  We conducted the evaluation using models trained on a 1d GP regression task, comparing our method with the baselines. The results, summarized in Table R.2, demonstrate that DANP achieves the best performance across a range of metrics. This observation reaffirms that DANP not only outperforms in terms of NLL but also achieves improved performance in calibration-related metrics compared to the baselines. These additional evaluations highlight the robustness of our method across diverse aspects of model performance. We will include this analysis in our revised manuscript to provide a more comprehensive evaluation of our approach.
>
> __Table R.2__ Results on additional metrics containing MAE, RMSE, $R^2$, RMSCE, MACE, and MA on 1d GP regression task. Except for $R^2$, lower values for all these metrics indicate better alignment with target and improved calibration performance.
>
> |Metric|MAE        |RMSE      |$R^2$      |RMSCE   |MACE      |MA           |
> |--------|-------------|--------------|-------------|--------------|--------------|-------------|
> |anp     |0.126±0.001|0.176±0.003 |0.788±0.012|0.273±0.007|0.238±0.003|0.240±0.003|
> |banp    |0.125±0.001|0.175±0.003|0.811±0.001|0.273±0.001|0.237±0.001|0.239±0.002|
> |canp   |0.127±0.001|0.178±0.002|0.801±0.005|0.267±0.008|0.239±0.002|0.237±0.005|
> |mpanp|0.124±0.001|0.173±0.003|0.807±0.005|0.274±0.014|0.242±0.007|0.244±0.008|
> |tnpd   |0.122±0.002|0.173±0.001|0.808±0.002|0.287±0.003|0.251±0.005|0.253±0.006|
> |danp  |**0.120**±0.001|**0.165**±0.002|**0.816**±0.002|**0.259**±0.002|**0.230**±0.003|**0.228**±0.002|

---

> ### Author Response · Authors · 2024-11-22
>
> Thank you for your dedication and interest in our paper. As the author and reviewer discussion period approaches its end, we are curious to know your thoughts on our rebuttal and whether you have any additional questions.

---

> > ### Comment · Reviewer_LwkF · 2024-11-25
> >
> > Dear authors,
> >
> > Thanks for your efforts, including explanations and additional evaluation.
> >
> > I find the approach a step forward in a more robust NP, with potential for further exploration for practical applicability, therefore I will  increase my score.

---

> ### Author Response · Authors · 2024-11-26
>
> Thank you for the positive comments on our paper. We promise to properly add the additional experiments and discussions conducted in the discussion period to the final manuscript.

---

### Author Response · Authors · 2024-11-18
**General Response**

Thank you to all reviewers for their insightful feedback. We are pleased that the novelty of our approach, including the dimension aggregation block (DAB) and latent path in a transformer-like architecture, was appreciated (LwkF). We are glad the extensive evaluation across various tasks and the utility of the ablation study were noted (LwkF). The flexibility introduced by DAB for handling variable dimensions and the model's stability across tasks was also recognized (8LUr). We appreciate the recognition of our clear presentation, the novel use of positional embeddings, and the value of submitting source code for reproducibility (1CSd, P7fT). Thank you for your constructive feedback and will continue to refine the paper based on these valuable insights. Thank you once again for your time and effort in reviewing our work.

---

### Author Response · Authors · 2024-11-28
**Revision of the paper**

We express our gratitude to all the reviewers who devoted their time and effort to help strengthen our paper and make it more self-contained through additional analyses, ablation studies, and new tasks demonstrating the usefulness of DANP. As the discussion period between reviewers and authors comes to an end, we want to inform you that we have incorporated the discussions and additional experiments into our revision. Because it requires careful reorganization for the paper to address Reviewer 1CSd's request for reorganization of the paper's contents and experiments as well as the inclusion of appropriate references for the appendix contents, it needs more time to reflect the reviewer's opinion. We **promise** to address these in our final manuscript to make the paper more readable and comprehensible.

Our revision consists of the following:

- Additional discussion on future work: Sec A
- Additional related works: Sec B
- Additional metrics: Sec D.2.1
- Ablation experiment on ELBO loss: Sec D.4
- Ablation experiment on RoPE: Sec D.5
- GP zero-shot experiment with different training dimension sets: Sec D.6
- Extrapolation experiment for the GP fine-tuning scenario: Sec D.7
- Additional extrapolation experiments for Image Completion: Sec D.9
- Time series experiments: Sec D.12
- Resource requirement analysis: Sec D.13

---

### Meta-Review · Area_Chair_Z4sz · 2024-12-20

**Metareview:**

The submission presents a novel approach to neural processes, introducing the Dimension Aggregator Block (DAB) module to handle dimension-agnostic tasks. The reviewers' initial assessment highlighted several strengths and weaknesses, including the need for more comprehensive experiments, clearer presentation of results, and addressing potential limitations. The authors' rebuttal and additional experiments effectively addressed many of the reviewer's concerns. The inclusion of new results, such as the comparison between sinusoidal PE and RoPE, and the evaluation of zero-shot performance on the video completion task, strengthened the paper. The authors also provided more detailed explanations and justifications for their design choices, which improved the clarity and readability of the manuscript. The reviewers' follow-up questions and comments further refined the discussion, leading to a more comprehensive understanding of the proposed approach. The authors' willingness to engage in a constructive dialogue and incorporate feedback demonstrated a commitment to improving the quality and presentation of their work. The revised manuscript is more self-contained, with a clearer presentation of results and a more comprehensive discussion of the limitations and potential future directions. After discussion with the authors and among themselves, the reviewers find the paper much improved, with four reviewers leaning towards acceptance and just one towards rejection. However, the one reviewer's concerns seem to have been mostly addressed by the rebuttal, as far as the AC can tell, and the main criticism seems to have been based on a misunderstanding regarding the notation. We are therefore happy to accept the paper. We would still like to encourage the authors to address all the reviewers' comments in the camera-ready version, especially regarding the notation for the parameters and latent variables and the distinction between global and local latent variables, also in relation to the notation used in other NP papers.

**Additional Comments On Reviewer Discussion:**

see above

---

### Decision · Program_Chairs · 2025-01-22

Accept (Poster)